# A dual spin-controlled chiral two-/three-dimensional perovskite artificial leaf for efficient overall photoelectrochemical water splitting

Hyungsoo Lee [1,2], Chan Uk Lee [1,2], Juwon Yun[1], Chang-Seop Jeong[1], Wooyong Jeong[1], Jaehyun Son[1], Young Sun Park [1], Subin Moon[1], Soobin Lee [1], Jun Hwan Kim[1] & Jooho Moon [1] ✉

The oxygen evolution reaction, which involves high overpotential and slow charge-transport kinetics, plays a critical role in determining the efficiency of solar-driven water splitting. The chiral-induced spin selectivity phenomenon has been utilized to reduce by-product production and hinder charge recombination. To fully exploit the spin polarization effect, we herein propose a dual spin-controlled perovskite photoelectrode. The three-dimensional (3D) perovskite serves as a light absorber while the two-dimensional (2D) chiral perovskite functions as a spin polarizer to align the spin states of charge carriers. Compared to other investigated chiral organic cations, $R$-/$S$-naphthyl ethylamine enable strong spin-orbital coupling due to strengthened π−π stacking interactions. The resulting naphthyl ethylamine-based chiral 2D/3D perovskite photoelectrodes achieved a high spin polarizability of 75%. Moreover, spin relaxation was prevented by employing a chiral spin-selective $L$-NiFeOOH catalyst, which enables the secondary spin alignment to promote the generation of triplet oxygen. This dual spin-controlled 2D/3D perovskite photoanode achieves a 13.17% of applied-bias photon-to-current efficiency. Here, after connecting the perovskite photocathode with $L$-NiFeOOH/$S$-naphthyl ethylamine 2D/3D photoanode in series, the resulting co-planar water-splitting device exhibited a solar-to-hydrogen efficiency of 12.55%.

Photoelectrochemical (PEC) water splitting, enabling the sustainable generation of solar-driven hydrogen fuel, has been recognized as a valuable environmentally friendly technology[1,2]. An overall water-splitting system consists of the hydrogen evolution reaction (HER) occurring at the photocathode and the oxygen evolution reaction (OER) taking place at the photoanode[3]. However, the OER process occurs via a sluggish multi-step reaction involving the transfer process of four electrons[4], which increases the complexity of the overall process[5]. The OER process involves spin-dependent chemical reactions in which the intermediate radicals must be in a spin co-aligned state to form triplet oxygen molecules ($^3O_2$) in a lower energy state (the $^3\Sigma_g^-$ triplet ground state) than their higher energy state (the $^1\Delta_g$ singlet excited state) singlet counterparts ($^1O_2$)[6,7]. Therefore, the reaction involving multiple electrons with uncontrolled spin states inevitably leads to the significant overpotential and sluggish kinetics of the OER[8], which has been identified as the limiting factor that causes low solar-

[1]Department of Materials Science and Engineering, Yonsei University, 50 Yonsei-ro Seodaemun-gu, Seoul 03722, Republic of Korea. [2]These authors contributed equally: Hyungsoo Lee, Chan Uk Lee. ✉e-mail: jmoon@yonsei.ac.kr

to-hydrogen (STH) efficiency for PEC water splitting. To address this problem, a distinct strategy based on spin-dependent electrochemistry (SDE), in which the spin state of the charge carrier is modulated using a magnetic field, has been reported[9–11]. Although SDE approaches utilizing magnetic materials for electrode and catalyst construction and employing external magnetic fields have provided valuable insight into the influence of the charge-carrier spin state on the OER process[12–14], challenges including economic feasibility and technical implementation continue to impede the practical integration of SDE into PEC water splitting.

In a recent development, it has been experimentally demonstrated that chiral organic molecules possessing a non-superimposable mirror image arrangement can be effectively utilized for manipulating spin-polarized current through a phenomenon known as chiral-induced spin selectivity (CISS)[15–17]. When a charge carrier passes through a chiral molecule, its spin orientation can become aligned in a particular direction due to the presence of an effective helical electric field generated by the chiral molecule[16,18]. Consequently, transmission of one specific spin-state is favored, in which preferred spin state is determined by the chiral axis of the medium (i.e., handedness of chiral materials) and the linear momentum of charge carrier, so called as spin flipping[17]. Owing to the intrinsic spin-control capability of chiral molecules, the CISS phenomenon not only mitigates the production of hydrogen peroxide as a by-product but also curtails the overpotential inherent in the OER process without needing to apply an external magnetic field. For example, Naaman et al.[19,20] reported that the $Fe_2O_3$ photoanode decorated with organic chiral molecules as a catalyst clearly enhanced the OER performance compared with one fabricated with organic achiral molecules. Despite the promising potential of chiral molecules used as a CISS enabler in their electrochemistry, limited performance is likely encountered in photoanodes coated with slow polaron-hopping mediated organic catalysts[21,22]. In order to enhance the low catalytic activity of organic chiral catalysts, a chirality-transferred $CoO_x$ catalyst has been developed to improve the OER performance of $BiVO_4$ photoanode[23]. However, such a photoanode generally exhibits a low photocurrent and high overpotential because the oxide semiconductor absorber retains poor light harvesting ability and electrical conductivity. The anisotropy of the spin-dependent current represents the spin-polarization degree ($P_V$), as defined by the following equation:

$$P_V = \frac{I_{down} - I_{up}}{I_{down} + I_{up}} \times 100\%, \qquad (1)$$

where $I_{down}$ and $I_{up}$ are the measured spin-dependent currents at specific voltage $V$ when the tip is pre-magnetized with either down or up field orientations, respectively. The spin polarization of the chiral catalysts remains relatively low at around 50%. Therefore, to fully exploit the capability of CISS in water splitting, it is imperative to replace not only the absorber possessing a higher light absorption coefficient but also the chiral material capable of improved spin-polarization efficiency[24].

In this regard, organic cation-based lead halide hybrid perovskites (OIHPs), recognized as the most promising light harvester in photovoltaics[25], have been adopted as an absorber layer in PEC applications[26,27]. The perovskite light absorber is first encapsulated and then coupled to a catalyst in an integrated PEC device for direct solar-to-chemical conversion when immersed in an electrolyte[28]. Subsequently, to induce the CISS effect, it is crucial to apply a chiral material as a spin polarizer (SP) with high spin-polarization efficiency on top of the perovskite light absorber layer. Two distinct enantiomers, namely (R)-(+)-α-methylbenzylamine (R-MBA) and (S)-(−)-α-MBA (S-MBA), have previously been employed as SPs[29,30]. The S-MBA SP exhibits a significantly improved CISS efficiency of 65% compared to other previously reported chiral catalysts[23]. However, due to the comparatively

weak helicity of the MBA organic cation arrangement, its spin-orbital coupling (SOC) essentially determined by the geometry of chiral organic ligands needs to be enhanced[31]. To reinforce this phenomenon, we utilized naphthyl ethylamine (NEA) having larger delocalized molecular orbitals than MBA, and thus capable of stronger π−π stacking interactions between the naphthalene rings. The introduction of NEA organic cations can improve the asymmetric distortion of inorganic layer, thereby leading to enhanced circular dichroism and improved chiroptical properties[32]. Furthermore, the incorporation of bulk organic cations is likely effective in promoting the water stability of a chiral 2D OIHP-based SP[33].

In this work, we demonstrate that the CISS phenomenon can be augmented by adopting NEA-based 2D chiral OIHP SP on top of the 3D OIHP photoanode. Our 2D chiral OIHP exhibited an impressive CISS efficiency of 75%, indicating an impressively enhanced spin-control capability compared to other chiral materials[23,30,34]. However, spin relaxation is inevitably prone to occur before the spin-polarized charges completely pass through the photoelectrode and participate in an electrochemical reaction[35]. It is essential to reorient the spin-polarized direction of charge carriers within the catalyst layer prior to the OER process. Therefore, an enantioselective NiFeOOH catalyst was employed to act as a chiral spin-selective catalyst, thereby enabling secondary spin filtration on the catalyst's surface. The chiral NiFeOOH catalyst (denoted as L-NF) was effectively synthesized by delicately manipulating the chirality inducible ligand molecules, which revealed superior catalytic activity to its achiral counterpart. Thus, to maximize the CISS effect in the OER process, we created a dual spin-controlled structure in which an NEA-based 2D chiral OIHP was directly deposited onto a 3D OIHP absorber layer, followed by the subsequent integration of the chiral spin-selective NiFeOOH catalyst as a secondary spin control. This dual spin-controlled chiral 2D/3D OIHP OER device (L-NF/S-NEA 2D/3D) achieved excellent oxygen evolution performance with an onset potential ($V_{onset}$) of 0.2 $V_{RHE}$ (the voltage vs. that of the reversible hydrogen electrode), a high fill factor, and a photocurrent of 23 mA cm$^{-2}$ at 1.23 $V_{RHE}$. An OIHP-based photocathode decorated with a spatially decoupled HER catalyst for unbiased solar water splitting was fabricated according to a previously reported method[36], which was subsequently connected in parallel with a dual spin-controlled chiral 2D/3D OIHP photoanode. Overall solar-driven water splitting was demonstrated by all of the fabricated OIHP-based photoelectrodes, resulting in a remarkable unassisted solar-to-hydrogen conversion efficiency (STH) of ~13% and continuous stable operation for 24 h, even when operated under a neutral pH electrolyte (0.5 M K-Pi, pH 6.5). Our noteworthy accomplishment of fabricating serially connected dual spin-controlled OIHP photoelectrodes provides a pathway for the rational design of chiral materials tailored for CISS-derived PEC water splitting.

## Results
### Fabrication of 2D chiral OIHP-derived SPs
To induce spin alignment in photogenerated carriers passing through the 3D OIHP absorber, it is crucial to select a 2D chiral OIHP with excellent spin-polarization efficiency. Besides the widely utilized MBA-based 2D chiral perovskite, we investigated two other chiral organic cations, (R)-(+)-sec-butylamine (R-BA) and (S)-(+)-sec-butylamine (S-BA) (characterized by the absence of a benzene ring), and NEA featuring an extra benzene ring. Circular dichroism (CD) analysis was employed to assess the chiroptic properties of MBA-, NEA-, and BA-based 2D chiral perovskites at wavelengths of 500, 390, and 420 nm, respectively (Fig. 1a). These wavelengths correspond to the absorption edges of the respective 2D chiral perovskites (Supplementary Fig. S1). The CD signal of BA-based 2D chiral OIHP shows a relatively smaller degree of ellipticity (1.9 mdeg) than the MBA-based 2D chiral OIHP (15.7 mdeg), implying that the former has lower chiroptical activity than the latter.

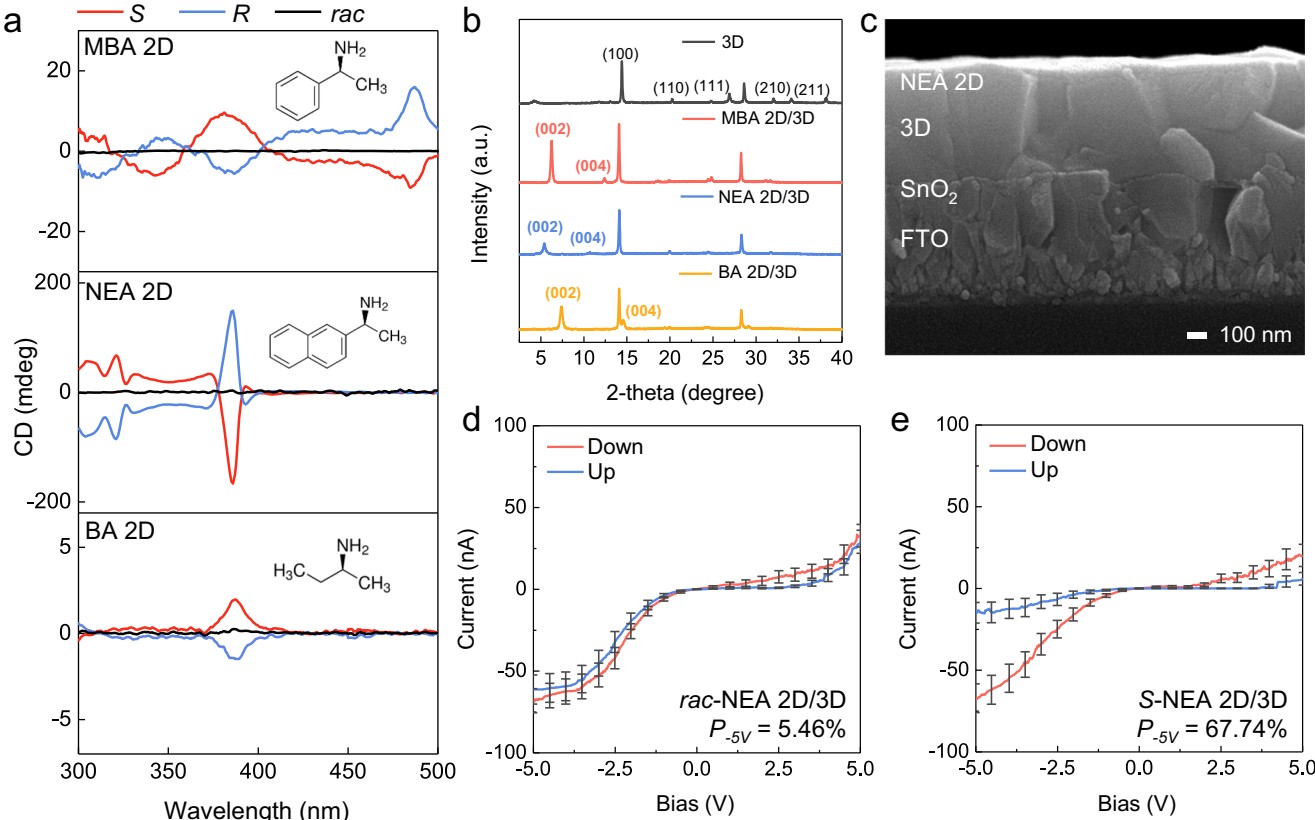

**Fig. 1 | The chiral characteristics of the 2D chiral OIHPs and their depositions onto 3D OIHPs. a** CD spectra of 2D chiral OIHPs obtained by using three different chiral molecules and then deposited on FTO. The molecular structures of the MBA, NEA, and BA 2D chiral OIHPs are illustrated in the inset images. **b** XRD spectra of 3D OIHP and the three different 2D chiral OIHP layers on 3D OIHP. **c** Cross-sectional SEM images of 2-NEA$_2$PbBr$_4$ (NEA 2D) on 3D (FAPbI$_3$)$_{0.95}$(MAPbBr$_3$)$_{0.05}$ deposited on an SnO$_2$/FTO substrate. mCP-AFM analysis to determine the spin-dependent currents for (**d**) *rac*-NEA 2D chiral OIHP and (**e**) *S*-NEA 2D chiral OIHP coated on 3D OIHP. The error bars in the mCP-AFM analysis represent the current range of 30 samples, while the solid line data represent the average values.

Chiroptical properties are strongly influenced by chiral-induced helical electric field, which is proportional to the radius of the helically arranged chiral cations. In this context, the smaller size of BA in comparison to MBA leads to the formation of a helix with lower asymmetric distortion, resulting in lower chiroptical activity[32]. On the other hand, NEA has a larger molecular structure and stronger π-π interactions than MBA, and thus the circular polarization of the noncoplanar arrangement of chiral cations in the former is enhanced owing to increased inorganic framework distortion[37]. As a result, the NEA-based 2D OIHP exhibited a notably improved chiroptical activity of 161.9 mdeg. However, as the chiroptic characteristics might not necessarily mean charge polarization ability, magnetic conducting probe atomic force microscopy (mCP-AFM) was conducted to estimate the anisotropy of the spin current (Supplementary Fig. S2, Supplementary Note 1). It was found that the 2D OIHPs utilizing a racemic mixture of MBA, NEA, or BA attained *P* of 4.31%, 4.99%, and 3.92%, respectively (Supplementary Fig. S3). This observation indicates that the racemic configuration of the 2D perovskite film exhibits achiral properties without prominent CISS. On the other hand, in the *S*-/*R*-configuration, all of the chiral films revealed significantly high $P_{-5V}$ surpassing 60%: those based on *S*-MBA and *S*-BA demonstrated commendable $P_{-5V}$ values of 71.30% and 66.26%, respectively, while those based on *S*-NEA and *R*-NEA exhibited notable $P_{-5V}$ of 75.53% and 73.45%, respectively. The spin polarization observed by these 2D chiral OIHPs is comparable to those of previously reported chiral perovskites owing to the huge CISS effect induced by the multiple quantum well-structured 2D chiral OIHPs. It is especially noteworthy that the NEA-based 2D chiral configuration exhibited the highest spin polarizability, which can be attributed to its enhanced SOC. Furthermore, its high SOC enables spin flipping and elastic scattering of charge carriers during spin propagation, thereby leading to high spin polarizability.

The 2D chiral OIHPs were heterogeneously grown on top of the 3D OIHP absorber. A chemical reaction occurred between the chiral organic molecules and the Pb ions of the 3D OIHP by dripping chiral cation salt-based ink on the 3D OIHP substrate, resulting in the formation of a chiral organic-based 2D OIHP layer (Supplementary Fig. S4). X-ray diffraction (XRD) spectroscopy verified successful deposition of the 2D chiral OIHP onto the 3D OIHP layer. As shown in the XRD spectrum in Fig. 1b, the (100) and (002) peaks at 14.41° and 28.61° indicate that the 3D OIHP had a composition of (FAPbI$_3$)$_{0.95}$(MAPbBr$_3$)$_{0.05}$[38]. After deposition of the molecular ink containing chiral organic cations on the 3D OIHP, distinct XRD peaks at 6.27° and 12.42° for MBA 2D[31], 5.41° and 10.72° for NEA 2D[33], and 7.37° and 14.58° for BA 2D[39] signify the successful growth of the highly crystalline 2D chiral perovskite layer along the [001] direction (the peaks at the lower and higher number of degrees correspond to the (002) and (004) planes, respectively). In the 2D XRD with general area detector diffraction system (2D GADDS XRD) analysis of the 3D OIHP in Supplementary Fig. S5a, ring-type peaks corresponding to the (001), (002), and (210) planes at 14.41°, 28.61°, and 35.01°, respectively, indicate that the crystal orientations are random. Meanwhile, after depositing the chiral molecules, the (002) and (004) planes of the 2D chiral OIHPs emerged as distinct dot-patterned peaks in the 2D GADDS XRD spectra in Supplementary Fig. S5b–d, which is consistent with the XRD spectrum in Fig. 1b. This observation suggests that the 2D chiral OIHPs have grown horizontally on top of the 3D OIHP regardless of

chiral molecule type. Under this circumstance, the charges photo-generated at the 3D OIHP can travel vertically within the 2D chiral structure in the out-of-plane direction, during which their spin states are readily polarized depending upon the handedness.

Scanning electron microscopy (SEM) revealed differences in the microstructure of bare 3D OIHP and 2D chiral OIHP-coated 3D OIHP. As shown in Supplementary Fig. S6a, the 3D OIHP layer had grown uniformly on the SnO$_2$/FTO substrate without pin-holes. After fabrication of 2D chiral OIHP on the 3D OIHP, it is observed that a thin film forms on top of the 3D perovskite, as shown in Supplementary Fig. S6b. Cross-sectional SEM images showed that the 3D perovskite was fabricated with a thickness of ~340 nm (Supplementary Fig. S6c). After the deposition of the NEA-based 2D chiral perovskite, a conformal thin film with a thickness of ~10 nm was observed on the upper surface of the 3D perovskite, as shown in Fig. 1c. To clarify the 2D and 3D perovskite, we performed the backscattered SEM images (COMPO mode). In the COMPO mode of cross-sectional SEM analysis, dark color typically signifies materials composed of light-weight atoms with low density. In Supplementary Fig. S6d, the prominent bright color observed in the 3D/SnO$_2$/FTO device indicates that the 3D perovskite is composed of heavy atoms with high density, whereas the 2D/3D/SnO$_2$/FTO device displays a thin and dark layer onto the 3D perovskite in Supplementary Fig. S6e. The thin and dark contrast of the 2D perovskite compared to the 3D perovskite is attributed to the lower density of the 2D perovskite, resulting from the presence of a greater quantity of organic cations in the layered structure. AFM topographical analysis confirmed comparable average roughness values of 3D OIHP and the NEA-based 2D/3D OIHP (16.65 and 15.94 nm, respectively) (Supplementary Fig. S7).

High-resolution transmission electron microscopy (HR-TEM) was conducted to investigate the crystallinity of the components in a hole transport layer (HTL)/2D chiral NEA OIHP (NEA$_2$PbBr$_4$)/3D OIHP ((FAPbI$_3$)$_{0.95}$(MAPbBr$_3$)$_{0.05}$) structure (Supplementary Fig. S8a). To this end, Supplementary Fig. S8b shows a magnified image of the orange box in Supplementary Fig. S8a, which was subjected to fast Fourier-transform (FFT) (the inset in Supplementary Fig. S8b). The lattices exhibit a lattice spacing of 6.35 Å indicative of the (100) plane of the 3D OIHP[38]. This observation is further supported by the reciprocal crystal lattice observable in the selected area electron diffraction (SAED) pattern in Supplementary Fig. S8c, in which the 3D OIHP manifests a spot diffraction pattern. Situated between the 3D OIHP and the HTL (the yellow dotted region), the NEA 2D chiral OIHP conformal layer with distinct crystallinity measures 10 nm in thickness. Supplementary Fig. S8d presents an HR-TEM image of the NEA 2D/3D OIHP region, showing a lattice spacing of 19.5 Å coinciding with the (002) plane of NEA 2D OIHP[40]. The crystallinity was also confirmed via FFT imaging of the white region. Furthermore, to calculate the crystal size of the 2D and 3D perovskites, we utilized the Scherrer equation:

$$\tau = \frac{K\lambda}{\beta\cos(\theta)} \quad (2)$$

$\tau$ represents the crystal size (in nanometers) of the ordered domains, $K$ is a dimensionless shape factor (typically assumed to be 0.9 for perovskites), $\lambda$ stands for the X-ray wavelength of Cu K$\alpha$ radiation (0.15405 nm), $\beta$ denotes the line broadening at full-width half the maximum intensity (FWHM), and $\theta$ is the Bragg angle. The FWHM of the (002) peak in the FFT image of the white region in Supplementary Fig. S8d was calculated using MATLAB (see Supplementary Fig. S9). For S-NEA 2D, the $\beta$ and $\theta$ values corresponding to the (002) plane were found to be 1.59 and 2.71 degrees, respectively, indicating a very small crystallite size of 0.0873 nm and thus a low crystallinity. This is a common issue that needs to be addressed in future research involving chiral 2D perovskites. The SAED pattern in Supplementary Fig. S8e showcases a pattern of spotted rings confirming the reciprocal lattices

of the thin 2D chiral and 3D perovskite components. These results confirm that the highly crystallized NEA 2D OIHP layer had developed uniformly.

To validate the operation of the CISS phenomenon at the chiral NEA 2D perovskite layer formed on top of the 3D perovskite layer, an I-V curve analysis was performed to evaluate the anisotropy of the spin-dependent current. The NEA racemic mixture (rac-NEA) 2D/3D OIHP device exhibited a relatively low $P_{-5V}$ of 5.46% (Fig. 1d) while a high $P_{-5V}$ of 67.74% was obtained with the S-NEA-based one, suggesting that the chirality of the NEA 2D OIHP was effectively preserved (Fig. 1e). Similar $P_{-5V}$ of 67.67% was also determined with the R-NEA 2D/3D OIHP device (Supplementary Fig. S10). Hence, it was confirmed that the spin-polarization efficiency was maintained even in the NEA-based chiral 2D/3D OIHP absorber configuration.

## The 3D OIHP photoanode with the NEA-based 2D OIHP spin polarizer

In a typical perovskite photovoltaic, SnO$_2$ and spiro-MeOTAD acting as the electron transport layer and HTL, respectively, facilitate charge separation through the formation of an excellent heterojunction with the 3D OIHP. However, spiro-MeOTAD intrinsically exhibits limited SOC, and consequently, the polarized charge carriers experience rapid spin relaxation, leading to a diminished CISS effect. After depositing spiro-MeOTAD on top of the S-NEA 2D/3D OIHP, mCP-AFM analysis revealed a low $P_{-8V}$ of 2% due to an exceedingly short spin-propagation length (Supplementary Fig. S11a). To effectively induce the spin state control of charge carriers prior to the electrochemical reaction, a comprehensive exploration of various hole transport materials was performed. Poly(3-hexylthiophene-2,5-diyl) (P3HT) was selected because of its considerable spin diffusion length of ~1 μm[41], which can be attributed to the highly organized structure of thiophene within P3HT inducing strong π − π interactions. The various mixed HTLs with different volume ratios of spiro-MeOTAD:P3HT (S:P) from 92:8 (denoted S92:P8) to 84:16 (S84:P16) were prepared to evaluate the preservation of the CISS phenomenon. The spin polarizability ($P_{-8V}$) gradually increased from 14% to 46% as more P3HT was blended with spiro-MeOTAD (Supplementary Fig. S11b−d). After developing a mixed HTL capable of mitigating spin relaxation, a water-splitting device needed to be implemented to evaluate whether the CISS phenomenon influences PEC performance. To prevent the degradation of the hygroscopic OIHP from electrolyte penetration, the HTL/2D/3D OIHP photoanode was covered with Ni foil. This physical passivation approach offers multiple advantages because Ni foil not only acts as a structural barrier against moisture penetration but also provides a rigid surface to which the pre-synthesized catalyst can be conveniently attached. To establish an electrical connection between the Ni foil and the HTL, Ketjen (a highly conductive type of carbon) was incorporated to facilitate interlayer bonding. To investigate the extent of spin relaxation caused by Ni foil and Ketjen, we fabricated the devices passivated with a bilayer of Ni foil/Ketjen (i.e., Ni/Ketjen/S88:P12/NEA 2D/3D OIHP structure) to observe spin-polarized current depending upon the handedness of chiral NEA 2D (Supplementary Fig. S12). Even with the passivation of Ni/Ketjen, S-NEA 2D exhibited a spin polarization degree ($P_{-10V}$) of 45.88% (Supplementary Fig. S12a), while R-NEA 2D showed a $P_{-10V}$ of 42.33% (Supplementary Fig. S12b), indicating that even with a passivation layer, the spin is maintained without significant spin relaxation during propagation. However, when using rac-NEA 2D, the $P_{-10V}$ decreased to 1.42% (Supplementary Fig. S12c), suggesting the absence of the CISS effect. Although the addition of the Ni/Ketjen layer results in a slight decrease in the degree of spin polarization, the spin relaxation time of carbon nanoparticles is sufficiently long ( ~100 ns) to enable spin transport[42,43], leading to the manifestation of the CISS-driven OER at the electrode/electrolyte interface. To complete photoanode fabrication, the NiFeOOH catalyst deposited on nickel foam was affixed to Ni foil using Ag paste (Fig. 2a).

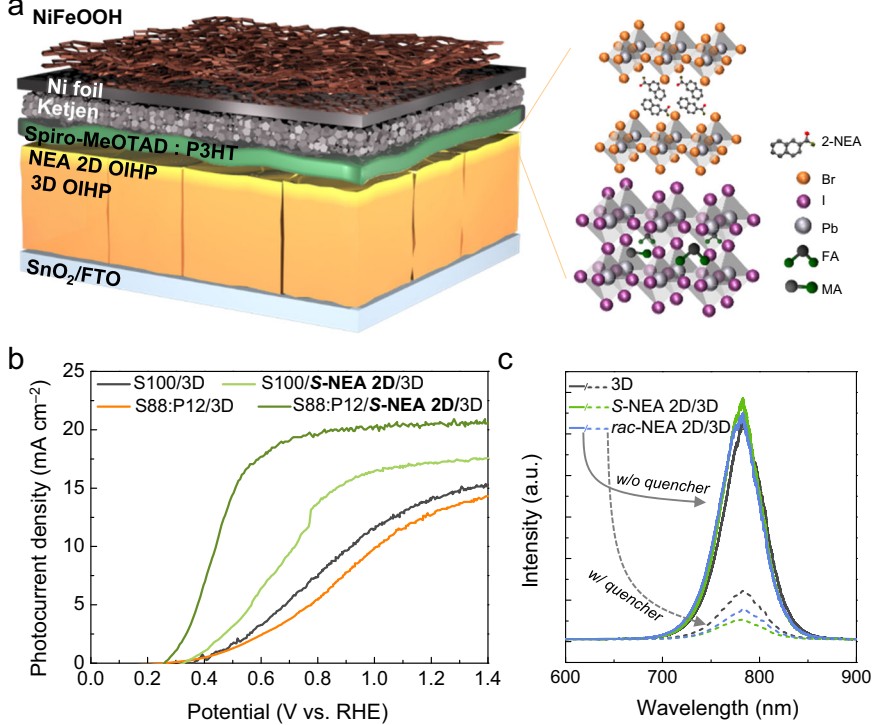

**Fig. 2 | The PEC performance of the 2D chiral OIHP-based photoanodes and the effect of spin polarization therein. a** A schematic illustration of a photoanode comprising NiFeOOH/Ni/Ketjen/HTL/NEA-2D/3D/SnO₂/FTO layers. **b** LSV curves of the OIHP photoanodes without and with the *S*-NEA 2D OIHP layer using various spiro-MeOTAD and P3HT formulations for the HTL conducted in K-Pi electrolyte (pH 6.5) under 1 sun illumination with an active area of 0.06 cm². **c** Steady-state PL spectra of 3D OIHP, *S*-NEA 2D/3D OIHP, and *rac*-NEA 2D/3D OIHP films with (w/) and without (w/o) a quencher on SLG or an SnO₂ substrate with an S88:P12 HTL, respectively.

To verify the CISS effect on OER performance, two photoanodes, one with 3D OIHP only (NiFeOOH/Ni/Ketjen/HTL/3D/SnO₂/FTO) and one with the *S*-NEA-based 2D/3D OIHP structure (NiFeOOH/Ni/Ketjen/HTL/*S*-NEA 2D/3D/SnO₂/FTO), were prepared. Figure 2b shows linear sweep voltammetry (LSV) measurements of the photoanodes in 0.5 M potassium phosphate buffer (K-Pi) electrolyte (pH 6.5). All of the photoanodes were analyzed for the same light absorption area of 0.06 cm², as defined by a metallic photomask (Supplementary Fig. S13a). Additionally, the active area of the catalyst also matched 0.06 cm². A photocurrent density ($J_{ph}$) of 14.27 mA cm⁻² at 1.23 V_RHE along with a $V_{onset}$ of 0.4 V_RHE was observed when the S100:P0 HTL was utilized in a photoanode without the *S*-NEA-based 2D perovskite layer (the black curve in Fig. 2b). On the other hand, an isotype hetero-junction formed between the *S*-NEA 2D chiral and 3D OIHPs, giving rise to a back-surface field at the heterojunction that effectively sup-pressed charge-carrier recombination and thereby prolonged their lifespan[44,45]. Therefore, the *S*-NEA-based 2D/3D OIHP photoanode even with S100:P0 HTL showed an acceptable PEC performance (a $J_{ph}$ of 17.19 mA cm⁻² at 1.23 V_RHE and a $V_{onset}$ of 0.33 V_RHE; the bright green curve in Fig. 2b). Gradually increasing the amount of P3HT in the HTL incrementally improved both the photocurrent and fill factor of the photoanode (Supplementary Fig. S13b). Moreover, the S88:P12 HTL in the *S*-NEA-based 2D/3D OIHP photoanode provided an improved PEC performance (a $J_{ph}$ of 20.54 mA cm⁻² at 1.23 V_RHE and a $V_{onset}$ of 0.26 V_RHE; the dark green curve in Fig. 2b). This enhancement in OER per-formance can be attributed to the S88:P12 HTL being capable of suf-ficient spin propagation (a $P_{-8V}$ of 43%, as shown in Supplementary Fig. S11c). The photoanode containing the *R*-NEA-based 2D chiral OIHP with the S88:P12 HTL exhibited the remarkable PEC performance (a $J_{ph}$ of 20.12 mA cm⁻² and a $V_{onset}$ of 0.32 V_RHE, Supplementary Fig. S13c). However, when more than 12 vol% P3HT was mixed with spiro-MeOTAD (e.g., S84:P16), a decreased OER performance was obtained despite higher spin-propagation capability (a $P_{-8V}$ of 46%, as shown in Supplementary Fig. S11c) due to the insufficient hole extraction ability of P3HT[46]. The 3D OIHP photoanodes without the NEA 2D chiral OIHP (the orange curve in Fig. 2b) and with *rac*-NEA 2D (Supplementary Fig. S13d) when using S88:P12 HTL provided $V_{onset}$ values of 0.41 at 0.38 V_RHE along with $J_{ph}$ values of 12.7 and 14.9 mA cm⁻², respectively. These results reveal that the spin polarization effect of the *S*-NEA-based 2D OIHP enables rapid transportation of spin-controlled charge carriers toward the semiconductor-liquid junction (SCLJ) without experiencing back-recombination.

To estimate the band alignment and to evaluate the effect of the band bending according to the handedness of NEA 2D, the band energy position of all the layers in the photoanode was obtained by ultraviolet photoelectron spectroscopy (UPS), as shown in Supple-mentary Fig. S14a. Details of the experimental procedure and relevant results are provided in Supplementary Fig. S14b, c and Supplementary Note 2. The relative band energy positions of the 3D, NEA 2D, HTL, and SnO₂ for the pre-equilibrium and equilibrium states are plotted in Supplementary Fig. S15a, b, respectively. Firstly, it is observed that both *S*-NEA 2D and *rac*-NEA 2D are positioned at the same energy levels, including the valence band maximum (E_VBM), Fermi energy level (E_F), and conduction band minimum. This indicates that the perfor-mance difference between the NiFeOOH/Ni/Ketjen/HTL/*S*-NEA 2D/3D/SnO₂/FTO photoanode (Fig. 2b) and the NiFeOOH/Ni/Ketjen/HTL/*rac*-NEA 2D/3D/SnO₂/FTO photoanode (Supplementary Fig. S13d) is not attributed to the band alignment mismatch of *S*-NEA 2D and *rac*-NEA 2D. Another notable observation is that the E_F of the 3D OIHP layer is approximately 0.2 eV higher than that of the NEA 2D OIHP layer. This leads to a back-field induced by the p-p heterojunction, explaining the enhanced performance of the S100/NEA 2D/3D photoanode although the use of spiro HTL having low spin propagation ability likely miti-gates the CISS effect. After reaching an equilibrium state, a Schottky

junction likely forms between the 2D and 3D OIHP heterojunction, resulting in a slight band energy offset. Thin layer (10 nm) of chiral NEA 2D is responsible for the observed performance enhancement through electron tunneling effect, as observed in Supplementary Fig. S16. However, as thickness increases, performance decreases. Therefore, because there is no difference in band alignment depending upon the different handedness of NEA 2D, the enhanced performance of the NiFeOOH/Ni/Ketjen/HTL/*S*-NEA 2D/3D/SnO$_2$/FTO photoanode compared to the NiFeOOH/Ni/Ketjen/HTL/*rac*-NEA 2D/3D/SnO$_2$/FTO photoanode is not originated from the interfacial energetics. Instead, the enhancement in photocurrent and onset potential stems from the introduction of a chiral spin polarizer, inducing the CISS effect.

We fabricated 3D OIHP photovoltaic (PV) cells in a configuration of Au/S88:P12/(NEA 2D/)3D/SnO$_2$/FTO structure, as shown in Supplementary Fig. S17a, b, and conducted PV characterization. When comparing champion PV cells, the 3D PV cell without NEA 2D exhibited notably lower performance (Supplementary Fig. S17c). Conversely, the 3D PV cell with an inserted NEA 2D showed similar performance regardless of the handedness of the chiral NEA molecule. Even when statistics were compiled for 10 PV cells, the NEA 2D/3D structure consistently showed higher average values, whereas PV parameters decreased when the NEA 2D layer was absent (Supplementary Fig. S18). The incorporation of NEA 2D OIHP improves the photovoltaic characteristics due to the passivation effect. However, since *rac*-, *R*-, and *S*-NEA 2D have identical interfacial energetics, the photovoltaic characteristics remain unaffected by the chirality of NEA 2D. Consequently, the enhancement in PEC performance upon adding chiral *S*-NEA 2D onto the 3D PEC device is primarily attributed to the CISS effect rather than photovoltaic performance enhancement.

Steady-state photoluminescence (PL) spectroscopy was conducted to investigate the behavior of photoexcited carriers within the devices with or without an SP. During front-excitation when photons are injected into the NEA 2D/3D OIHP interface (Supplementary Fig. S19a), PL was observed in devices without a quenching layer (i.e., the 3D/soda-lime glass (SLG) and NEA 2D/3D/SLG devices). Both devices revealed a prominent peak at a wavelength of 780 nm in the PL spectra in Fig. 2c, which is an emission peak due to radiative recombination. Compared to the device without an SP, the introduction of the *rac*-NEA-based 2D SP caused a slight increase in the PL intensity whereas using the *S*-NEA-based 2D SP provided the highest PL intensity, thereby revealing the suppressed recombination of photogenerated charges in the devices with the SP. On the other hand, when back-excitation was employed (as depicted in Supplementary Fig. S19b) wherein photons are excited toward the interface between the substrate and the 3D OIHP, the absence of a quenching layer resulted in no significant difference in PL intensity (Supplementary Fig. S19c). However, after introducing both the HTL and SnO$_2$ quencher layer under front-excitation, the *S*-NEA 2D/3D OIHP film showed the lowest PL intensity, indicating rapid charge extraction (the dashed line in Fig. 2c). The results of the PL analysis indicate that the introduction of the *S*-NEA 2D SP enhances not only the lifetime of the spin-polarized charge carriers but also the charge extraction kinetics. We additionally conducted time-resolved photoluminescence (TRPL) measurements to quantitatively evaluate the recombination rate. The devices having a configuration of 3D, *S*-NEA 2D/3D, and *rac*-NEA 2D/3D films on SLG were utilized for the measurements, as demonstrated in Supplementary Fig. S20. It is well-known that when the excitation fluence exceeds $10^{15}$ carriers per cm$^3$, both mono- and bimolecular recombination coexist. In the case of the TRPL without a quencher, the fast lifetime ($\tau_1$) possibly implies the non-radiative bimolecular recombination of charge carriers, while the slow lifetime ($\tau_2$) represents the non-radiative monomolecular recombination of the carriers by trapping or detrapping. Therefore, the obtained curves were fitted with bi-exponential decay curve and the fitting results are summarized in Supplementary Table S1. Both $\tau_1$ and $\tau_2$ for the *S*-NEA 2D/3D and *rac*-

NEA 2D/3D films were found to be longer as compared to the 3D film without NEA 2D. These results suggest that the lifetime of photogenerated charge carriers was effectively elongated by introduction of NEA 2D, which likely results from the reduced defect density at the 2D/3D interface. Furthermore, the recombination process between bimolecular and monomolecular can be discernible by comparing the weight fraction of the two lifetime values. The weight fraction of $\tau_2$ in 3D film is much higher (63.07%) than that of *S*-NEA 2D/3D (40.60%) and *rac*-NEA 2D/3D (40.19%). This speculates that the 3D OIHP without the NEA 2D exhibits higher monomolecular recombination induced by higher trapping mechanism as compared to the 3D OIHP with NEA 2D. This observation highly supports our research in which the introduction of chiral NEA 2D OIHP results in the elongated lifetime of photogenerated charge carriers.

### A highly efficient *S*-NEA 2D/3D OIHP photoanode with the *L*-NiFeOOH spin-selective catalyst

The OER process was improved by incorporating the *S*-NEA 2D SP with a charge polarization efficiency of 43%. However, spin relaxation likely occurs during the charge transport to the SCLJ, suggesting the necessity for further enhancing the charge polarization efficiency at the catalyst/electrolyte interface. In this regard, we exploited the chirality of the NiFeOOH catalyst. When achiral *meso*-tartaric acid (TA) was used in the fabrication of *meso*-NiFeOOH (NF), the latter covered Ni foam poorly due to inhomogeneous growth, as observed via SEM (Supplementary Fig. S21a). This is because weak covalent bonds formed between the metal cations and the carboxylate group in *meso*-TA, thereby providing a non-directional structure due to its random interactions during metal cation binding and dimer formation. Consequently, this leads to a non-uniform micro-structured *meso*-NF. On the other hand, *L*-/*D*-TA, which is a chiral molecular structure, forms direction-dependent dimer interactions due to stronger covalent bonding of ligand complex than *meso*-TA. The chiral ligand molecules make it difficult to stack them in an unfavorable conformation and their resulting homogeneity provides good coverage when grown on Ni foam. Furthermore, enantiomeric TA, which forms a dimer structure depending on the chirality, can effectively modify the reactivity of electrodeposition by inducing a preferential orientation at a specific reaction site, resulting in a chirality-transferred catalyst[23]. The *L*-/*D*-NF chiral catalyst in the presence of *L*-/*D*-TA exhibited a uniform microstructure (the SEM image in Supplementary Fig. S21b), which was further confirmed via energy dispersive X-ray (EDX) mapping images in Supplementary Fig. S22 showing that Fe and O were evenly distributed on the Ni foam. In addition, the (012) plane of NiFeOOH provided a peak at 37.2° in the XRD pattern shown in Supplementary Fig. S23a. Meanwhile, the Raman spectrum in Supplementary Fig. S23b shows the Ni-O vibration mode at 490 and 512 cm$^{-2}$ and the Fe-O vibration mode at 521 cm$^{-2}$. Furthermore, we performed X-ray photoelectron spectroscopy (XPS) analysis. The results revealed identical XPS spectra for both *L*-NF and *meso*-NF. Peaks corresponding to Ni$^{3+}$ $2p_{1/2}$ and $2p_{3/2}$ were observed at 867.5 and 854.1 eV, respectively, while a peak at 851.6 eV originated from Ni$^0$ in the Ni foil substrate (Supplementary Fig. S23c). Similarly, peaks for Fe$^{3+}$ $2p_{1/2}$ and $2p_{3/2}$ were observed at 722.4 and 709.9 eV, respectively (Supplementary Fig. S23d). Finally, the O 1$s$ peak was deconvoluted into peaks at 530.8 and 529.5 eV (Supplementary Fig. S23e), representing the metal-O bond (O$^{2-}$) and metal-hydroxyl bond (-OH), respectively. These results serve as additional evidence confirming the successful formation of NiFeOOH. These observations signify the successful synthesis of the high-quality chiral NF catalyst. LSV sweeps were performed at a current density of 10 mA cm$^{-2}$ to compare the electrochemical performances of the chiral and achiral NF catalysts (Fig. 3a), as detailed in a previous study[45]. The potentials necessary to achieve a current density of 10 mA cm$^{-2}$ through water oxidation electrolysis were determined as 1.71 and 1.79 V$_{RHE}$ for the chiral *L*-NF and achiral *meso*-NF catalysts,

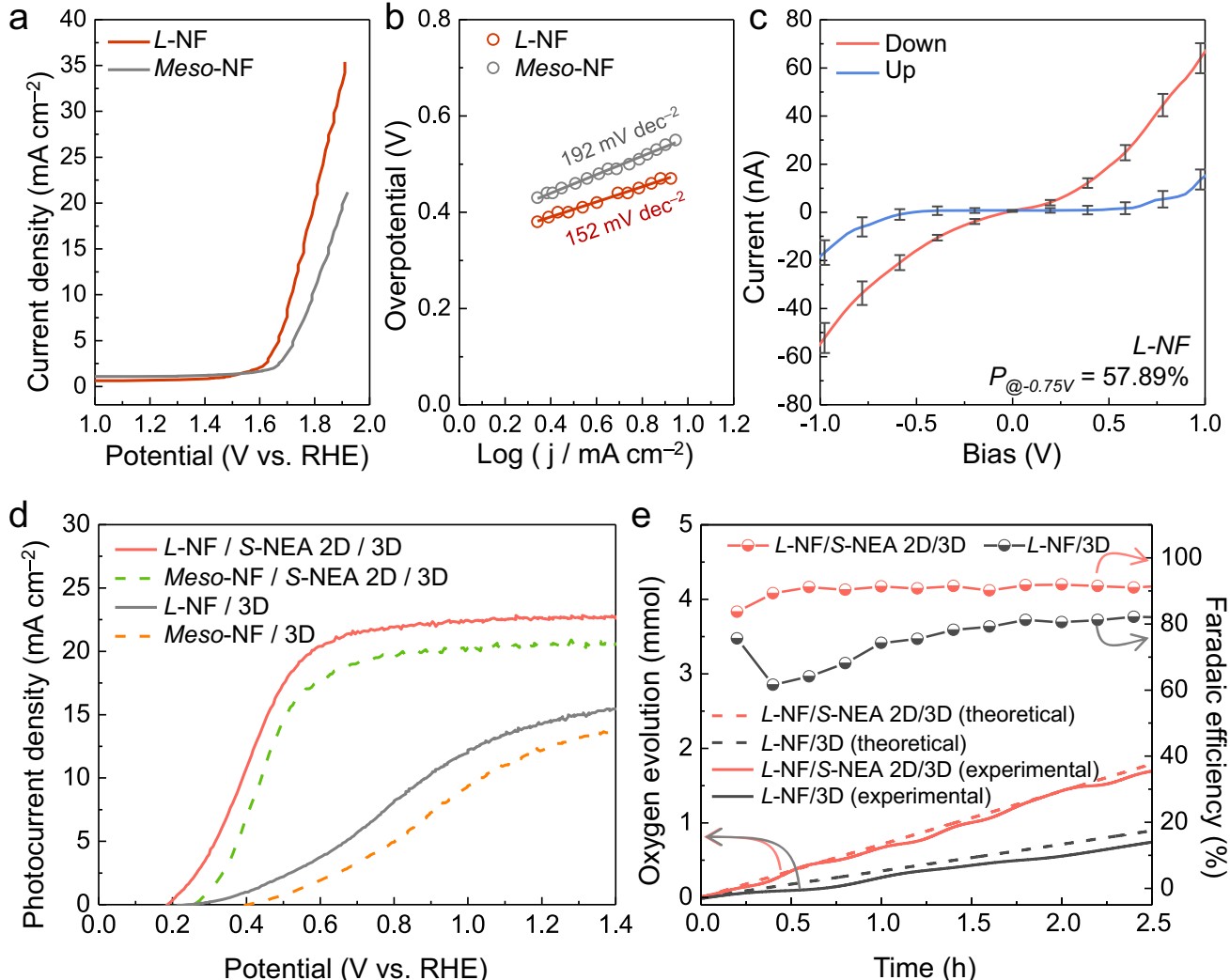

**Fig. 3 | The chirality of the prepared NiFeOOH catalysts and their application in OIHP-based photoanodes. a** LSV curves and (**b**) the corresponding Tafel plots of *L*-NF (dark red) and *meso*-NF (gray) on Ni foam. The voltage was not iR corrected. **c** The spin-polarization degrees of *L*-NF/Ni and *meso*-NF/Ni measured using mCP-AFM. The error bars represent the current range of 30 samples, while the solid line data represent the average values. **d** LSV curves of *L*-NF/*S*-NEA 2D/3D (red), *meso*-NF/*S*-NEA 2D/3D (green), *L*-NF/3D (gray), and *meso*-NF/3D (orange) photoanodes with an active area of 0.06 cm². **e** O₂ generation during the PEC reaction by the *L*-NF/*S*-NEA 2D/3D and *L*-NF/3D photoanodesdetected using an oxygen sensor. The Faradaic efficiency of O₂ generation by both photoanodes was calculated by considering the theoretical O₂ generation. All of the electrochemical measurements were conducted in 0.5 M K-Pi (pH 6.5) under 1 sun illumination.

respectively. In addition, the estimated Tafel slope of *L*-NF (152 mV dec⁻²) was lower than that of *meso*-NF (192 mV dec⁻²), thereby suggesting the more rapid electrochemical reaction kinetics associated with the chiral catalyst. These results demonstrate that spin selectivity can significantly reduce the overpotential required for the OER process.

CD analysis was performed to evaluate whether the synthesized *L*-NF retains its chirality (Supplementary Fig. S24). While *meso*-NF exhibited no variation in CD signal according to wavelength, both *L*-NF and *D*-NF displayed obvious CD signals in opposing directions at 530 nm, which corresponds to the light absorption edge. In addition, the *L*-/*D*-NF catalysts showed high $P_{-0.75V}$ of 57.89% and 56.48%, respectively, determined by mCP-AFM (Fig. 3c, Supplementary Fig. S25a), which were far superior to *rac*-NF and *meso*-NF ($P_{-0.75V}$ of 1.87% and 0%, respectively; Supplementary Fig. S25b). This signifies that the chirality of *L*-/*D*-NF is not an intrinsic property but rather induced by the presence of either *L*-TA or *D*-TA, respectively. To eliminate the influence of the microstructural difference between *meso*-NF and *L*-NF (Supplementary Fig. S21), their electrochemical surface area (ECSA) was obtained via cyclic voltammetry (CV) at

various scan rates ranging from 10 to 50 mV s⁻¹ (Supplementary Fig. S26). The measured electrical double-layer capacitance ($C_{dl}$) of *L*-NF (0.1626 μF cm⁻²) was considerably lower than that of *meso*-NF (0.2433 μF cm⁻²), indicating that the ECSA of *meso*-NF is much higher (Supplementary Fig. S27a; calculation details in Supplementary Note 2)[47,48]. From the LSV curves in Supplementary Fig. S27b, it is evident that the overpotential of *L*-NF remained lower than that of *meso*-NF, thereby supporting that the *L*-NF chiral catalyst exhibits superior electrochemical catalytic activity owing to the spin-selective capability.

Photoanodes containing a spin selective catalyst (SSC) and 2D chiral and 3D OIHP layers were configured as either *L*-NF/Ni/*S*-NEA 2D/3D OIHP/SnO₂/FTO (denoted as *L*-NF/*S*-NEA 2D/3D) or *L*-NF/Ni/3D OIHP/SnO₂/FTO (denoted as *L*-NF/3D). For comparison, photoanodes without SSC were configured as *meso*-NF/Ni/3D OIHP/SnO₂/FTO (denoted as *meso*-NF/3D) and *meso*-NF/Ni/*S*-NEA 2D/3D OIHP/SnO₂/FTO (denoted as *meso*-NF/*S*-NEA 2D/3D). The *L*-NF/3D photoanode showed a higher $J_{ph}$ by 2 mA cm⁻² and smaller overpotential by 0.14 V compared to the *meso*-NF/3D photoanode (Fig. 3d), which is attributed to the spin-selective *L*-NF catalyst on which the spin-aligned OH

radicals undergo the rapid OER process. Similarly, the dual spin-controlled $L$-NF/$S$-NEA 2D/3D photoanode exhibited a superior OER performance ($J_{ph}$ ~ 22.56 mA cm$^{-2}$ and $V_{onset}$ ~ 0.20 V$_{RHE}$) over the *meso*-NF/$S$-NEA 2D/3D photoanode ($J_{ph}$ ~ 20.67 mA cm$^{-2}$ and $V_{onset}$ ~ 0.27 V$_{RHE}$). Moreover, judging from the performance of the *meso*-NF/$rac$-NEA 2D/3D photoanode ($J_{ph}$ ~ 14.01 mA cm$^{-2}$ and $V_{onset}$ ~ 0.34 V$_{RHE}$, Supplementary Fig. S28), it can be inferred that the enhancement in PEC performance is not solely attributed to the introduction of 2D OIHP. This can be attributed to the spin polarization of charge carriers when passing through the $S$-NEA 2D SP followed by additional spin orientation by the $L$-NF catalyst. This dual spin alignment synergistically enhances the PEC performance of the photoanode. To clarify this aspect, we have evaluated the PEC performance using LSV curves for the various combinations of $R$-/$S$-NEA 2D and $L$-/$D$-NF SCCs, each possessing different chirality handedness. When the chirality configuration of $S$-NEA 2D is changed to $R$-type, the photocurrent and onset potential of the $L$-NF/$R$-NEA 2D/3D photoanode significantly decrease, as indicated by red-dotted line in Supplementary Fig. S29. This indicates that the spin orientation induced by $L$-NF and $R$-NEA 2D is in opposite directions. Furthermore, as evident in Supplementary Fig. S29, $L$-NF and $R$-NEA 2D serve as opposite spin polarizers inducing down-spin and up-spin, respectively. Consequently, using $L$-NF together with $S$-NEA 2D to induce the same direction of down-spin carriers, the PEC performance is greatly enhanced. Similarly, the performance also improves when using $D$-NF and $R$-NEA 2D, inducing the same direction of up-spin carriers. However, the spin-controlled photoanode revealed the decreased photocurrent density when stacking $D$-NF and $S$-NEA 2D, which presumably manipulates the opposite directions of spin state. To confirm that the current density generated by the photoanode is not the result of side reactions, incident photon-to-current conversion efficiency (IPCE) measurement at 1.23 V$_{RHE}$ was conducted. The edge of the IPCE plot in Supplementary Fig. S30a corresponds to the absorption edge of the 3D OIHP (Supplementary Fig. S30b). Remarkably, the integrated current density of 22.8 mA cm$^{-2}$ closely matches the $J_{ph}$ of the $L$-NF/$S$-NEA 2D/3D photoanode in Fig. 3d. The applied-bias photon-current conversion efficiency (ABPE) can be used to compare the PEC performances of photoanodes. Owing to efficient charge transfer of the spin-polarized carriers from the OIHP layers to the electrolyte, our dual spin-controlled $L$-NF/$S$-NEA 2D/3D photoanode exhibited a remarkable ABPE of 13.17% (Supplementary Fig. S31), even when operated under a neutral pH electrolyte, which is higher than current state-of-the-art OIHP-based photoanodes (Supplementary Table S1).

The stability of the dual spin-controlled OER device was analyzed in 0.5 M K-Pi (pH 6.5) at 1.23 V$_{RHE}$ under simulated 1-sun continuous illumination. The device retained 80% of the initial current density for 160 h before experiencing a rapid decline in performance (Supplementary Fig. S32a). To identify the cause of the abrupt performance degradation after long-term operation, the front and rear of the sample were examined (Supplementary Fig. S32b). While no catalyst detachment or Ni foil delamination was observed at the front photoelectrode, a color change from black to yellow was observed in the area of the 3D OIHP at the back side. This suggests that the 3D OIHP layer degrades during electrochemical reaction. Other studies have indicated that unextracted excess electrons are a major contributing factor to iodine migration-induced degradation[49], with iodine vacancies having the lowest activation energy for migration compared to other ions, making them more likely to migrate first to the electron-accumulated interfacial region[50]. Consequently, when iodine ions migrate toward the HTL, they trigger oxidation reactions, leading to the formation of PbI$_2$ and degradation of the perovskite. To confirm that catalyst degradation was not the main cause of stability decline, the LSV curves of $L$-NF before and after stability measurements were compared (Supplementary Fig. S32c). The $L$-NF sustained nearly similar current density even after 160 h, supporting that the catalyst was

not the primary cause of degradation. We tested 20 samples, and the device performance statistics determined via the corresponding J–V curves confirmed its reproducibility and reliability (Supplementary Fig. S33). An oxygen sensor was used to precisely detect the product (O$_2$) generated during the PEC reaction (Fig. 3e), after which the O$_2$ formation efficiency without producing hydrogen peroxide was calculated. The $L$-NF/3D photoanode revealed the O$_2$ generation Faradaic efficiency of 81.9%, whereas $L$-NF/$S$-NEA 2D/3D photoanode exhibited significantly higher O$_2$ Faradaic efficiency of 90.6%. This observation clearly suggests the success of implementing the CISS effect in our dual spin-controlled photoanode so that the photogenerated charges appropriately generate the desired product (O$_2$) while effectively suppressing the hydrogen peroxide side reaction. To confirm the suppression of by-product (H$_2$O$_2$) as an evidence for spin-controlled OER, we conducted colorimetric experiment to quantify the generated amount of H$_2$O$_2$ by observing the time-dependent absorbance spectra of the electrolyte aliquot collected as a function of reaction duration, as shown in Supplementary Fig. S34. The absorbance arises from the presence of a reactant between o-tolidine and H$_2$O$_2$ so that the absorption varies depending upon the amount of H$_2$O$_2$ in the electrolyte. Colorimetric experiment was conducted under potentiostatic condition in 0.5 M Na$_2$SO$_4$ electrolyte, enabling the detection of generated H$_2$O$_2$ at 1.0 V$_{RHE}$. Furthermore, we utilized an oxygen sensor to detect O$_2$ simultaneously, allowing us to calculate the FE$_{O2}$ as a function of reaction duration, as illustrated in Supplementary Fig. S35. For the 3D OIHP photoanode utilizing *meso*-NF catalysts, the FE$_{O2}$ was calculated to be 73%, while the FE for H$_2$O$_2$ (FE$_{H2O2}$) was 16%, as shown in Supplementary Fig. S35a. Even with the addition of *racemic*-type achiral 2D on top of 3D OIHP (Supplementary Fig. S35b), the FE$_{O2}$ and FE$_{H2O2}$ remained similar at 72% and 14%, respectively. However, upon the insertion of $S$-NEA-based chiral 2D OIHP (Supplementary Fig. S35c), the FE$_{O2}$ increased to 78%, while the FE$_{H2O2}$ decreased to 8%, indicating that the CISS effect works on OER device. Moreover, replacing *meso*-NF with chiral $L$-NF further enhanced the FE$_{O2}$ and reduced the FE$_{H2O2}$, achieving FE$_{O2}$ and FE$_{H2O2}$ of 90% and 3%, respectively (Supplementary Fig. S35d). This clearly demonstrates that the dual spin control strategy manipulates the spin state of charge carriers for an extended period, leading to the suppression of by-product H$_2$O$_2$ and the formation of lower-energy triplet oxygen, resulting in a significant improvement in FE$_{O2}$.

## Elucidating the effect of dual spin control on PEC performance

To examine the charge-carrier transport behavior at the interface of the NEA based chiral 2D OIHP, we performed electrochemical impedance spectroscopy (EIS) on the $L$-NF/3D, $L$-NF/$S$-NEA 2D/3D, and $L$-NF/*rac*-NEA 2D/3D photoanodes at 0.4, 0.8, and 1.2 V$_{RHE}$ in the range of 0.1 Hz to 300 kHz under 1 sun illumination. The Nyquist plots of the OIHP photoanodes without (Fig. 4a) or with the NEA 2D layer (Fig. 4b, c) clearly contain at least two arcs, with the large and small ones apparent in the ranges of 15–50 kHz (the high-frequency region) and 0.5–1 Hz (the low-frequency region), respectively. The PEC parameters were extracted from the EIS results by adopting a simple Voight circuit model (Supplementary Fig. S36; interpretation of the equivalent circuit model is provided in Supplementary Note 3). The large arc in the high-frequency region is the area-specific resistance at high frequency ($R_{HF}$), which is responsible for the photogenerated charge transport influenced by the charge extraction kinetics within the multi-layered photoanode, while the small arc in the low-frequency region is the area-specific resistance ($R_{ct}$) representing the charge transfer resistance resulting from the spin-dependent radical-mediated PEC reaction. With the NEA-based 2D OIHP layer, a sub-arc in the middle frequency of 10 Hz (the area-specific resistance; $R_{inter}$) appeared (Supplementary Fig. S37b, c), which represents the intermediate state at the interfaces acting as a one-way spin-transport channel while suppressing back-recombination of the charge carriers[51].

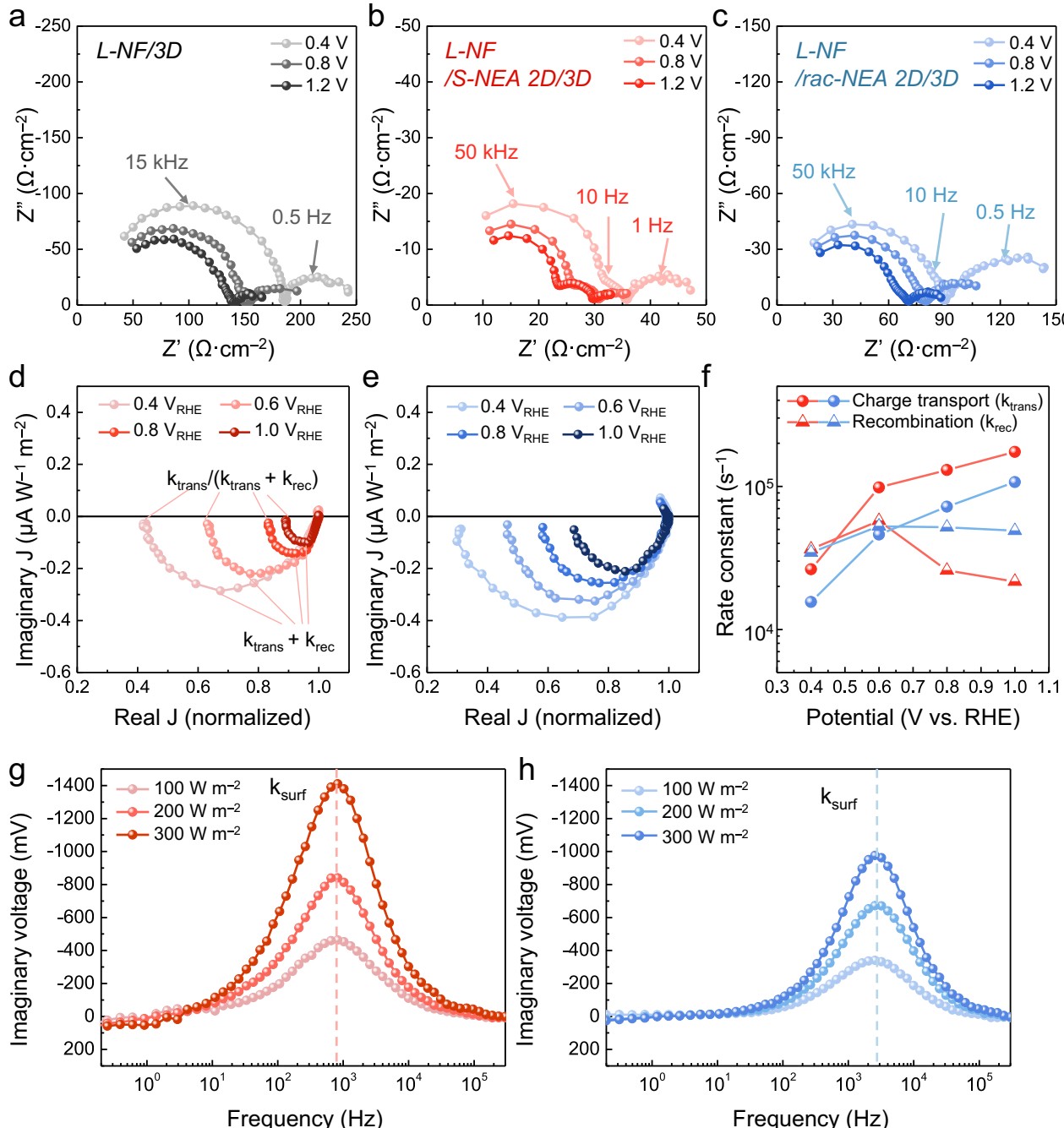

**Fig. 4 | Disclosure of the CISS phenomenon in the spin-dependent PEC device.**
Nyquist plots of the EIS data for the (**a**) *L*-NF/3D, (**b**) *L*-NF/*S*-NEA 2D/3D, and (**c**) *L*-NF/*rac*-NEA 2D/3D photoanodes in K-Pi electrolyte (pH 6.5) under one sun illumination in the potential range from 0.4 to 1.2 V$_{RHE}$. IMPS spectra of the (**d**) *L*-NF/*S*-NEA 2D/3D, and (**e**) *L*-NF/*rac*-NEA 2D/3D devices and (**f**) their charge-transport and recombination-rate constants extracted from the IMPS data in the potential range from 0.4 to 1.0 V$_{RHE}$. **g, h** The imaginary component of the complex photovoltage in the IMVS spectra of the three OIHP photoanodes in 0.5 M K-Pi electrolyte (pH 6.5) under one sun illumination.

As the applied bias was increased from 0.4 to 1.2 V$_{RHE}$ (Fig. 4a), the $R_{HF}$ and $R_{ct}$ values of the *L*-NF/3D device decreased from 168.9 to 115.3 $\Omega\cdot cm^2$ and 61.24 to 31.02 $\Omega\cdot cm^2$, respectively (Supplementary Table S2). These resistance reductions signify enhanced charge transport and suppression of the side reaction in the *L*-NF/3D photoanode at higher bias voltages. However, the *L*-NF/3D photoanode still displays higher resistance with respect to the 3D photoanodes with NEA 2D OIHP. Furthermore, the apex frequency of the semicircle in the Bode plot of the *L*-NF/3D photoanode in Supplementary Fig. S37a is 15 kHz, which is lower than those for the *L*-NF/*S*-NEA 2D/3D and *L*-NF/*rac*-NEA 2D/3D photoanodes (both 50 kHz). This indicates the faster charge-transport kinetics with the NEA 2D chiral OIHP layer. Comparing the Nyquist plot of the devices in Fig. 4a confirmed that introducing the NEA-based 2D OIHP layer produced an additional arc in the middle-frequency region (10 Hz, as mentioned above) regardless of the chirality of the 2D OIHP (Fig. 4b, c). The Bode plots of the photoanodes with the NEA-based 2D OIHP layer in Supplementary Fig. S37b, c also reveal a noticeable peak at a characteristic frequency of ~10 Hz. The $R_{inter}$ of the *L*-NF/*S*-NEA 2D/3D photoanode steadily increased from 6.00 to 9.94 $\Omega\cdot cm^2$ when increasing the applied bias, which implies that the *S*-NEA 2D OIHP layer effectively provides an intermediate channel inducing high spin state, resulting in the efficient propagation

of spin-polarized charges and the reduction of elastic backscattering-induced recombination due to Pauli exclusion. In contrast, the opposite behavior of $R_{inter}$ is observed for the $L$-NF/$rac$-NEA 2D/3D photoanode, where $R_{inter}$ gradually decreases with the applying bias (from 8.01 to 6.10 $\Omega \cdot cm^2$). This suggests that it was difficult for the spin-uncontrolled charge carriers to propagate from the intermediate channel in the $rac$-NEA 2D chiral OIHP layer owing to the low spin state. As the applied bias was increased, both $R_{HF}$ and $R_{ct}$ decrease in $L$-NF/$S$-NEA 2D/3D and $L$-NF/$rac$-NEA 2D/3D devices. However, the decreased $R_{inter}$ of the $L$-NF/$rac$-NEA 2D/3D device implies that back-recombination is not prevented due to the absence of the spin polarization capability, resulting in a higher overall resistance. This is consistent with the enhancement in photocurrent suggested by the LSV measurements.

To gain an in-depth understanding of the dual spin-control layer, we performed intensity modulated photocurrent/photovoltage spectroscopy (IMPS/IMVS) to evaluate the rate constants for charge recombination and transfer in the PEC devices under an applied potential and open-circuit conditions, respectively[52]. The IMPS analysis provided data to evaluate the charge-transport kinetics and interface charge recombination rate within the photoanode due to the presence of an SP. The IMVS analysis reveals surface recombination kinetics at the SCLJ to elucidate the lifetime of the photogenerated charges under the open-circuit condition. Figure 4d, e exhibit Nyquist plots revealing the intricacy of the photocurrents in the $L$-TA/$S$-NEA 2D/3D and $L$-TA/$rac$-NEA 2D/3D photoanodes under light modulation at different applied potentials. From the plots, the charge transfer efficiency can be calculated by determining the ratio of the real photocurrents at the low- and high-frequency intercepts, as follows:

$$k_{trans}/(k_{trans} + k_{rec}), \qquad (3)$$

where $k_{trans}$ and $k_{rec}$ are the bulk charge-transport and interface-recombination rate constants, respectively[53]. Thus, by normalizing the real photocurrent to the high-frequency intercept, the low-frequency intercept directly reflects the charge transport efficiency. The rate constants at the apex of the semicircle, where the maximum phase shift is measured, indicate the combined rate of charge transport and recombination ($k_{trans} + k_{rec}$). By utilizing the low-frequency intercept and apex frequency, both $k_{trans}$ and $k_{rec}$ can be obtained, as shown in Fig. 4f. The calculation method for $k_{trans}$ and $k_{rec}$ is specified in Supplementary Note 4 and Supplementary Fig. S38. As the anodic potential increased, $k_{trans}$ increased and $k_{rec}$ decreased in both photoanodes. $k_{trans}$ surpasses $k_{rec}$ after 0.8 $V_{RHE}$. However, in the $L$-NF/$S$-NEA 2D/3D photoanode, the intersect of $k_{trans}$ and $k_{rec}$ occurs at the lower potential of 0.6 $V_{RHE}$. This observation suggests that the spin polarization effect induced by the $S$-NEA 2D SP facilitates rapid charge transport even under lower bias conditions. At 1.0 $V_{RHE}$, the differences between $k_{trans}$ and $k_{rec}$ for the $L$-NF/$rac$-NEA 2D/3D OIHP and $L$-NF/$S$-NEA 2D/3D devices were two-fold and eight-fold, respectively, which implies that the $L$-NF/$S$-NEA 2D/3D device achieved enhanced charge separation and generated spin-aligned OH radicals. The spin-polarized OH radical readily undergoes a rapid kinetic reaction to evolve $^3O_2$, followed by an accelerated Volmer-Tafel process, consequently leading to an improved fill factor[54].

In order to confirm the improvement in the PEC reaction when spin-controlled OH radicals are present on the photoanode's surface due to the dual spin-control layer, IMVS analysis was performed as shown in the Bode plots in Fig. 4g, h. the peak at high frequency (in the kilohertz range) showing the maximum value of the imaginary photovoltage represents the characteristic rate constant for surface charge recombination ($k_{rec}$). Thus, the characteristic time constant

($\tau_{rec}$) at which recombination occurs can be calculated as follows[55]:

$$\tau_{rec} = (1/k_{rec}) = (1/2\pi v_{rec}) \qquad (4)$$

The time constants for the $L$-NF/$rac$-NEA 2D/3D and $L$-NF/$S$-NEA 2D/3D devices were 31.45 and 97.64 μs, respectively, speculating that the spin-polarized charge carriers in the latter are less susceptible to the back-scattering-mediated surface recombination. The imaginary voltage of $L$-NF/$S$-NEA 2D/3D device consistently increased as the light was intensified from 100 to 300 $W\,m^{-2}$. This result suggests the absence of trap-assisted or Auger non-radiative recombination, signifying that surface recombination occurs primarily via radiative recombination. In addition, the imaginary voltage of $L$-NF/$S$-NEA 2D/3D photoanode was higher than that of the $L$-NF/$rac$-NEA 2D/3D photoanode. Hence, it is speculated that the $L$-NF/$S$-NEA 2D/3D photoanode retains large amounts of spin-aligned charge carriers at the active sites (high spin state) on the catalytic surface, thereby enabling their effective utilization in the spin-dependent OER process and promoting the production of $^3O_2$ from the spin-parallel OH radicals. Thus, the introduction of S-NEA resulted in an extended surface charge lifetime that enables full exploitation of the spin-polarized OH radicals in the OER process.

## All OIHPs artificial leaf for unassisted water splitting

Since our dual spin-controlled $L$-NF/$S$-NEA 2D/3D photoanode demonstrates excellent performance and durability, we opted to apply it to unbiased solar water splitting under parallel dual illumination. We fabricated an OIHP-based photocathode as the counterpart electrode by simply relocating the Cu wire position from the bottom to the top electrode. The structure of the OIHP photocathode is depicted in Supplementary Fig. S39. The active area, which refers to the catalytic interface, is fully separated from the absorber layers in both OIHP photoelectrode, allowing the water redox reactions to take place independently from the OIHP absorber layers[36]. The OIHP photocathode and the dual spin-controlled OIHP photoanode were assembled in a co-planar structure (depicted in Fig. 5a), enabling water splitting through solar irradiation. In Fig. 5b, the LSV curves of the remarkable performing $L$-NF/$S$-NEA 2D/3D photoanode and the Pt/OIHP photocathode in 0.5 M K-Pi (pH 6.5) overlap to determine the maximum operation point of the OIHP-based co-planar photocathode–photoanode system: the curves intersect at 20.7 $mA\,cm^{-2}$ at 0.81 $V_{RHE}$. Due to the co-planar configuration, the integrated active area comprises both the photocathode and photoanode, resulting in the expected operating current being halved to 10.35 $mA\,cm^{-2}$ (Fig. 5c). The operational PEC performance of the co-planar photocathode–photoanode unbiased water-splitting device was assessed using a two-electrode configuration under 1-sun illumination by scanning from –1.0 to 1.0 V as shown in Fig. 5c; it exhibited a current density of 10.2 $mA\,cm^{-2}$ at 0 V (unbiased condition), which closely matches the estimated operational current density. Compared with the J-V curve of the OIHP photoanode (Fig. 5c), the co-planar configured unbiased photoelectrode revealed an anodic shift of approximately 1 V induced by the Pt/OIHP photocathode, implying that the OIHP photocathode provides a 1 V photovoltage for unassisted solar water splitting. Moreover, the operational current of the photoelectrode corresponds to an STH efficiency of 12.55%. As shown in Supplementary Fig. S40, an unassisted water-splitting system driven solely by natural sunlight was achieved by serially connecting the OIHP photocathode and photoanode using Cu wire and tape, while Supplementary Movie 1 demonstrates that solar water splitting using the co-planar photocathode–photoanode system is a feasible technology. Potentiostatic chronoamperometry analysis of the co-planar photocathode–photoanode system was conducted under continuous illumination to evaluate its long-term stability (Fig. 5d); it retained 100% of its initial current density after 24 h. Gas

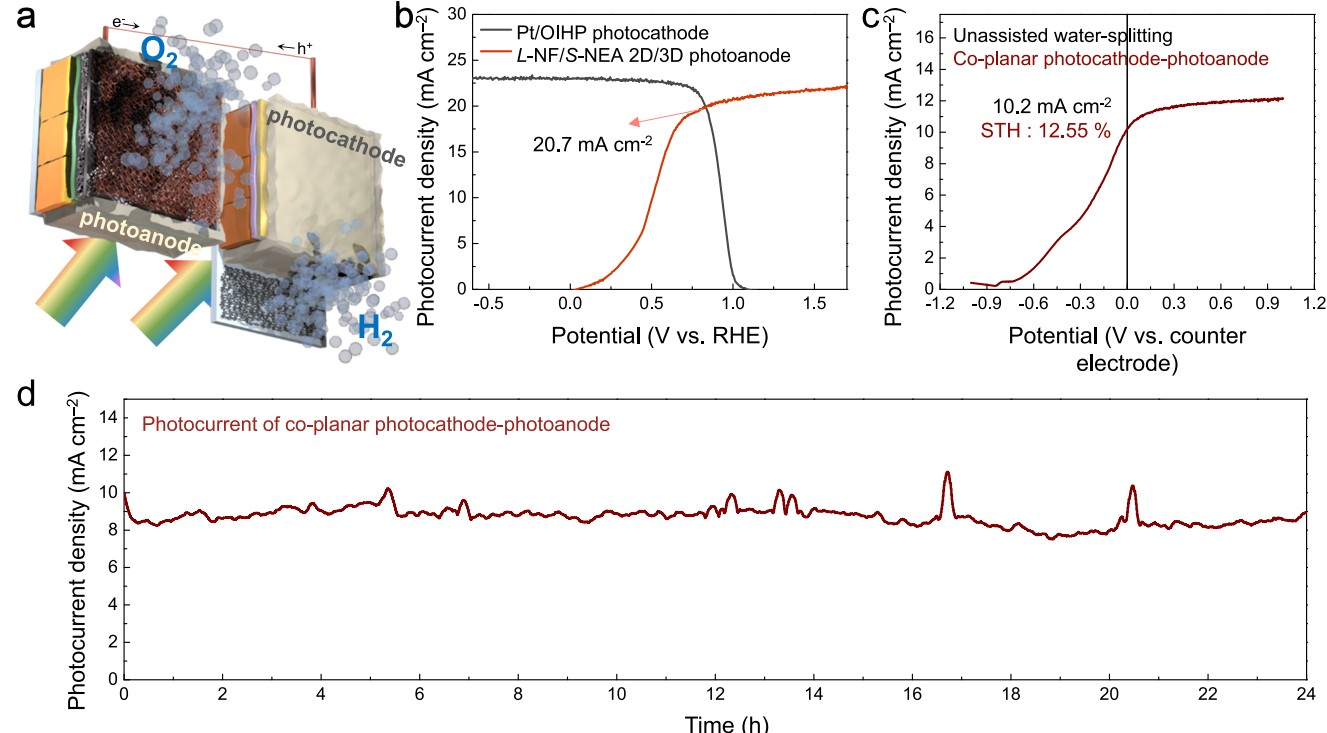

**Fig. 5 | PEC performance and operational stability of the co-planar photo-cathode-photoanode water-splitting system. a** A schematic illustration of the co-planar-configured device. **b** LSV curves of the *L*-NF/*S*-NEA 2D/3D photoanode–Pt/OIHP photocathode in 0.5 M K-Pi (pH 6.5) under 1 sun illumination. The operating points and active areas (0.06 cm²) of the co-planar-configured unbiased photoelectrodes are indicated. **c** An LSV curve and (**d**) long-term stability of the co-planar photocathode-photoanode device in 0.5 M K-Pi (pH 6.5) under 1-sun dual illumination without an applied bias. It had an integrated active area of 0.12 cm² (the sum of those of the photocathode and photoanode).

chromatography was conducted to quantitatively assess the products from unbiased solar water-splitting (Supplementary Fig. S41), indicating that hydrogen and oxygen gases were generated at rates of 11.2 and 5.6 μmol h⁻¹ with the active areas of 0.06 cm² for the each photoelectrodes, respectively. Notably, the Faradaic efficiencies of hydrogen and oxygen evolution were 98.78% and 91.40%, respectively. Hence, our unassisted water-splitting device exhibited excellent water-splitting performance that outperforms previous OIHP-based photoelectrodes and is even comparable to the STH efficiency achieved with expensive III-V group semiconductors (Supplementary Fig. S42). Details of the overall PEC–PEC water-splitting devices shown in Supplementary Fig. S42 are summarized in Supplementary Table S3.

## Discussion

We successfully demonstrated spin-dependent performance enhancement of water splitting by creating and employing a dual spin-control strategy-based perovskite photoanode. The sluggish kinetics of the oxygen evolution reaction were promoted by exploiting the CISS effect in the photoanode, which also suppressed side reactions and intermediate hydrogen peroxide formation. The CISS effect was introduced by growing an *S*-NEA-based 2D chiral perovskite layer on top of a 3D perovskite light absorber through which the photogenerated charge carriers were spin-polarized by chiral NEA molecules capable of strong π–π interactions and enhanced spin orbital coupling. Spin relaxation during charge transport to the catalyst/electrolyte interface was further diminished not only by adopting a mixed HTL having a relatively long spin diffusion length but also by employing a chiral spin selective catalyst on top of the HTL. The 2D perovskite SP deposited on the 3D perovskite in the photoanode decorated with a chiral spin selective catalyst (*L*-NF) augmented the spin alignment to promote the generation of ³O₂ while lowering the overpotential and enhancing the PEC water-splitting efficiency.

Furthermore, we elucidated the mechanism of the CISS phenomenon synergistically induced by the combined use of the SP, spin-transportable HTL, and the SCC on charge transport and recombination in the oxygen evolution reaction. Our dual spin-controlled *L*-NF/*S*-NEA 2D/3D photoanode demonstrated a ABPE of 13.17% with a $J_{ph}$ of 23 mA cm⁻² at 1.23 $V_{RHE}$ and a $V_{onset}$ of 0.2 $V_{RHE}$. Interestingly, after connecting the decoupled catalyst-structured perovskite photocathode with an *L*-NF/*S*-NEA 2D/3D photoanode in series, the co-planar PEC–PEC water-splitting device exhibited an STH efficiency of 12.55%, the figure reported for a perovskite-based overall PEC water-splitting device. Our unassisted water-splitting device also demonstrated impressive stability by maintaining 100% of its initial current density after 24 h while producing highly pure hydrogen and oxygen.

## Methods

### Material preparation

Tin(II) chloride dihydrate (SnCl₂·2H₂O, ACS reagent, 98%), urea (ACS reagent, 99.0 – 100.5%), thioglycolic acid (TGA, 99%), methylammonium iodide (MAI), cesium iodide (CsI, 99.999%, trace metal basis), lead bromide (PbBr₂, 99.999%), (R)-(+)-α-methylbenzylamine (*R*-MBA, 98%), (S)-(−)-α-methylbenzylamine (*S*-MBA, 98%), (R)-(+)-sec-butylamine (*R*-BA, 99%), (S)-(+)-sec-butylamine (*S*-BA, 99%), hydroiodic acid (HI, 57 wt% in H₂O, stabilized with 1.5% hypophosphorous acid, 99.95%), hydrobromic acid (HBr, 48 wt% in H₂O, >99.99%), 4-*tert*-butylpyridine (TBP, 96%), the lithium salt of bis(trifluoromethane)sulfonimide (Li-TFSI, 99.95%), 2,2′,7,7′-tetrakis[N,N-di(4-methoxyphenyl)amino]-9,9′-spirobifluorene (spiro-OMeTAD), and Ni foil were purchased from Sigma-Aldrich (USA). Hydrochloric acid (HCl, extra pure grade), isopropyl alcohol (IPA, 99.5%), ethyl acetate (EA, 99.5%), and diethyl ether (99.5%) were purchased from Duksan General Science (South Korea). Poly(3-hexylthiophene-2,5-diyl) (P3HT, 97.3%) was purchased from Ossila (UK). (R)-(+)-1-(2-naphthyl) ethylamine (*R*-NEA,

98%), (S)-(−)-1-(2-naphthyl) ethylamine (*S*-NEA, 98%), lead (II) iodide (PbI$_2$, metal basis, 99.99%) were purchased from TCI chemicals (Japan). Formamidinium iodide (FAI, >99.9%), methylammonium bromide (MABr, >99.99%), and methylammonium chloride (MACl, >99.9%) were purchased from Greatcell Solar Materials (Australia). For the NiFeOOH catalysts, Ni foam ( >99.99%, porosity <50%, 30 × 10 × 1.5 mm) used as a porous substrate was purchased from MTI Korea. *L*-(+)-tartaric acid (*L*-TA, >99.5%), *D*-(-)-tartaric acid (*D*-TA, 99%), and meso-tartaric acid monohydrate (*meso*-TA, >97%) were purchased from Sigma-Aldrich (USA). Nickel (II) nitrate hexahydrate (99.999%, Sigma-Aldrich, USA), iron (III) nitrate nonahydrate ( >99.95%, Sigma-Aldrich, USA), and sodium carbonate (Na$_2$CO$_3$, >99.5%, Sigma-Aldrich, USA) were utilized to synthesize the enantioselective NiFeOOH catalysts. Potassium phosphate dibasic (K$_2$HPO$_4$, >99.5%, Sigma-Aldrich, USA), and potassium phosphate monobasic (KH$_2$PO$_4$, >99.0%, Sigma-Aldrich, USA) were used to create the K-Pi electrolyte.

## Preparation of the 2D chiral OIHPs

To prepare the MBA derivatives, a mixture consisting of 5 mL of MBA, 5.5 mL of hydroiodic acid, and 10 mL IPA was vigorously stirred for 60 min, after which the solvent was fully evaporated off at 110 °C under vacuum to obtain an iodized solid chiral organic precipitate. The product was washed with diethyl ether several times and dried in a vacuum oven at 60 °C for 12 h. To prepare the NEA derivatives, 9.6 mmol of NEA (1643.9 mg) was dissolved in 4 ml of ethanol followed by stirring. After 5 min, HBr (10.4 mmol, 1.176 mL) was added and the solution was stirred for 12 h at room temperature. Subsequently, the solvent was completely evaporated off under vacuum in an oven at 80 °C for 2 days. The resulting white precipitate was purified using diethyl ether and then dried under vacuum at 80 °C for an additional 2 days. To prepare the BA derivatives, a mixture containing 0.5 mg of BA (6.83 mmol), 1.61 mg of hydroiodic acid (7.17 mmol), and 10 mL of IPA was vigorously stirred for 60 min. Afterward, the solvent was evaporated off at 110 °C under vacuum, resulting in the formation of an iodized chiral organic precipitate. The white precipitate was treated with diethyl ether multiple times before drying in a vacuum oven at 60 °C for 12 h.

Before the fabrication of the 2D chiral OIHPs on FTO substrates, the latter were sequentially sonicated in deionized (DI) water, acetone, and ethanol for 15 min each. To achieve the designated chemical formulas of MBA$_2$PbI$_4$, NEA$_2$PbBr$_4$, and BA$_2$PbI$_4$, the respective chiral cation salt was dissolved in DMF with either PbI$_2$ or PbBr$_2$ while ensuring a precursor concentration of 20 wt%. Meanwhile, 10 mol% of MABr was added to the NEA solution to address roughness issues. The resulting precursor was then filtered and spin-coated onto an FTO substrate at 2000 rpm for 30 s. After spin-coating, the as-prepared substrate was sequentially annealed at 65 °C for 15 min, at 120 °C for 30 min, and at 100 °C for 15 min. For CISS measurements, 5 wt% precursor solutions were prepared as thin films with a thickness of ~10 nm.

## Fabrication of the OIHP photoanode and photocathode

The same SnO$_2$/FTO substrate was used in both photoelectrodes. Tin (II) chloride dihydrate (275 mg) and urea (1.25 g) were dissolved in 100 mL of DI water under continuous stirring. In an ambient atmosphere, 1.25 mL of HCl was introduced to the solution, followed by the addition of 25 µL of TGA. FTO substrates (7 × 4 cm) were cleaned in a sonicator containing DI, acetone, or ethanol for 15 min each. The FTO substrate was then immersed in the aforementioned solution and bathed at 90 °C for 3 h. Afterward, the substrate was sonicated in deionized water and IPA for 10 min each. The SnO$_2$ substrate was obtained via annealing at 170 °C for 1 h.

For the 3D OIHP photoanode, the 3D perovskite precursor solution of (FAPbI$_3$)$_{0.95}$(MAPbBr$_3$)$_{0.05}$ was prepared by dissolving FAI (240.758 mg), PbI$_2$ (705.3453 mg), MABr (8.25 mg), PbBr$_2$ (27.04 mg), MACl (33.76 mg), and CsI (18.1867 mg) in a mixture of DMF (0.96 mL)

and DMSO (0.24 mL) under continuous stirring. The SnO$_2$ substrate was exposed to UV for 15 min, after which 60 µL of the perovskite precursor solution was deposited thereon. The substrate was then spin-coated at 1200 rpm for 12 s and 5800 rpm for 20 s. An anti-solvent, ethyl acetate (0.8 mL), was dripped onto the substrate 10 s before the end of the spin-coating process. The 3D perovskite film was prepared by annealing at 100 °C for 1 h.

To prepare the 2D chiral perovskite layers as following the previous reports[56,57], either MBAI (2.5 mg), NEABr (2.5 mg), or BAI (2.5 mg) chiral cation salt was dissolved in 1 mL of IPA under continuous stirring. Afterward, either 150 µL, 40 µL, or 150 µL, respectively, was spin-coated onto the 3D perovskite surface at 6000 rpm for 30 s. For crystallization of the 2D chiral perovskite and to evaporate off any residual IPA solvent, the sample was thermally annealed at 100 °C for 1 min, 120 °C for 3 min, or 100 °C for 1 min, respectively. When dripping the IPA-based solution on the 3D perovskite, 2D PbX$_2$ (PbI$_2$, PbBr$_2$) forms an intermediate phase on the surface of the 3D perovskite, combined with IPA. Subsequently, amine ions (NH$^{3+}$) involved in MBA, NEA, and BA, which is electrostatically interact to the PbX$_2$, replace the IPA between the IPA-PbX$_2$ intermediate, leading to the initiation of 2D layer. The intercalated bulk cation experiences van der Waals interactions between organic cations or π-π interactions between benzene rings, resulting in the formation of a thermodynamically stable layered structure. Consequently, 2D chiral perovskites with molecular formulas of MBA$_2$PbI$_4$, 2-NEA$_2$PbBr$_4$, and BA$_2$PbI$_4$ were synthesized on 3D perovskite layers.

For the XRD and 2D-XRD analyses, 300 µl, 120 µl, or 300 µl salt solution, respectively, was spin-coated onto a 3D perovskite surface. The HTL layer was spin-coated onto the substrate at 3000 rpm for 60 s using a mixture of spiro-OMeTAD and P3HT solutions with a volumetric ratio of 100:0, 92:8, 88:12, or 84:16. The spiro-OMeTAD solution was formulated with 72 mg of spiro-OMeTAD, 17.5 µL of Li-TFSI (520 mg of Li-TFSI in 1 mL of acetonitrile), and 28.8 µL of TBP dissolved in 1 mL of chlorobenzene. The P3HT solution contained 15 mg of P3HT, 20.4 µL of Li-TFSI (28.3 mg of Li-TFSI in 1 mL of acetonitrile), and 28.8 µL of TBP, all of which are dissolved in 1 mL of chlorobenzene.

Following OIHP preparation, conductive carbon powder Ketjen was spread thereon, and then Ni foil was placed on top. Subsequently, to prevent water infiltration, the OER device was thoroughly sealed using an epoxy resin (Henkel Loctite, E-120HP, Germany).

The NiFeOOH catalysts were electrodeposited onto Ni foam. Briefly, 4 mmol of *L*-TA, *D*-TA, or *meso*-TA and 0.1 mmol of Fe(NO$_3$)$_3$·9H$_2$O and Ni(NO$_3$)$_2$·6H$_2$O were dissolved in 100 ml of 10 mmol aqueous Na$_2$CO$_3$ solution. The resulting solution was purged with Ar gas before and during deposition. For the deposition of NiFeOOH catalysts, CV was conducted with a bias from −0.15 V to 0.25 V against the reference electrode at a fixed scan rate of 60 mV s$^{-1}$. The sample was washed with DI water and dried with air. Subsequently, the NiFeOOH catalyst/Ni foil was affixed onto the Ni foil/OIHP photoanode with Ag paste.

For the OIHP photocathode, the same 3D perovskite film of (FAPbI$_3$)$_{0.95}$(MAPbBr$_3$)$_{0.05}$ was used for the absorber layer following the same fabrication method. The HTL layer was prepared by spin-coating a spiro-OMeTAD solution formulated with 72 mg of spiro-OMeTAD, 8.8 µL of Li-TFSI solution (520 mg of Li-TFSI in 1 mL of acetonitrile), and 4.4 µL of TBP, dissolved in 1 mL of acetonitrile onto the OIHP films at 3000 rpm for 60 s. Subsequently, an Au top electrode with a thickness of 70 nm was deposited via thermal evaporation, after which Cu wire was connected to the Au electrode region using Ag paste. The structure was then solidified at room temperature for 12 h, ensuring a firm and stable attachment. To prevent water infiltration, the OIHP photocathode was thoroughly sealed using epoxy resin. The Pt catalyst was placed on the decoupled region of the FTO substrate by employing a 108 Auto Sputter coater (Ted Pella, Redding, CA, USA), which was operated for 120 s under 0.1 mbar of Ar atmosphere and a current of 10 mA.

## Characterization of the materials

CD and absorbance spectra were measured by using a J-815 spectrometer (JASCO Corporation, Tokyo, Japan). The CD data were measured 10 times to eliminate noise peaks, after which the average was calculated. XRD spectroscopy was conducted with a Rigaku Miniflex 600 instrument (Tokyo, Japan) using Cu K$_\alpha$ radiation with scanning from 3° to 40°. Cross-sectional images of the samples were obtained by using field-emission SEM (JSM-7001F, JEOL, Japan). EDX mapping was conducted using the same equipment at an acceleration voltage of 15 kV equipped with an Oxford EDX instrument (Ultim Max, UK). To study the electrical conduction according to the handedness of the 2D chiral perovskites, mCP-AFM measurements were conducted by using an atomic force microscope (SPA400, Seiko Instruments, Inc., Chiba, Japan). A Co-coated magnetic cantilever (Multi75M-G, Budgent sensors, Sofia, Bulgaria) was employed after pre-magnetization with a 5000 gauss permanent magnet for 20 min before use. Steady-state PL analysis was conducted using a Raman microscope (Alpha 300 Apyron, WITec, Germany) at an excitation wavelength of 581 nm. The surface morphology of the perovskite films was examined via AFM (NX-10, Park Systems, South Korea). 2D-XRD patterns were collected using a D8-Discover instrument (Bruker, USA) equipped with a GADDS four-circle detector. The 2D XRD spectra with GADDS were measured as a function of the γ-angle, which indicates the tilted angle with respect to the substrate, from 70° to 110°. HR-TEM and elemental mapping were conducted on a JEM-ARM200F (JEOL, Tokyo, Japan) with an integrated EDX system at an accelerating voltage of 200 kV. Before HR-TEM analysis, the sample was prepared by subjecting it to a focused ion beam (Crossbeam 350, ZEISS, Germany) operating at an accelerating voltage of 2 kV. The energy level of each semiconductor was analyzed by ultraviolet photoelectron spectroscopy (UPS, Axis-NOVA, and Ultra DLD, UK) under He I radiation (21.21 eV) to investigate the energy band position of sample. The secondary-electron cutoff region (E$_{cutoff}$) and valence-band edge (E$_{edge}$) were calculated according to Supplementary Note 2 (Supporting Information). TRPL spectra were recorded with an excitation beam wavelength of 340 nm (FluoroMaxPlus, Horiba, Kyoto, Japan).

## PEC and voltammetric analysis

The PEC and CV measurements were performed using a potentiostat (SI 1287, Solartron, UK) in 0.5 M K-Pi buffer (pH 6.5) with a three-electrode configuration (Pt wire as the counter electrode and an Ag/AgCl/KCl (sat. M) as the reference electrode). The experimental set-up was shown in Supplementary Fig. S43. A commercial AM 1.5 G solar simulator and an Si reference cell (Newport Corporation, USA) were utilized for the simulated sunlight and 1 sun calibration, respectively. For LSV measurements, the applied potentials were based on the RHE scale to allow comparisons with other reports. The following equation was used to convert the potential:

$$E_{RHE} = E_{Ag/AgCl} + 0.059\,pH + 0.197. \tag{5}$$

EIS was conducted using the same instrument as for the PEC measurements, along with a frequency analyzer (1252 A, Solartron, Leicester, UK). The experimental set-up was the same as PEC and CV measurements (Supplementary Fig. S43). The IPCE, IMPS, and IMPV measurements were conducted using an electrochemical workstation (Zennium, Zahner, Germany) and a potentiostat (PP211, Zahner, Germany) with a monochromatic light source (TLS03, Zahner). Electrochemical analysis was performed at 1.0 V$_{RHE}$ in 0.5 M Na$_2$SO$_4$ (pH 6.5) under AM 1.5 G (100 mW cm$^{-2}$) illumination for measuring H$_2$O$_2$ and O$_2$. The measurements of O$_2$ and H$_2$O$_2$ were taken every 30 min during the electrochemical reaction of 3 h. An oxygen sensor (NeoFox Phase

Fluorometer, Ocean Insight, USA) was utilized for the O$_2$ detection. All of the device connections were completely sealed with rubber bulkheads to prevent gas leakage from the quartz reactor. The cell was purged with Ar prior to the measurement to ensure an O$_2$ content lower than 0.5%. Hydrogen peroxide formation was detected using colorimetric titration method. To detect hydrogen peroxide, 1 mL of o-tolidine, a redox indicator, and 3 mL of the electrolyte aliquot extracted as a function of reaction duration were mixed, followed by performing UV-vis absorption spectroscopy. To accurately determine the H$_2$O$_2$ amount, a calibration curve was constructed using a 30% w/w commercial H$_2$O$_2$ solution (Sigma-Aldrich, USA).

## Data availability

All data generated or analyzed during this study along with its Supplementary Information are included in this published article. Source data are provided as a Source Data file. Source data are provided with this paper.

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

## Acknowledgements

This work was supported by National Research Foundation (NRF) of Korea grants (Nos. 2021R1A3B1068920(J.M.), and 2021M3H4A1A03049662(J.M.)) funded by the Ministry of Science and ICT. This work was supported (in part) by the Yonsei Signature Research Cluster Program for 2022 (2022-22-0001) (J.M.) and Yonsei University Research Fund (Post Doc. Researcher Supporting Program) of 2023 (project no.: 2023-12-0016) (J.M.).

## Author contributions

H.L. drafted the manuscript. Also H.L. conceived the idea, conducted experiments, and analyzed the data with C.U.L. J.Y. conducted the experiments and analyzed the data. C.J. manufactured the OIHP photocathodes. W.J. and J.S. assisted in the fabrication of chiral 2D/3D OIHP

photoanode. Y.S.P., S.M., S.L. and J.H.K. supported the experiments. J.M. supervised the project, directed the research, and contributed to writing the manuscript.

## Competing interests

The authors declare no competing interests.
