## [Peer Review File · Nature Communications]

REVIEWER COMMENTS

Reviewer #1 (Remarks to the Author):

In this manuscript, the authors claimed to fabricate a dual spin-controlled OIHP photoelectrode with enhance PEC performance by inserting a 2D chiral OIHP as spin-filter layer and using a chiral spin-selective L-NiFeOOH catalyst. The idea of using chiral spin-selective material and catalyst in PEC water splitting is interesting. However, it lacks direct evidence to prove that spin indeed plays a role in boosting the performance, as inserting normal 2D layer and using normal OER catalyst can also enhance the performance. I cannot recommend it for publication at the current stage. It can be reconsidered if the authors could address the following comments.

1. The most important point in this manuscript is that the authors believe that the spin effect improves the PEC performance, but there is no direct evidence, and it lacks in-depth explanation of the spin effect, such as how spin affects the PEC performance or other properties. The authors should provide some direct evidences.
2. Is it possible that the enhanced performance comes from the normal passivation effect of the 2D perovskite and normal OER catalyst effect?
3. For the 2D chiral perovskites and chiral spin-selective catalysts, do they have the same spin orientation?
4. How to judge the catalyst is a high-quality chiral NF catalyst by XRD pattern and Raman spectrum?
5. How to understand the spin filtering of charge carriers when passing through the S-NEA 2D SFL? Does it filter out photogenerated holes with different spin directions or electrons?
6. Is the greater the degree of spin polarization the better? In addition, what factors does the degree of spin polarization depend on?
7. How can IMVS reveal the electrochemical reaction mechanism originating from spin selectivity?
8. For the equivalent circuit in Supplementary Fig. S25, why are CPE1 and CPE2 in parallel mode and CPE3 in series mode? Based on what criteria to choose series mode or parallel mode?
9. In page 6, line 120, the authors state that absorption can be proved to belong to a 2D chiral structure. Why is absorption associated with a chiral structure?
10. For the perovskite used in this work, when its solar cell form is directly connected to electrocatalysts for water splitting (i.e., PV-EC), are there any advantages to use chiral spin-selective catalysts to integrate with perovskite PVs?
11. In Figure 2b, there is a mistake in the legend for the orange curve and green curve.

Reviewer #2 (Remarks to the Author):

This is a bold work of significant and novel results on halide perovskite photoanodes with enhanced photocurrents, assigned to spin filtering. However, there are some key characterizations missing to understand how much of the enhancement results from spin filtering and how much from just interfacial energetics favouring charge separation. Please address the following comments:

1. Please provide energy diagrams of studied photoanode and photocathode with all its layers involved

2. Related to the previous comment, it is known that 2D perovskite layers can be used on the surface of a 3D perovskite layer to promote charge separation and improve photocurrents and photovoltages. Such enhanced charge separation is obvious in the analysis of the energy diagrams, considering VB, CB, and Ef/workfunction alignment between the device layers including the transport layers (i.e. interfacial energetics). It is not clear in the manuscript that the authors have considered the resulting energy diagram. How much of the enhanced photocurrents are related to spin filtering across layers and how much to simple favoured interfacial energetics?
3. There is not much characterization of the device as a PV before being used as a photoanode, before and after the addition of those chiral layers. Please provide clear PV characterization. This may help to address previous comments
4. What is the maximum stability of the device, and what causes its death? Please explain it in the manuscript
5. Please avoid acronyms in abstract and conclusions
6. H₂O₂ is claimed to be a byproduct of the photoanode. Any proof of it?
7. When hydrogen and oxygen gases are claimed to be generated at rates of 11.2 and 5.6 $\mu\text{mol h}^{-1}$, respectively, please provide area.
8. Is the Ni foil in direct contact with the Ni foam, or do holes travel through the silver paste? Can this result in any voltage losses? How mechanically stable is that silver paste?
9. In figure 2b, the legend shows two different S88:P12 J-V curves. An error?
10. It is not explained how Cu wire was connected to the Au electrode region.
11. How much Pt was deposited?

Reviewer #3 (Remarks to the Author):

The manuscript describes the use of 2D chiral perovskite combined with 3D perovskite and chiral catalysts for overall water splitting. While it addresses an interesting application of 2D chiral perovskites, and is a comprehensive piece of work, there are several confusing points, as detailed below, that should be addressed before the consideration for publication.

“OIHP” and “ABPE” in the abstract are not defined.

In discussing “In addition, the spin polarization of 71 chiral catalysts remains relatively low at around 50%”, it would be useful to use an equation to define the spin polarization explicitly.

The sentence “The introduction of NEA organic cations can improve the angular momentum of chiral 2D OIHPs, thereby leading to enhanced circular polarization” is ambiguous. What angular momentum is being improved (electrons in the conduction band, electrons in the valence band, etc)? The phrase “angular momentum” has been used in other paragraphs as well, so it is important to clarify what it means. Is “enhanced circular polarization” referring to enhanced circular dichroism, or something else? For a strictly racemic mixture compound, the spin polarization degree should be zero. Is there any rationale behind the observed small degree of spin polarization (a few percent)?

While P3HT was used in partial replacement of the Spiro-MeOTAD to preserve spin polarization of the electrons, how does the Kapchen film impact spin transport? Was mCP-AFM experiments done to confirm the effect of Kapchen film, as well as the outer nickel layer?

Figure S5 shows the X-ray diffraction pattern in the reciprocal space. All the dots on the same ring should

correspond to the same 2θ . The x label and y label should be q_{xyz} (momentum transfer), rather than 2θ . For the 2D/3D perovskite structure fabrication, the ink only includes chiral organic ammonium salts, so what is the formation mechanism of 2D perovskite? Ion exchange due to thermal diffusion? When 2D perovskites are formed, there should be a byproduct of achiral ammonium salts. Why not use the 2D perovskite solution as the precursor to directly form a 2D perovskite layer?

For the discussion of the X-ray diffraction (the end of page 7), The observation of (002) and (004) plane can't suggest that the layered inorganic framework is perpendicular to the substrate. Instead, the appearance of (002) and (004) suggests that the layered inorganic framework is parallel to the substrate. For the SAED (page 9, line 186), how was the crystallinity confirmed via FFT imaging of the white region? In terms of the FWHM?

For the PL, the author claims that the PL revealed the suppressed recombination of photogenerated charges in the devices without the SFL. Is this accurate? The radiative recombination rate is an intrinsic property of the material and is hard to change, while the rate of non-radiative recombination will vary with the density of the defect states. The weakening of PL intensity means the enhancement of non-radiative recombination. The author can quantitatively see the recombination rate by testing the fluorescence lifetime.

'photons are preferentially directed toward the 3D/SLG interface in all three devices (i.e., the 3D OIHP, S-NEA 2D/3D OIHP, and rac-NEA 2D/3D OIHP devices) without traveling through the 2D OIHP layer.' is not clear to me. How can it be connected to the PL intensity?

Reviewer #4 (Remarks to the Author):

Response Letter

Journal: Nature communications

Previous Manuscript ID:

Title: A Dual Spin-Controlled Chiral 2D/3D Perovskite Artificial Leaf for Efficient Overall Photoelectrochemical Water Splitting

Author(s): *Hyungsoo Lee, Chan Uk Lee, Juwon Yun, Chang-Seop Jeong, Wooyong Jeong, Jaehyun Son, Young Sun Park, Subin Moon, Soobin Lee, Jun Hwan Kim, and Jooho Moon*

<Reviewer 1>

In this manuscript, the authors claimed to fabricate a dual spin-controlled OIHP photoelectrode with enhance PEC performance by inserting a 2D chiral OIHP as spin polarizer and using a chiral spin-selective L-NiFeOOH catalyst. The idea of using chiral spin-selective material and catalyst in PEC water splitting is interesting. However, it lacks direct evidence to prove that spin indeed plays a role in boosting the performance, as inserting normal 2D layer and using normal OER catalyst can also enhance the performance. I cannot recommend it for publication at the current stage. It can be reconsidered if the authors could address the following comments.

Remark:

We would like to thank the reviewer for helpful comments and evaluating our study as a detailed study. The quality of our manuscript has been improved according to the reviewer's comments. The following are the answers to the reviewer's particular questions:

Comment 1:

The most important point in this manuscript is that the authors believe that the spin effect improves the PEC performance, but there is no direct evidence, and it lacks in-depth explanation of the spin effect, such as how spin affects the PEC performance or other properties. The authors should provide some direct evidences.

Author's Response:

We appreciate the reviewer for important comment on the origin of enhanced PEC performance related to the spin effect. Regarding the investigations of synthesized 2D chiral OIHP and chiral spin-selective catalysts, circular dichroism analysis in Figure 1a and Supplementary Figure 17 reveal their chiroptic properties. To confirm whether the chiral materials can control the spin state of charge carriers, we attempted to measure spin-dependent current flow using magnetic conductive probe atomic force microscopy (mCP-AFM), as depicted in Figure 1d, e, and Figure 3c. Our results clearly demonstrate the application of the CISS effect depending on the presence of chirality.

Conventional OER devices exhibit OH radicals with random spin states, leading to the simultaneous formation of hydrogen peroxide (H₂O₂) and singlet oxygen as their formation requires antiparallel spin states. However, upon the transport of the spin-polarized charges to the photoanode surface in spin-controlled devices, OH radicals, responsible for O₂ formation, accept the spin-controlled carriers, being aligned with a specific spin state at the Volmer step. As the OER consist of sub-elementary reaction steps where each sub-reaction step involves the four-electron transfer and associated intermediate radical removal, the OER can be simply expressed by the following sequence of sub-elementary reaction step that occurs at the catalytic surface;

where MO^\uparrow , MOOH^\uparrow , MOH^\uparrow and 2O^\uparrow represent three different intermediate radicals on the active sites of NiFeOOH catalyst with one specific spin state and triplet state of oxygen molecules ($^3\text{O}_2$), respectively. The high spin state in active site NiFeOOH catalyst can be preserved during the OER process by constant injection of spin-aligned charge carrier. As a result, the intermediate radicals adsorbed on the active site also have one specific spin direction, preventing the formation of H_2O_2 and facilitating the evolution of triplet state oxygen. Considering the energy difference between a singlet $^1\text{O}_2$ and triplet $^3\text{O}_2$ (the energy of a singlet $^1\text{O}_2$ is approximately 100 kJ mol^{-1} higher than that of triplet $^3\text{O}_2$), spin-dependent OER process at the surface of active site with one specific spin state (high spin state) can significantly improve the photocurrent and fill factor compared to a photoanode without a spin-polarization layer. We have presented the evidence of this phenomenon in Figure 3e, showing the Faradaic efficiency for O_2 formation (FE_{O_2}). Nevertheless, as suggested by the reviewer, additional evidence is required to demonstrate the impact of the spin effect on PEC performance.

To confirm the suppression of by-product (H_2O_2) as an evidence for spin-controlled OER, we conducted colorimetric experiment to quantify the generated amount of H_2O_2 by observing the time-dependent absorbance spectra of the electrolyte aliquot collected as a function of reaction duration, as shown in Fig. R1. The absorbance arises from the presence of a reactant between o-tolidine and H_2O_2 so that the absorption varies depending upon the amount of H_2O_2 in the electrolyte. Colorimetric experiment was conducted under potentiostatic condition in $0.5 \text{ M Na}_2\text{SO}_4$ electrolyte, enabling the detection of generated H_2O_2 at $1.0 \text{ V}_{\text{RHE}}$. Furthermore, we utilized an oxygen sensor to detect O_2 simultaneously, allowing us to calculate the FE_{O_2} as a function of reaction duration, as illustrated in Fig. R2. For the 3D OIHP

photoanode utilizing *meso*-NF catalyst, the FE_{O_2} was calculated to be 73%, while the FE for H_2O_2 ($FE_{H_2O_2}$) was 16%, as shown in Fig. R2a. Even with the addition of *racemic*-type achiral 2D on top of 3D OIHP (Fig. R2b), the FE_{O_2} and $FE_{H_2O_2}$ remained similar at 72% and 14%, respectively. However, upon the insertion of *S*-NEA-based chiral 2D OIHP (Fig. R2c), the FE_{O_2} increased to 78%, while the $FE_{H_2O_2}$ decreased to 8%, indicating that the CISS effect works on OER device. Moreover, replacing *meso*-NF catalyst with chiral *L*-NF catalyst further enhanced the FE_{O_2} and reduced the $FE_{H_2O_2}$, achieving FE_{O_2} and $FE_{H_2O_2}$ of 90% and 3%, respectively (Fig. R2d). This clearly demonstrates that the dual spin control strategy manipulates the spin state of charge carriers for an extended period, leading to the suppression of by-product H_2O_2 and the formation of lower-energy triplet oxygen, resulting in a significant improvement in FE_{O_2} . Thanks to the critical comments from the reviewer, we are able to perform appropriate revisions to enhance the quality of our paper.

Fig. R1. UV-vis absorption spectra from o-toluidine titration of electrolytes (0.5 M Na₂SO₄) as a function of reaction duration for (a) *meso*-NF/3D OIHP, (b) *meso*-NF/*rac*-NEA 2D/3D OIHP, (c) *meso*-NF/*S*-NEA 2D/3D OIHP, and (d) *L*-NF/*S*-NEA 2D/3D OIHP.

Fig. R2. Calculated O_2 and H_2O_2 Faradaic efficiency stacked bar chart for (a) *meso*-NF/3D OIHP, (b) *meso*-NF/*rac*-NEA 2D/3D OIHP, (c) *meso*-NF/*S*-NEA 2D/3D OIHP, and (d) *L*-TA/*S*-NEA 2D/3D OIHP.

Revision made (colored in blue):

(Supporting Information; Supplementary Fig. S34 are added)

Supplementary Fig. S34. UV-vis absorption spectra from o-toluidine titration of electrolytes (0.5 M Na₂SO₄) as a function of reaction duration for (a) *meso*-NF/3D OIHP, (b) *meso*-NF/*rac*-NEA 2D/3D OIHP, (c) *meso*-NF/S-NEA 2D/3D OIHP, and (d) *L*-TA/S-NEA 2D/3D OIHP.

(Supporting Information; Supplementary Fig. S35 are added)

Supplementary Fig. S35. Calculated O₂ and H₂O₂ Faradaic efficiency stacked bar chart for (a) *meso*-NF/3D OIHP, (b) *meso*-NF/*rac*-NEA 2D/3D OIHP, (c) *meso*-NF/*S*-NEA 2D/3D OIHP, and (d) *L*-TA/*S*-NEA 2D/3D OIHP.

(page 21, line 18)

This observation clearly suggests the success of implementing the CISS effect in our dual spin-controlled photoanode so that the photogenerated charges appropriately generate the desired product (O₂) while effectively suppressing the hydrogen peroxide side reaction. To confirm the suppression of by-product (H₂O₂) as an evidence for spin-controlled OER, we conducted colorimetric experiment to quantify the generated amount of H₂O₂ by observing the time-dependent absorbance spectra of the electrolyte aliquot collected as a function of reaction duration, as shown in Supplementary Fig. S34. The absorbance arises from the presence of a

reactant between o-tolidine and H₂O₂ so that the absorption varies depending upon the amount of H₂O₂ in the electrolyte. Colorimetric experiment was conducted under potentiostatic condition in 0.5 M Na₂SO₄ electrolyte, enabling the detection of generated H₂O₂ at 1.0 V_{RHE}. Furthermore, we utilized an oxygen sensor to detect O₂ simultaneously, allowing us to calculate the FE_{O₂} as a function of reaction duration, as illustrated in Supplementary Fig. S35. For the 3D OIHP photoanode utilizing *meso*-NF catalysts, the FE_{O₂} was calculated to be 73%, while the FE for H₂O₂ (FE_{H₂O₂}) was 16%, as shown in Supplementary Fig. S35a. Even with the addition of *racemic*-type achiral 2D on top of 3D OIHP (Supplementary Fig. S35b), the FE_{O₂} and FE_{H₂O₂} remained similar at 72% and 14%, respectively. However, upon the insertion of *S*-NEA-based chiral 2D OIHP (Supplementary Fig. S35c), the FE_{O₂} increased to 78%, while the FE_{H₂O₂} decreased to 8%, indicating that the CISS effect works on OER device. Moreover, replacing *meso*-NF with chiral *L*-NF further enhanced the FE_{O₂} and reduced the FE_{H₂O₂}, achieving FE_{O₂} and FE_{H₂O₂} of 90% and 3%, respectively (Supplementary Fig. S35d). This clearly demonstrates that the dual spin control strategy manipulates the spin state of charge carriers for an extended period, leading to the suppression of by-product H₂O₂ and the formation of lower-energy triplet oxygen, resulting in a significant improvement in FE_{O₂}.

(Methods, page 41)

EIS was conducted using the same instrument as for the PEC measurements, along with a frequency analyzer (1260, Solartron, Leicester, UK). The IPCE, IMPC, and IMPV measurements were conducted using an electrochemical workstation (Zennium, Zahner, Germany) and a potentiostat (PP211, Zahner, Germany) with a monochromatic light source (TLS03, Zahner). Electrochemical analysis was performed at 1.0 V_{RHE} in 0.5 M Na₂SO₄ (pH 6.5) under AM 1.5 G (100 mW cm⁻²) illumination for measuring H₂O₂ and O₂. The

measurements of O₂ and H₂O₂ were taken every 30 min during the electrochemical reaction of 3 h. An oxygen sensor (NeoFox Phase Fluorometer, Ocean Insight, USA) was utilized for the O₂ detection. All of the device connections were completely sealed with rubber bulkheads to prevent gas leakage from the quartz reactor. The cell was purged with Ar prior to the measurement to ensure an O₂ content lower than 0.5%. Hydrogen peroxide formation was detected using colorimetric titration method. To detect hydrogen peroxide, 1 mL of o-tolidine, a redox indicator, and 3 mL of the electrolyte aliquot extracted as a function of reaction duration were mixed, followed by performing UV-vis absorption spectroscopy. To accurately determine the H₂O₂ amount, a calibration curve was constructed using a 30% w/w commercial H₂O₂ solution (Sigma-Aldrich, USA).

Comment 2:

Is it possible that the enhanced performance comes from the normal passivation effect of the 2D perovskite and normal OER catalyst effect?

Author's Response:

In response to the reviewer's comment, we have conducted additional experiment to discern whether the observed enhancement in PEC performance is due to the passivation effect of the NEA 2D OIHP or the influence of the OER catalyst. As depicted in Fig. R3a, when achiral *rac*-NEA 2D OIHP was inserted into the *meso*-NF/3D OIHP photoanode, a slight improvement in PEC performance was observed. However, this improvement can be attributed to the additional photovoltage formation resulting from the back-field effect facilitated by the heterojunction of 2D/3D OIHPs. By contrast, substantial enhancement in PEC performance was observed upon

replacing *rac*-NEA 2D with chiral *S*-NEA 2D capable of inducing the CISS effect to suppress H₂O₂ production and facilitate triplet oxygen generation, leading to a significant improvement in PEC performance, as evident in Fig. R3.

To assess the effect of the OER catalyst on performance, we compared the catalytic activity of bare Ni foam, achiral *meso*-NF, and chiral *L*-NF catalyst, as depicted in Fig. R4a. While bare Ni foam exhibited very low catalytic activity, *meso*-NF showed higher catalytic efficiency. Notably, *L*-NF demonstrated the highest catalytic activity. Considering that both *meso*-NF and *L*-NF consist of the identical stoichiometry of NiFeOOH catalyst, it suggests that the performance enhancement of *L*-NF is attributed to its chirality. To evaluate its impact on PEC performance, we deposited bare Ni foam, *meso*-NF, and *L*-NF onto the 3D OIHP photoanode and conducted LSV measurements as shown in Fig. R4b. Consistent with the trends observed in Fig. R4a, the *L*-NF/3D photoanode exhibited superior catalytic activity compared to the achiral *meso*-NF/3D and Ni foam/3D photoanodes. This suggests that the introduction of *L*-NF not only improves OER catalytic activity but also induces a reduction in overpotential due to the CISS effect resulting from the use of a chiral spin-selective catalyst, thus enhancing PEC performance. Accordingly, the manuscript is revised to clarify that the observed enhancement in PEC performance is not solely attributed to the passivation effect of the 2D perovskite.

Fig. R3. LSV curves of *L*-NF/*S*-NEA 2D/3D (red), *meso*-NF/*S*-NEA 2D/3D (green), *meso*-NF/*rac*-NEA 2D/3D (navy), and *meso*-NF/3D (orange) photoanodes.

Fig. R4. (a) LSV curves for *meso*-NF and *L*-NF on the Ni foam and bare Ni foam. (b) LSV curves of *L*-NF, *meso*-NF, and bare Ni foam on the 3D OIHP photoanodes under 0.5 M K-Pi electrolyte (pH 7.0).

Revision made (colored in blue):

(Supporting Information; Supplementary Fig. S28 are added)

Supplementary Fig. S28. LSV curves of *meso*-NF/*rac*-NEA 2D/3D photoanode.

(page 19, line 18)

Similarly, the dual spin-controlled *L*-NF/*S*-NEA 2D/3D photoanode exhibited a superior OER performance ($J_{ph} \sim 22.56 \text{ mA cm}^{-2}$ and $V_{onset} \sim 0.20 \text{ V}_{RHE}$) over the *meso*-NF/*S*-NEA 2D/3D photoanode ($J_{ph} \sim 20.67 \text{ mA cm}^{-2}$ and $V_{onset} \sim 0.27 \text{ V}_{RHE}$). Moreover, judging from the performance of the *meso*-NF/*rac*-NEA 2D/3D photoanode ($J_{ph} \sim 14.01 \text{ mA cm}^{-2}$ and $V_{onset} \sim 0.34 \text{ V}_{RHE}$, Supplementary Fig. S28), it can be inferred that the enhancement in PEC performance is not solely attributed to the introduction of 2D OIHP. This can be attributed to the

Comment 3:

For the 2D chiral perovskites and chiral spin-selective catalysts, do they have the same spin orientation?

Author's Response:

As the reviewer mentioned, the spin orientation matching induced by both chiral 2D OIHP and chiral spin-selective catalysts (SCCs) plays a crucial issue. To clarify this aspect, we have evaluated the PEC performance using LSV curves for the various combinations of *R*-/*S*-NEA 2D and *L*-/*D*-NF SCCs, each possessing different chirality handedness. When the chirality configuration of *S*-NEA 2D is changed to *R*-type, the photocurrent and onset potential of the *L*-NF/*R*-NEA 2D/3D photoanode significantly decrease, as indicated by red-dotted line in Fig. R5. This indicates that the spin orientation induced by *L*-NF and *R*-NEA 2D is in opposite directions. Furthermore, as evident in Fig. R6, *L*-NF and *R*-NEA 2D serve as opposite spin polarizers inducing down-spin and up-spin, respectively. Consequently, using *L*-NF together with *S*-NEA 2D to induce the same direction of down-spin carriers, the PEC performance is greatly enhanced. Similarly, the performance also improves when using *D*-NF and *R*-NEA 2D, inducing the same direction of up-spin carriers. However, the spin-controlled photoanode revealed the decreased photocurrent density when stacking *D*-NF and *S*-NEA 2D, which presumably manipulates the opposite directions of spin state. In response to the insightful comment from the reviewer, we have revised the manuscript.

Fig. R5. LSV curves of *L*-NF/*S*-NEA 2D/3D (red-solid), *L*-NF/*R*-NEA 2D/3D (red-dotted), *D*-NF/*R*-NEA 2D/3D (blue-solid), and *D*-NF/*S*-NEA 2D/3D (blue-dotted) photoanodes.

Fig. R6. The mCP-AFM analysis to determine the spin-dependent currents for (a) *S*-NEA 2D, (b) *R*-NEA 2D deposited on 3D OIHP, and (c) *L*-NF, (d) *D*-NF on Ni substrate.

Revision made (colored in blue):

(Supporting Information; Supplementary Fig. S29 are added)

Supplementary Fig. S29. LSV curves of *L*-NF/*S*-NEA 2D/3D (red-solid), *L*-NF/*R*-NEA 2D/3D (red-dotted), *D*-NF/*R*-NEA 2D/3D (blue-solid), and *D*-NF/*S*-NEA 2D/3D (blue-dotted) photoanodes.

(page 19, line 19)

This dual spin alignment synergistically enhances the PEC performance of the photoanode. To clarify this aspect, we have evaluated the PEC performance using LSV curves for the various combinations of *R*-/*S*-NEA 2D and *L*-/*D*-NF SCCs, each possessing different chirality handedness. When the chirality configuration of *S*-NEA 2D is changed to *R*-type, the photocurrent and onset potential of the *L*-NF/*R*-NEA 2D/3D photoanode significantly decrease, as indicated by red-dotted line in Fig. R5. This indicates that the spin orientation induced by *L*-NF and *R*-NEA 2D is in opposite directions. Furthermore, as evident in Supplementary Fig. S29, *L*-NF and *R*-NEA 2D serve as opposite spin polarizers inducing down-spin and up-spin, respectively. Consequently, using *L*-NF together with *S*-NEA 2D to induce the same direction of down-spin carriers, the PEC performance is greatly enhanced. Similarly, the performance also improves when using *D*-NF and *R*-NEA 2D, inducing the same direction of up-spin carriers. However, the spin-controlled photoanode revealed the decreased photocurrent density when stacking *D*-NF and *S*-NEA 2D, which presumably manipulates the opposite directions of spin state. To confirm.....

Comment 4:

How to judge the catalyst is a high-quality chiral NF catalyst by XRD pattern and Raman spectrum?

Author's Response:

In our manuscript, we firstly confirmed the well-dispersion of Ni and Fe by SEM analysis and energy dispersive X-ray (EDX) mapping. XRD analysis revealed the formation of NiFeOOH phase, with prominent peaks corresponding to Ni (111), (200), and (220) as well as the observation of the NiFeOOH (012) plane at 37.2° , indicating the formation of the desired phase. The Raman spectrum exhibited distinct vibration modes, including the $\text{Ni}^{3+}\text{-O-Ni}^{3+}$ modes at 490 and 512 cm^{-2} and the $\text{Fe}^{3+}\text{-O-Fe}^{3+}$ mode at 521 cm^{-2} .

Additionally, circular dichroism (CD) analysis was conducted to assess the chirality retention of the synthesized *L*-NF. While *meso*-NF showed no significant variation in CD signal across wavelengths, both *L*-NF and *D*-NF exhibited noticeable CD signals in opposite directions at 530 nm, corresponding to the light absorption edge. Furthermore, the $P_{-0.75V}$ values obtained from mCP-AFM indicated highly spin-dependent currents for both *L*- and *D*-NF catalysts compared to *meso*-NF. However, as suggested by the reviewer, evaluating whether a high-quality NF catalyst was produced based solely on these observations might be still insufficient. To address this, we performed X-ray photoelectron spectroscopy (XPS) analysis. The results revealed identical XPS spectra for both *L*-NF and *meso*-NF. Peaks corresponding to Ni^{3+} $2p_{1/2}$ and $2p_{3/2}$ were observed at 867.5 and 854.1 eV, respectively, while a peak at 851.6 eV originated from Ni^0 in the Ni foil substrate (Fig. R7a). Similarly, peaks for Fe^{3+} $2p_{1/2}$ and $2p_{3/2}$ were observed at 722.4 and 709.9 eV, respectively (Fig. R7b). Finally, the O 1s peak was deconvoluted into peaks at 530.8 and 529.5 eV (Fig. R7c), representing the metal-O bond (O^{2-}) and metal-hydroxyl bond (-OH), respectively. These results serve as additional evidence confirming the successful formation of NiFeOOH.

Fig. R7. XPS spectra of *L*-NiFeOOH and *meso*-NiFeOOH: deconvolution of (a) Ni 2p, (b) Fe 2p, and (c) O 1s regions.

Revision made (colored in blue):

(Supporting Information; Supplementary Fig. S23 are added)

Supplementary Fig. S23. (a) XRD and (b) Raman spectra of *L*-NiFeOOH on Ni foam. XPS spectra of *L*-NiFeOOH and *meso*-NiFeOOH: deconvolution of (c) Ni 2p, (d) Fe 2p, and (e) O 1s regions.

(page 17, line 17)

Meanwhile, the Raman spectrum in Supplementary Fig. S15b shows the Ni-O vibration mode at 490 and 512 cm^{-2} and the Fe-O vibration mode at 521 cm^{-2} . Furthermore, we performed X-ray photoelectron spectroscopy (XPS) analysis. The results revealed identical XPS spectra for both *L*-NF and *meso*-NF. Peaks corresponding to Ni^{3+} 2p_{1/2} and 2p_{3/2} were observed at 867.5 and 854.1 eV, respectively, while a peak at 851.6 eV originated from Ni^0 in the Ni foil substrate (Supplementary Fig. S23c). Similarly, peaks for Fe^{3+} 2p_{1/2} and 2p_{3/2} were observed at 722.4 and 709.9 eV, respectively (Supplementary Fig. S23d). Finally, the O 1s peak was deconvoluted into peaks at 530.8 and 529.5 eV (Supplementary Fig. S23e), representing the metal-O bond (O^{2-}) and metal-hydroxyl bond (-OH), respectively. These results serve as additional evidence confirming the successful formation of NiFeOOH. These observations signify the successful synthesis of the high-quality chiral NF catalyst.

Comment 5:

How to understand the spin filtering of charge carriers when passing through the S-NEA 2D SFL? Does it filter out photogenerated holes with different spin directions or electrons?

Author's Response:

The chirality-induced spin selectivity (CISS) effect is a spin-polarization process that occurs when charge carriers move through the chiral materials. As the majority carrier passes through a chiral molecule or material, the curvature of the potential energy associated with the chiral system induces a centripetal force acting on the charge carrier, perpendicular to its velocity. The direction of this force depends on the 'handedness' of the chiral material, adhering to Fleming's left-hand rule. Analogous to a Lorentz force, this centripetal force acts on the charge carrier due to an effective magnetic field oriented along the propagation direction of the charge

carrier, stabilizing one spin direction, and destabilizing the other. Consequently, transmission of one specific spin-state is favored, in which preferred spin state is determined by the chiral axis of the medium (handedness of chiral materials) and the linear momentum of charge carrier, so called as spin flipping. Especially, majority hole carriers photogenerated from perovskite light absorber can also experience CISS effect, so that their spin state is controlled as they pass through a chiral material. Therefore, it is not about filtering out photogenerated holes with opposite spin orientations but rather polarizing photogenerated holes with random spin orientations to align in one direction. We acknowledge that the term "spin filtering" may cause confusion so as to revise it to "spin polarizer" for clarity.

Revision made (colored in blue):

(page 3, line 25)

the chiral molecule.^{16,18} Consequently, transmission of one specific spin-state is favored, in which preferred spin state is determined by the chiral axis of the medium (*i.e.*, handedness of chiral materials) and the linear momentum of charge carrier, so called as spin flipping.¹⁷ Owing to the ...

(page 5, line 6)

Subsequently, to induce the CISS effect, it is crucial to apply a chiral material as a **spin polarizer (SP)** with high spin-polarization efficiency on top of the perovskite light absorber layer.

Comment 6:

Is the greater the degree of spin polarization the better? In addition, what factors does the degree of spin polarization depend on?

Author's Response:

As the degree of spin polarization increases, the spin selectivity also enhances, enabling more photogenerated holes to adopt a specific spin state. Factors influencing the degree of spin polarization include the spin-orbital coupling (SOC) behavior resulting from the interaction between electron spins and lattice structures, as well as the chirality of the material inducing a helical field. Multi-layered chiral 2D OIHP materials have been proposed as an effective spin control layer due to the combined effect of the large SOC of the inorganic layers and the chirality of the organic molecules. Moreover, the self-assembled chiral organic molecules between the inorganic layers are capable of generating the repetitive helical fields, known as augmented CISS process. This can effectively manipulate the spin state of transporting charge carriers in a specific direction. Particularly, chiral 2D OIHP exhibits a high degree of spin polarization, surpassing 80%. In this study, we utilized NEA based 2D OIHP to enhance the helical arrangement of NEA cations and amplify SOC behavior, thereby maximizing chirality transfer efficiency from chiral organic cations to inorganic layer. We have provided additional details to elucidate these aspects in the manuscript.

Revision made (colored in blue):

(page 7, line 9)

implying that the former has lower chiroptical activity than the latter. Chiroptical properties are strongly influenced by chiral-induced helical electric field, which is proportional to the radius of the helically arranged chiral cations. In this context, the smaller size of BA in comparison to MBA leads to the formation of a helix with lower asymmetric distortion, resulting in lower chiroptical activity.³² On the other hand,

Comment 7:

How can IMVS reveal the electrochemical reaction mechanism originating from spin selectivity?

Author's Response:

IMVS measurements involve perturbing the photoelectrode under open-circuit conditions with intensity modulated monochromatic light, while observing changes in photovoltage. This allows us to understand the kinetics of the electrical double layer formed at the photoelectrode-electrolyte interface and the electrochemical reactions. In particular, the IMVS measurement in our research is able to reveal charge recombination occurring at the kilohertz range on the photoelectrode surface. We observed a higher time constant of 97.64 μs in the photoanode using *S*-NEA 2D compared to *rac*-NEA 2D (31.45 μs), indicating a longer charge carrier lifetime on the surface of *L*-NF/*S*-NEA 2D/3D devices. We hypothesized that this phenomenon could be attributed to the suppression of back recombination due to spin alignment induced by the CISS effect, allowing us to infer spin state preservation. We have revised the manuscript to clarify that this analysis indirectly confirms the presence of spin selectivity, rather than directly observing it.

Revision made (colored in blue):

(page 24, line 18)

within the photoanode due to the presence of an SP. The IMVS analysis reveals surface recombination kinetics at the SCLJ to elucidate the lifetime of the photogenerated charges under the open-circuit condition. Fig. 4d and e show Nyquist plots

(page 26, line 2)

The time constants for the *L*-NF/*rac*-NEA 2D/3D and *L*-NF/*S*-NEA 2D/3D devices were 31.45 and 97.64 μ s, respectively, speculating that the spin-polarized charge carriers in the latter are less susceptible to the back-scattering-mediated surface recombination. The imaginary voltage

Comment 8:

For the equivalent circuit in Supplementary Fig. S25, why are CPE1 and CPE2 in parallel mode and CPE3 in series mode? Based on what criteria to choose series mode or parallel mode?

Author's Response:

We opted for the parallel mode of CPE1 and CPE2 when photogenerated charges influence both distinct reactions simultaneously, while the series mode of CPE1 and CPE2 was chosen when the charges act independently on two separate reactions. Upon further consideration of the reviewer's feedback, we determined that considering CPE1 and CPE2 as independent processes passing through the 3D OIHP and NEA 2D OIHP layers, respectively, would be more appropriate for our study. Consequently, we changed the original matryoshka model to a Voigt model and updated the manuscript accordingly (Fig. R8). We are deeply grateful for the critical comment.

Fig. R8. The equivalent circuit using the Voigt model to interpret the EIS analysis results.

Revision made (colored in blue):

(Supporting Information; Supplementary Fig. S36 are modified)

- R_1 : R_s (series)
 CPE_1 : CPE_{HF} (high freq.)
 CPE_2 : CPE_{inter} (interface charging)
 CPE_3 : CPE_{edl} (electrical double layer)
- R_2 : R_{HF} (charge transport)
 R_3 : R_{inter} (intermediate channel)
 R_4 : R_{ct} (charge transfer)

Supplementary Fig. S36. The equivalent circuit using the Voight model to interpret the EIS analysis results.

(Supporting Information; Supplementary Table S3 are modified)

Supplementary Table S3. Summary of the fitted impedance data for the *L*-NF/3D, *L*-NF/*S*-NEA 2D/3D, and *L*-NF/*rac*-NEA 2D/3D photoanodes.

L -NF/ S -NEA 2D/3D	R_{series} ($\Omega \cdot \text{cm}^2$)	R_{HF} ($\Omega \cdot \text{cm}^2$)	CPE_{HF} ($\text{F} \cdot \text{s}^{n-1} \cdot \text{cm}^{-2}$)	R_{inter} ($\Omega \cdot \text{cm}^2$)	CPE_{inter} ($\text{F} \cdot \text{s}^{n-1} \cdot \text{cm}^{-2}$)	R_{ct} ($\Omega \cdot \text{cm}^2$)	CPE_{edl} ($\text{F} \cdot \text{s}^{n-1} \cdot \text{cm}^{-2}$)
0.4 V_{RHE}	2.97	30.51	1.11×10^{-8} ($n = 1.09$)	6.00	2.01×10^{-6} ($n = 0.94$)	14.47	5.41×10^{-4} ($n = 0.78$)
0.8 V_{RHE}	2.27	22.24	1.02×10^{-8} ($n = 1.08$)	6.96	2.11×10^{-6} ($n = 0.95$)	6.01	5.28×10^{-4} ($n = 0.81$)
1.2 V_{RHE}	2.17	19.89	2.01×10^{-8} ($n = 1.09$)	9.94	1.96×10^{-6} ($n = 0.94$)	5.01	4.97×10^{-4} ($n = 0.78$)

L -NF/ rac -NEA 2D/3D	R_{series} ($\Omega \cdot \text{cm}^2$)	R_{HF} ($\Omega \cdot \text{cm}^2$)	CPE_{HF} ($\text{F} \cdot \text{s}^{n-1} \cdot \text{cm}^{-2}$)	R_{inter} ($\Omega \cdot \text{cm}^2$)	CPE_{inter} ($\text{F} \cdot \text{s}^{n-1} \cdot \text{cm}^{-2}$)	R_{ct} ($\Omega \cdot \text{cm}^2$)	CPE_{edl} ($\text{F} \cdot \text{s}^{n-1} \cdot \text{cm}^{-2}$)
0.4 V_{RHE}	2.12	82.11	9.92×10^{-9} ($n = 1.02$)	8.01	1.96×10^{-6} ($n = 0.93$)	72.02	5.97×10^{-4} ($n = 0.75$)
0.8 V_{RHE}	2.04	70.28	8.57×10^{-9} ($n = 1.01$)	7.45	1.98×10^{-6} ($n = 0.94$)	37.21	6.27×10^{-4} ($n = 0.71$)

1.2 V _{RHE}	1.89	58.9	8.88×10^{-9} (n = 1.02)	6.10	2.01×10^{-6} (n = 0.94)	24.46	7.79×10^{-4} (n = 0.73)
------	------	-------------------------------------	------	-------------------------------------	-------	-------------------------------------

(page 23, line 25)

The R_{inter} of the *L*-NF/*S*-NEA 2D/3D photoanode steadily increased from 6.00 to 9.94 $\Omega\cdot\text{cm}^2$ when increasing the applied bias, which implies that the *S*-NEA 2D OIHP layer effectively provides an intermediate channel inducing high spin state, resulting in the efficient propagation of spin-polarized charges and the reduction of elastic backscattering-induced recombination due to Pauli exclusion. In contrast, the opposite behavior of R_{inter} is observed for the *L*-NF/*rac*-NEA 2D/3D photoanode, where R_{inter} gradually decreases with the applying bias (from 8.01 to 6.10 $\Omega\cdot\text{cm}^2$).

Comment 9:

In page 6, line 120, the authors state that absorption can be proved to belong to a 2D chiral structure. Why is absorption associated with a chiral structure?

Author's Response:

We appreciate the insightful comment. When MBA-, NEA-, and BA-based 2D chiral perovskites are synthesized, the wavelengths at which each chiral 2D OIHP can absorb light are 500 nm, 390 nm, and 420 nm, respectively. This does not signify the chiral structure but rather aims to specify the wavelength where the CD spectra intersect depending on the handedness of each chiral cation. The manuscript is revised to avoid potential confusion, as per the reviewer's suggestion.

Revision made (colored in blue):

(page 7, line 5)

Circular dichroism (CD) analysis was employed to assess the chiroptic properties of MBA-, NEA-, and BA-based 2D chiral perovskites at wavelengths of 500, 390, and 420 nm, respectively (**Fig. 1a**). *These wavelengths correspond to the absorption edges of the respective 2D chiral perovskites (Supplementary Fig. S1).* The CD signal of BA-based 2D chiral OIHP

Comment 10:

For the perovskite used in this work, when its solar cell form is directly connected to electrocatalysts for water splitting (i.e., PV-EC), are there any advantages to use chiral spin-selective catalysts to integrate with perovskite PVs?

Author's Response:

Our PEC device configuration is shown in Fig. R9a. Additionally, we attempted to operate a PV-EC system, as depicted in Fig. R9b, where a perovskite solar cell was connected to electrocatalysts via Cu wires. While this device structure offers advantages in terms of the moisture stability of perovskites, the performance of the PV-EC system significantly decreases compared to the PEC system as shown in Fig. R9c. This can be attributed to the device characteristics utilizing the CISS effect. The spin controlled by chiral materials has a spin propagation length on the scale of just several hundred micrometers (*Nat. Electron.* **2019**, 98, 98-107; *Nat. Commun.* **2023**, 14, 2831). However, connecting PV with EC through centimeter scale Cu wires leads to spin relaxation, causing the spin carriers to return to their random spin states before reaching the chiral spin-selective catalyst. Therefore, to apply the dual spin control strategy, it is imperative to utilize the configuration of the PEC device.

Fig. R9. Device configuration of (a) PEC system and (b) PV-EV system. (c) LSV curves of L-NF/S-NEA 2D/3D adopted as PEC and PV-EV system.

Comment 11:

In Figure 2b, there is a mistake in the legend for the orange curve and green curve.

Author's Response:

Thank you for your kindness. We have noted that the legends for the orange and dark green curves in Fig. 2b are correct. The orange curve represents the S88:P12/3D photoanode without NEA 2D, but instead utilizing the S88:P12 mixed HTL, while the dark green curve represents the S88:P12/S-NEA 2D/3D photoanode with S-NEA 2D inserted between the S88:P12 mixed HTL and 3D OHIP, as stated in the manuscript. To avoid confusion, we have also updated the legend in Fig. R10, highlighting "S-NEA 2D" in bold font for better visibility. We appreciate the reviewer's consideration.

Fig. R10. LSV curves of the OIHP photoanodes without and with the S-NEA 2D OIHP layer using various spiro-MeOTAD and P3HT combinations for the HTL conducted in K-Pi electrolyte (pH 6.5) under 1 sun illumination.

Revision made (colored in blue):

(Fig. 2b are modified)

Fig. 2. The PEC performance of the 2D chiral OIHP-based photoanodes and the effect of spin polarization therein.

<Reviewer 2>

This is a bold work of significant and novel results on halide perovskite photoanodes with enhanced photocurrents, assigned to spin filtering. However, there are some key characterizations missing to understand how much of the enhancement results from spin filtering and how much from just interfacial energetics favouring charge separation. Please address the following comments:

Remark:

We would like to thank the reviewer for evaluating our work. As reviewer suggested, we need to provide a deeper understanding of the spin-related catalysis and relevant explanation. Our response to the reviewer's comments can be found below.

Comment 1 & 2:

Please provide energy diagrams of studied photoanode and photocathode with all its layers involved. Related to the previous comment, it is known that 2D perovskite layers can be used on the surface of a 3D perovskite layer to promote charge separation and improve photocurrents and photovoltages. Such enhanced charge separation is obvious in the analysis of the energy diagrams, considering VB, CB, and E_f /work function alignment between the device layers including the transport layers (i.e. interfacial energetics). It is not clear in the manuscript that the authors have considered the resulting energy diagram. How much of the enhanced photocurrents are related to spin filtering across layers and how much to simple favoured interfacial energetics?

Author's Response:

As per the reviewer's comment, to estimate the band alignment and to evaluate the effect of the interfacial energetics according to the handedness of NEA 2D, we have constructed the band energy diagram for all the layers in the photoanode. Ultraviolet photoelectron spectroscopy (UPS) measurements were conducted to determine the precise position of the Fermi level (E_F) and to elucidate the band alignment between each layer (Fig. R11a). Figure R11b depicts the valence-band edge (E_{edge}) for each material, which represents the difference between valence band maximum (E_{VBM}) and E_F . Using the E_{edge} value, the definite VBM level of the sample was calculated as follows:

$$E_{\text{VBM}} = E_F - E_{\text{edge}} \quad (1)$$

The secondary-electron cutoff (E_{cutoff}) obtained via extrapolation to the linear part of the binding-energy edge is shown in Fig. R11c. The E_F of each material was calculated using the following equation:

$$E_F = E_{\text{cutoff}} - 21.21 \text{ eV (under He I radiation)} \quad (2)$$

Subsequently, the absolute value of the conduction band minimum (CBM) was obtained by adding the band gap with the E_{VBM} . The calculated band energy positions, as shown in Fig. R12a, can be organized according to energy levels, and when all layers form a junction, they align relative to the E_F , as depicted in Fig. R12b, forming a band alignment after equilibrium.

Firstly, it is observed that both *S*-NEA 2D and *rac*-NEA 2D are positioned at the same energy levels, including the valence band maximum (E_{VBM}), Fermi energy level (E_F), and conduction band minimum. This indicates that the performance difference between the NiFeOOH/Ni/Ketjen/HTL/*S*-NEA 2D/3D/SnO₂/FTO photoanode (Fig. 2b) and the NiFeOOH/Ni/Ketjen/HTL/*rac*-NEA 2D/3D/SnO₂/FTO photoanode (Supplementary Fig. S13c) is not attributed to the interfacial energetics resulting from the band alignment mismatch

of *S*-NEA 2D and *rac*-NEA 2D. Another notable observation is that the E_F of the 3D OIHP layer is approximately 0.2 eV higher than that of the NEA 2D OIHP layer. This leads to a back-field induced by the p-p heterojunction, explaining the enhanced performance of the S100/NEA 2D/3D photoanode although the use of spiro HTL having low spin propagation ability likely mitigates the CISS effect. Meanwhile, a Schottky junction likely forms between the 2D and 3D OIHP heterojunction, resulting in a slight band energy offset. Thin layer (20 nm) of chiral NEA 2D is responsible for the observed PEC performance enhancement through charge carrier tunneling effect, as observed in Supplementary Fig. S16. In conclusion, since there is no difference in band alignment depending upon the different handedness of NEA 2D, the enhanced performance of the NiFeOOH/Ni/Ketjen/HTL/*S*-NEA 2D/3D/SnO₂/FTO photoanode compared to the NiFeOOH/Ni/Ketjen/HTL/*rac*-NEA 2D/3D/SnO₂/FTO photoanode is not originated from the interfacial energetics. Instead, the enhancement in photocurrent and onset potential stems from the introduction of a chiral spin polarizer, inducing the CISS effect.

Fig. R11. (a) Ultraviolet photoelectron spectra for the $(\text{FAPbI}_3)_{0.95}(\text{MAPbBr}_3)_{0.05}$ 3D OIHP, (*S*-/*rac*-NEA) $_2\text{PbBr}_4$ 2D OIHP, and spiro-MeOTAD:P3HT obtained using He I radiation at 21.21 eV. (b) E_{edge} and (c) E_{cutoff} to calculate the band energy position of valence band maximum and Fermi energy level.

Fig. R12. Band alignment of Ni/S88:P12/NEA 2D/3D/SnO₂ configuration for (a) pre-equilibrium state and (b) equilibrium state.

Fig. R13. As the concentration of organic *S*-NEA cation increases (from 2.5 to 7.5 mg), it is possible to control the thickness of *S*-NEA layer on top of the 3D OIHP. LSV curves of *S*-NEA/3D photoanodes as a function of the thickness of *S*-NEA layer under 1 sun illumination in a K-Pi electrolyte (pH 6.5).

Revision made (colored in blue):

(Supporting Information; Supplementary Fig. S14 are added)

Supplementary Fig. S14. (a) Ultraviolet photoelectron spectra for the $(\text{FAPbI}_3)_{0.95}(\text{MAPbBr}_3)_{0.05}$ 3D OIHP, $(S\text{-}/rac\text{-NEA})_2\text{PbBr}_4$ 2D OIHP, and spiro-MeOTAD:P3HT obtained using He I radiation at 21.21 eV. (b) E_{edge} and (c) E_{cutoff} to calculate the band energy position of valence band maximum and Fermi energy level.

(Supporting Information; Supplementary Fig. S15 are added)

Supplementary Fig. S15. Band alignment of Ni/S88:P12/NEA 2D/3D/SnO₂ configuration for (a) pre-equilibrium state and (b) equilibrium state.

Supplementary Fig. S16. As the concentration of organic *S*-NEA cation increases (from 2.5 to 7.5 mg), it is possible to control the thickness of *S*-NEA layer on top of the 3D OHIP. LSV curves of *S*-NEA/3D photoanodes as a function of the thickness of *S*-NEA layer under 1 sun illumination in a K-Pi electrolyte (pH 6.5).

(Supporting Information; Supplementary Note 2 are added)

Supplementary Note 2. Methodology of UPS spectra analysis.

Firstly, Supplementary Fig. S14b depicts the valence-band edge (E_{edge}) for each material, which represents the difference between valence band maximum (E_{VBM}) and Fermi energy level (E_{F}).

Using the E_{edge} value, the definite VBM level of the sample was calculated as follows:

$$E_{\text{VBM}} = E_{\text{F}} - E_{\text{edge}} \quad (\text{S2})$$

The secondary-electron cutoff (E_{cutoff}) obtained via extrapolation to the linear part of the binding-energy edge is shown in Supplementary Fig. S14c. The E_{F} of each material was calculated using the following equation:

$$E_{\text{F}} = E_{\text{cutoff}} - 21.21 \text{ eV (under He I radiation)} \quad (\text{S3})$$

(page 13, line 16)

To estimate the band alignment and to evaluate the effect of the band bending according to the handedness of NEA 2D, the band energy position of all the layers in the photoanode was obtained by ultraviolet photoelectron spectroscopy (UPS), as shown in Supplementary Fig. S14a. Details of the experimental procedure and relevant results are provided in Supplementary Fig. S14b,c and Supplementary Note 2. The relative band energy positions of the 3D, NEA 2D, HTL, and SnO₂ for the pre-equilibrium and equilibrium states are plotted in Supplementary Fig. S15a and b, respectively. Firstly, it is observed that both *S*-NEA 2D and *rac*-NEA 2D are positioned at the same energy levels, including the valence band maximum (E_{VBM}), Fermi energy level (E_{F}), and conduction band minimum. This indicates that the performance difference between the NiFeOOH/Ni/Ketjen/HTL/*S*-NEA 2D/3D/SnO₂/FTO photoanode (Fig.

2b) and the NiFeOOH/Ni/Ketjen/HTL/*rac*-NEA 2D/3D/SnO₂/FTO photoanode (Supplementary Fig. S13c) is not attributed to the band alignment mismatch of *S*-NEA 2D and *rac*-NEA 2D. Another notable observation is that the E_F of the 3D OIHP layer is approximately 0.2 eV higher than that of the NEA 2D OIHP layer. This leads to a back-field induced by the p-p heterojunction, explaining the enhanced performance of the S100/NEA 2D/3D photoanode although the use of spiro HTL having low spin propagation ability likely mitigates the CISS effect. After reaching an equilibrium state, a Schottky junction likely forms between the 2D and 3D OIHP heterojunction, resulting in a slight band energy offset. Thin layer (20 nm) of chiral NEA 2D is responsible for the observed performance enhancement through electron tunneling effect, as observed in Supplementary Fig. S16. However, as thickness increases, performance decreases. Therefore, because there is no difference in band alignment depending upon the different handedness of NEA 2D, the enhanced performance of the NiFeOOH/Ni/Ketjen/HTL/*S*-NEA 2D/3D/SnO₂/FTO photoanode compared to the NiFeOOH/Ni/Ketjen/HTL/*rac*-NEA 2D/3D/SnO₂/FTO photoanode is not originated from the interfacial energetics. Instead, the enhancement in photocurrent and onset potential stems from the introduction of a chiral spin polarizer, inducing the CISS effect.

(Methods, page8)

operating at an accelerating voltage of 2 kV. The energy level of each semiconductor was analyzed by ultraviolet photoelectron spectroscopy (UPS, Axis-NOVA, and Ultra DLD, UK) under He I radiation (21.21 eV) to investigate the energy band position of sample. The secondary-electron cutoff region (E_{cutoff}) and valence-band edge (E_{edge}) were calculated according to Supplementary Note 2 (Supporting Information).

Comment 3:

There is not much characterization of the device as a PV before being used as a photoanode, before and after the addition of those chiral layers. Please provide clear PV characterization. This may help to address previous comments

Author's Response:

We appreciate the reviewer for important comment about the characterization of the PV device to reveal the origin of the performance enhancement. We fabricated 3D OIHP photovoltaic (PV) cells in a configuration of Au/S88:P12/(NEA 2D)/3D/SnO₂/FTO structure, as shown in Fig. R14a, b, and conducted PV characterization. When comparing champion PV cells, the 3D PV cell without NEA 2D exhibited notably lower performance (Fig. R14c). Conversely, the 3D PV cell with an inserted NEA 2D showed similar performance regardless of the handedness of the chiral NEA molecule. Even when statistics were compiled for 10 PV cells, the NEA 2D/3D structure consistently showed higher average values, whereas PV parameters decreased when the NEA 2D layer was absent (Fig. R15). The incorporation of NEA 2D OIHP improves the photovoltaic characteristics due to the passivation effect. However, since *rac*-, *R*-, and *S*-NEA 2D have identical interfacial energetics, the photovoltaic characteristics remain unaffected by the chirality of NEA 2D. Consequently, the enhancement in PEC performance upon adding chiral *S*-NEA 2D onto the 3D PEC device is primarily attributed to the CISS effect rather than photovoltaic performance enhancement.

Fig. R14. Schematic illustration of the 3D OIHP photovoltaic cells configuration (a) with or (b) without NEA 2D OIHP. (c) J–V curves and photovoltaic parameters of the champion PV cells based on the 3D, *rac*-NEA 2D/3D, *R*-NEA 2D/3D, and *S*-NEA 2D/3D OIHPs.

Fig. R15. Statistical distribution of the photovoltaic parameters for the 3D, *rac*-NEA 2D/3D, *R*-NEA 2D/3D, and *S*-NEA 2D/3D based PV cells.

Revision made (colored in blue):

(Supporting Information; Supplementary Fig. S17 are added)

Supplementary Fig. S17. Schematic illustration of the 3D OIHP photovoltaic cells configuration (a) with or (b) without NEA 2D OIHP. (c) J–V curves and photovoltaic parameters of the champion PV cells based on the 3D, *rac*-NEA 2D/3D, *R*-NEA 2D/3D, and *S*-NEA 2D/3D OIHPs.

(Supporting Information; Supplementary Fig. S18 are added)

Supplementary Fig. S18. Statistical distribution of the photovoltaic parameters for the 3D, *rac*-NEA 2D/3D, *R*-NEA 2D/3D, and *S*-NEA 2D/3D based PV cells.

(page 14, line 17)

We fabricated 3D OIHP photovoltaic (PV) cells in a configuration of Au/S88:P12/(NEA 2D)/3D/SnO₂/FTO structure, as shown in Supplementary Fig. S17a, b, and conducted PV characterization. When comparing champion PV cells, the 3D PV cell without NEA 2D exhibited notably lower performance (Supplementary Fig. S17c). Conversely, the 3D PV cell with an inserted NEA 2D showed similar performance regardless of the handedness of the chiral NEA molecule. Even when statistics were compiled for 10 PV cells, the NEA 2D/3D structure consistently showed higher average values, whereas PV parameters decreased when the NEA 2D layer was absent (Supplementary Fig. S18). The incorporation of NEA 2D OIHP improves the photovoltaic characteristics due to the passivation effect. However, since *rac*-, *R*-, and *S*-NEA 2D have identical interfacial energetics, the photovoltaic characteristics remain unaffected by the chirality of NEA 2D. Consequently, the enhancement in PEC performance upon adding chiral *S*-NEA 2D onto the 3D PEC device is primarily attributed to the CISS effect rather than photovoltaic performance enhancement.

Comment 4:

What is the maximum stability of the device, and what causes its death? Please explain it in the manuscript

Author's Response:

The maximum stability of the dual spin-controlled OER device was measured under simulated 1-sun continuous illumination condition at 0.5 M K-Pi (pH 6.5), maintaining up to 80% of the initial current density for 160 h before experiencing a rapid decline in performance (Fig. R16a). To identify the cause of the abrupt performance degradation after long-term operation, the front and rear of the sample were examined (Fig. R16b). While no catalyst detachment or Ni foil delamination was observed at the front photoelectrode, a color change from black to yellow was observed in the area of the 3D OIHP at the back side. This suggests that the 3D OIHP layer degrades during electrochemical reaction. Other studies have indicated that unextracted excess electrons are a major contributing factor to iodine migration-induced degradation (*Nat. Commun.* **2018**, 9, 4981), with iodine vacancies having the lowest activation energy for migration compared to other ions, making them more likely to migrate first to the electron-accumulated interfacial region (*Nat. Commun.* **2015**, 6, 7497). Consequently, when iodine ions migrate toward the HTL, they trigger oxidation reactions, leading to the formation of PbI_2 and degradation of the perovskite. To confirm that catalyst degradation was not the main cause of stability decline, the LSV curves of *L*-NF before and after stability measurements were compared (Fig. R16c). The *L*-NF sustained nearly similar current density even after 160 h, supporting that the catalyst was not the primary cause of degradation. We express our gratitude to the reviewer for enhancing the solidity of our manuscript.

Fig. R16. (a) Operational stability of *L*-NF/*S*-NEA 2D/3D OIHP photoanode under continuous operation in 0.5 M K-Pi (pH 6.5) electrolyte at 1.2 V_{RHE} in an ambient atmosphere. (b) Photographs of front and back of the OIHP photoanode after the stability test. (c) LSV curves of *L*-NF before and after stability test.

Revision made (colored in blue):

(Supporting Information; Supplementary Fig. S32b and c are added)

Supplementary Fig. S32. (a) Operational stability of *L*-NF/*S*-NEA 2D/3D OIHP photoanode under continuous operation in 0.5 M K-Pi (pH 6.5) electrolyte at 1.2 V_{RHE} in an ambient atmosphere. (b) Photographs of front and back of the OIHP photoanode after the stability test. (c) LSV curves of *L*-NF before and after stability test.

(page 20, line 25)

The device retained 80% of the initial current density for 160 h before experiencing a rapid decline in performance (Supplementary Fig. S32a). To identify the cause of the abrupt performance degradation after long-term operation, the front and rear of the sample were examined (Supplementary Fig. S32b). While no catalyst detachment or Ni foil delamination was observed at the front photoelectrode, a color change from black to yellow was observed in the area of the 3D OIHP at the back side. This suggests that the 3D OIHP layer degrades during

electrochemical reaction. Other studies have indicated that unextracted excess electrons are a major contributing factor to iodine migration-induced degradation,⁴⁹ with iodine vacancies having the lowest activation energy for migration compared to other ions, making them more likely to migrate first to the electron-accumulated interfacial region.⁵⁰ Consequently, when iodine ions migrate toward the HTL, they trigger oxidation reactions, leading to the formation of PbI_2 and degradation of the perovskite. To confirm that catalyst degradation was not the main cause of stability decline, the LSV curves of *L*-NF before and after stability measurements were compared (Supplementary Fig. S32c). The *L*-NF sustained nearly similar current density even after 160 h, supporting that the catalyst was not the primary cause of degradation. We tested 20 samples,

(References)

49. Lin, Y., Chen, B., Fang, Y., Zhao, J., Bao, C., Yu, Z., Deng, Y., Rudd, P. N., Yan, Y., Yuan Y. & Huang J. Excess charge-carrier induced instability of hybrid perovskites. *Nat. Commun.* **9**, 4981 (2018)

50. Eames, C., Frost, J. M., Barnes, P. R. F., O'Regan, B. C., Walsh A. & Islam, M. S. Ionic transport in hybrid lead iodide perovskite solar cells. *Nat. Commun.* **6**, 7497 (2015)

Comment 5:

Please avoid acronyms in abstract and conclusions

Author's Response:

Thank you for your thoughtful feedback. We have made efforts to modify the acronyms in both the Abstract and Conclusion sections as suggested.

Revision made (colored in blue):

(Abstract)

The oxygen evolution reaction, which involves high overpotential and slow charge-transport kinetics, plays a critical role in determining the efficiency of solar-driven water splitting. The chiral-induced spin selectivity phenomenon has been utilized to reduce by-product production and hinder charge recombination, thereby enhancing the oxygen evolution reaction process and encompassing a major breakthrough in photoelectrochemical water splitting. To fully exploit the spin polarization effect, we herein propose a dual spin-controlled perovskite photoelectrode. The 3D perovskite serves as an efficient light absorber layer while the 2D chiral perovskite functions as a spin polarizer to effectively align the spin states of charge carriers in the 2D chiral perovskite/3D-stacked arrangement. Compared to other investigated chiral organic cations, *R/S*-NEA enable strong spin-orbital coupling due to strengthened π - π stacking interactions. The resulting NEA-based chiral 2D/3D perovskite photoelectrodes achieved a high spin polarizability of approximately 75%. Moreover, spin relaxation was prevented by employing a chiral spin-selective *L*-NiFeOOH catalyst on top of the hole transport layer, which enables the secondary spin alignment to promote the generation of triplet oxygen. This dual spin-controlled 2D/3D perovskite photoanode provides efficient solar-driven water splitting, achieving a remarkable 13.17% of applied-bias photon-to-current efficiency with a photocurrent density of 23 mA cm⁻² at 1.23 V_{RHE} and an onset potential of 0.2 V_{RHE}. After connecting the perovskite photocathode with *L*-NiFeOOH/*S*-NEA 2D/3D photoanode in series, the resulting co-planar photoelectrochemical–photoelectrochemical water-splitting device exhibited a solar-to-hydrogen efficiency of 12.55%, the best reported for perovskite-based overall photoelectrochemical water splitting so far.

(Conclusion)

We successfully demonstrated spin-dependent performance enhancement of water splitting by creating and employing a dual spin-control strategy-based **perovskite** photoanode. The sluggish kinetics of the **oxygen evolution reaction** were promoted by exploiting the CISS effect in the photoanode, which also suppressed side reactions and intermediate hydrogen peroxide formation. The CISS effect was introduced by growing an *S*-NEA-based 2D chiral **perovskite** layer on top of a 3D **perovskite** light absorber through which the photogenerated charge carriers were spin-polarized by chiral NEA molecules capable of strong π - π interactions and enhanced **spin orbital coupling**. Spin relaxation during charge transport to the catalyst/electrolyte interface was further diminished not only by adopting a mixed HTL having a relatively long spin diffusion length but also by employing a chiral spin selective catalyst on top of the HTL. The 2D **perovskite** SP deposited on the 3D **perovskite** in the photoanode decorated with a chiral spin selective catalyst (*L*-NF) augmented the spin alignment to promote the generation of $^3\text{O}_2$ while lowering the overpotential and enhancing the PEC water-splitting efficiency. Furthermore, we elucidated the mechanism of the CISS phenomenon synergistically induced by the combined use of the SP, spin-transportable HTL, and the SCC on charge transport and recombination in the **oxygen evolution reaction**. Our dual spin-controlled *L*-NF/*S*-NEA 2D/3D photoanode demonstrated a remarkable ABPE of 13.17% with a J_{ph} of 23 mA cm⁻² at 1.23 V_{RHE} and a V_{onset} of 0.2 V_{RHE}. Interestingly, after connecting the decoupled catalyst-structured **perovskite** photocathode with an *L*-NF/*S*-NEA 2D/3D photoanode in series, the co-planar PEC-PEC water-splitting device exhibited an excellent STH efficiency of 12.55%, the best figure reported for a **perovskite**-based overall PEC water-splitting device. Our unassisted water-splitting device also demonstrated impressive stability by maintaining 100% of its initial current density after 24 h while producing highly pure hydrogen and oxygen.

Comment 6:

H₂O₂ is claimed to be a byproduct of the photoanode. Any proof of it?

Author's Response:

We greatly appreciate the reviewer's insightful comment. As indicated, one of the pieces of evidence supporting the enhancement of PEC performance due to the CISS effect is the effective suppression of the byproduct, H₂O₂. To confirm the suppression of by-product (H₂O₂) as an evidence for spin-controlled OER, we conducted colorimetric experiment to quantify the generated amount of H₂O₂ by observing the time-dependent absorbance spectra of the electrolyte aliquot collected as a function of reaction duration, as shown in Fig. R17. The absorbance arises from the presence of a reactant between o-tolidine and H₂O₂ so that the absorption varies depending upon the amount of H₂O₂ in the electrolyte. Colorimetric experiment was conducted under potentiostatic condition in 0.5 M Na₂SO₄ electrolyte, enabling the detection of generated H₂O₂ at 1.0 V_{RHE}.

Furthermore, we utilized an oxygen sensor to detect O₂ simultaneously, allowing us to calculate the FE_{O₂} as a function of reaction duration, as illustrated in Fig. R18. For the 3D OIHP photoanode utilizing *meso*-NF catalysts, the FE_{O₂} was calculated to be 73%, while the FE for H₂O₂ (FE_{H₂O₂}) was 16%, as shown in Fig. R18a. Even with the addition of *racemic*-type achiral 2D on top of 3D OIHP (Fig. R18b), the FE_{O₂} and FE_{H₂O₂} remained similar at 72% and 14%, respectively. However, upon the insertion of *S*-NEA-based chiral 2D OIHP (Fig. R18c), the FE_{O₂} increased to 78%, while the FE_{H₂O₂} decreased to 8%, indicating that the CISS effect works on OER device. Moreover, replacing *meso*-NF with chiral *L*-NF further enhanced the FE_{O₂} and reduced the FE_{H₂O₂}, achieving FE_{O₂} and FE_{H₂O₂} of 90% and 3%, respectively (Fig. R18d). This clearly demonstrates that the dual spin control strategy manipulates the spin state of charge carriers for an extended period, leading to the suppression of by-product H₂O₂ and the formation of lower-energy triplet oxygen, resulting in a significant improvement in FE_{O₂}.

Thanks to the reviewer's critical comment, we were able to enhance the quality of the paper through appropriate revisions.

Fig. R17. UV-vis absorption spectra from o-toluidine titration of electrolytes (0.5 M Na₂SO₄) as a function of reaction duration for (a) *meso*-NF/3D OIHP, (b) *meso*-NF/*rac*-NEA 2D/3D OIHP, (c) *meso*-NF/*S*-NEA 2D/3D OIHP, and (d) *L*-TA/*S*-NEA 2D/3D OIHP.

Fig. R18. Calculated O₂ and H₂O₂ Faradaic efficiency stacked bar chart for (a) meso-NF/3D OIHP, (b) meso-NF/rac-NEA 2D/3D OIHP, (c) meso-NF/S-NEA 2D/3D OIHP, and (d) L-TA/S-NEA 2D/3D OIHP.

Revision made (colored in blue):

(Supporting Information; Supplementary Fig. S34 are added)

Supplementary Fig. S34. UV-vis absorption spectra from o-toluidine titration of electrolytes (0.5 M Na₂SO₄) as a function of reaction duration for (a) *meso*-NF/3D OIHP, (b) *meso*-NF/*rac*-NEA 2D/3D OIHP, (c) *meso*-NF/S-NEA 2D/3D OIHP, and (d) L-TA/S-NEA 2D/3D OIHP.

(Supporting Information; Supplementary Fig. S35 are added)

Supplementary Fig. S35. Calculated O₂ and H₂O₂ Faradaic efficiency stacked bar chart for (a) *meso*-NF/3D OIHP, (b) *meso*-NF/*rac*-NEA 2D/3D OIHP, (c) *meso*-NF/*S*-NEA 2D/3D OIHP, and (d) *L*-TA/*S*-NEA 2D/3D OIHP.

(page 21, line 19)

This observation clearly suggests the success of implementing the CISS effect in our dual spin-controlled photoanode so that the photogenerated charges appropriately generate the desired product (O₂) while effectively suppressing the hydrogen peroxide side reaction. To confirm the suppression of by-product (H₂O₂) as an evidence for spin-controlled OER, we conducted colorimetric experiment to quantify the generated amount of H₂O₂ by observing the time-dependent absorbance spectra of the electrolyte aliquot collected as a function of reaction duration, as shown in Supplementary Fig. S34. The absorbance arises from the presence of a

reactant between o-tolidine and H_2O_2 so that the absorption varies depending upon the amount of H_2O_2 in the electrolyte. Colorimetric experiment was conducted under potentiostatic condition in 0.5 M Na_2SO_4 electrolyte, enabling the detection of generated H_2O_2 at 1.0 V_{RHE} . Furthermore, we utilized an oxygen sensor to detect O_2 simultaneously, allowing us to calculate the FE_{O_2} as a function of reaction duration, as illustrated in Supplementary Fig. S35. For the 3D OIHP photoanode utilizing *meso*-NF catalysts, the FE_{O_2} was calculated to be 73%, while the FE for H_2O_2 ($\text{FE}_{\text{H}_2\text{O}_2}$) was 16%, as shown in Supplementary Fig. S35a. Even with the addition of *racemic*-type achiral 2D on top of 3D OIHP (Supplementary Fig. S35b), the FE_{O_2} and $\text{FE}_{\text{H}_2\text{O}_2}$ remained similar at 72% and 14%, respectively. However, upon the insertion of *S*-NEA-based chiral 2D OIHP (Supplementary Fig. S35c), the FE_{O_2} increased to 78%, while the $\text{FE}_{\text{H}_2\text{O}_2}$ decreased to 8%, indicating that the CISS effect works on OER device. Moreover, replacing *meso*-NF with chiral *L*-NF further enhanced the FE_{O_2} and reduced the $\text{FE}_{\text{H}_2\text{O}_2}$, achieving FE_{O_2} and $\text{FE}_{\text{H}_2\text{O}_2}$ of 90% and 3%, respectively (Supplementary Fig. S35d). This clearly demonstrates that the dual spin control strategy manipulates the spin state of charge carriers for an extended period, leading to the suppression of by-product H_2O_2 and the formation of lower-energy triplet oxygen, resulting in a significant improvement in FE_{O_2} .

(Methods, page 41)

EIS was conducted using the same instrument as for the PEC measurements, along with a frequency analyzer (1260, Solartron, Leicester, UK). The IPCE, IMPC, and IMPV measurements were conducted using an electrochemical workstation (Zennium, Zahner, Germany) and a potentiostat (PP211, Zahner, Germany) with a monochromatic light source (TLS03, Zahner). Electrochemical analysis was performed at 1.0 V_{RHE} in 0.5 M Na_2SO_4 (pH 6.5) under AM 1.5 G (100 mW cm^{-2}) illumination for measuring H_2O_2 and O_2 . The

measurements of O₂ and H₂O₂ were taken every 30 min during the electrochemical reaction of 3 h. An oxygen sensor (NeoFox Phase Fluorometer, Ocean Insight, USA) was utilized for the O₂ detection. All of the device connections were completely sealed with rubber bulkheads to prevent gas leakage from the quartz reactor. The cell was purged with Ar prior to the measurement to ensure an O₂ content lower than 0.5%. Hydrogen peroxide formation was detected using colorimetric titration method. To detect hydrogen peroxide, 1 mL of o-tolidine, a redox indicator, and 3 mL of the electrolyte aliquot extracted as a function of reaction duration were mixed, followed by performing UV-vis absorption spectroscopy. To accurately determine the H₂O₂ amount, a calibration curve was constructed using a 30% w/w commercial H₂O₂ solution (Sigma-Aldrich, USA).

Comment 7:

When hydrogen and oxygen gases are claimed to be generated at rates of 11.2 and 5.6 $\mu\text{mol h}^{-1}$, respectively, please provide area.

Author's Response:

We appreciate the reviewer's meticulous advice. The data presented in our study correspond to measurements conducted using an active area of 0.06 cm², identical to the photocathode and photoanode. We have specified the area in our manuscript.

Revision made (colored in blue):

(Supplementary Information; Caption of the Supplementary Fig. S41 are modified)

Supplementary Fig. S41. The Faradaic efficiency of the co-planar photocathode-photoanode water-splitting device obtained by comparing the experimental gas chromatography values (spot patterns) with the theoretical values in Fig. 5d (dashed lines) under an unbiased voltage. The active area was 0.06 cm² for each photoelectrode.

(page 27, line 25)

Gas chromatography was conducted to quantitatively assess the products from unbiased solar water-splitting (Supplementary Fig. S29), indicating that hydrogen and oxygen gases were generated at rates of 11.2 and 5.6 μmol h⁻¹ with the active area of 0.06 cm² for each photoelectrode, respectively.

Comment 8:

Is the Ni foil in direct contact with the Ni foam, or do holes travel through the silver paste? Can this result in any voltage losses? How mechanically stable is that silver paste?

Author's Response:

We agree with the reviewer's concern regarding the feasibility of conductivity loss due to the passivation layer during the conversion of PV cells to PEC cells. To clearly discern this, we observed the changes in PV characteristics when adding either Ketjen/Ni foil or Ketjen/Ni foil/Ag paste/Ni foam onto the Au/Spiro:P3HT/S-NEA 2D/3D/SnO₂/FTO structure. As shown in Fig. R19a, the open circuit voltage (V_{oc}) and fill factor (FF) slightly decreased with the increasing thickness of the passivation layer. However, there was no significant difference in short circuit current density (J_{sc}), and the average power conversion efficiency (PCE) remained similar. To address the reviewer's concerns regarding voltage loss through Ni and Ag, we evaluated the performance by creating a PV-EC system in which PV is directly connected to

electrocatalysts, as shown in Fig. R9b. However, the performance of PV-EC system significantly decreases compared to the PEC system. This can be attributed to the device characteristics utilizing the CISS effect. The spin controlled by chiral materials has a spin propagation length on the scale of just several hundred micrometers. However, connecting PV with EC through centimeter scale Cu wires leads to spin relaxation, causing the spin carriers to return to their original spin states before reaching the chiral spin-selective catalysts. Therefore, to apply the dual spin control strategy, it is imperative to utilize the configuration of the PEC device. Additionally, as evident from Fig. R16b, even after stability measurements of over 160 h, the *L*-NF catalyst adhered well by the Ag paste, indicating sufficient mechanical and electrochemical stability.

Fig. R19. (a) J–V curves and photovoltaic parameters of the champion *S*-NEA 2D/3D based PV cells as well as the same cells additionally passivated with the Ketjen/Ni foil and Ketjen/Ni foil/Ag/Ni foam. (b) Statistical distribution of the photovoltaic parameters for the *S*-NEA 2D/3D based PV cells as the same cells additionally passivated with the Ketjen/Ni foil and Ketjen/Ni foil/Ag/Ni foam.

Comment 9:

In figure 2b, the legend shows two different S88:P12 J-V curves. An error?

Author's Response:

Thank you for your consideration. We have noted that the legends for the orange and dark green curves in Fig. 2b are correct. The orange curve represents the S88:P12/3D photoanode without NEA 2D, but instead utilizing the S88:P12 mixed HTL, while the dark green curve represents the S88:P12/S-NEA 2D/3D photoanode with S-NEA 2D inserted between the S88:P12 mixed HTL and 3D OHIP, as stated in the manuscript. To avoid confusion, we have also updated the legend in Fig. R20, highlighting "S-NEA 2D" in bold font for better visibility. We appreciate the reviewer's thoughtfulness.

Fig. R20. LSV curves of the OIHP photoanodes without and with the S-NEA 2D OIHP layer using various spiro-MeOTAD and P3HT formulations for the HTL conducted in K-Pi electrolyte (pH 6.5) under 1 sun illumination.

Revision made (colored in blue):

(Fig. 2b are modified)

Fig. 2. The PEC performance of the 2D chiral OIHP-based photoanodes and the effect of spin polarization therein.

Comment 10:

It is not explained how Cu wire was connected to the Au electrode region.

Author's Response:

We appreciate the valuable comment from the reviewer regarding the fabrication method of our 3D OIHP photocathode. To enhance the top layer of the photocathode, we employed the evaporation technique to deposit a 70 nm layer of Au. Subsequently, we attached the Cu wire

with the Au electrode using Ag paste. The structure was then solidified at room temperature for 12 h, ensuring a firm and stable attachment. We have provided additional details on these methodological aspects.

Revision made (colored in blue):

(Methods, page 39)

Subsequently, an Au top electrode with a thickness of 70 nm was deposited via thermal evaporation, after which Cu wire was connected to the Au electrode region using Ag paste. The structure was then solidified at room temperature for 12 h, ensuring a firm and stable attachment. To prevent water infiltration, the OIHP photocathode was thoroughly sealed using epoxy resin.

Comment 11:

How much Pt was deposited?

Author's Response:

We utilized a 108 Auto Sputter coater to deposit Pt, which was operated for 120 s under 0.1 mbar of Ar atmosphere and a current of 10 mA. Upon Pt deposition, a decrease in transmittance was observed using UV-Vis spectroscopy, indicating uniform sputtering of the Pt catalyst, as depicted in Fig. R21a. Detecting the difference in weight before and after sputtering Pt catalyst is challenging due to the very low loading amount (measurement is difficult under 10 μg due to the instrument limit of the electronic scale). Instead, SEM analysis was conducted to observe the microstructure of the sputtered Pt catalyst, confirming the deposition of Pt catalyst in the form of nanoparticles of 10 nm or less, as shown in Fig. R21b. We inferred from this that the

size of Pt deposited via sputtering was in the sub-nanometer range, and additional details regarding the experimental methodology have been included. Thanks to the reviewer's comments, our paper has been further enhanced.

Fig. R21. (a) Transmittance data of bare FTO and Pt sputtered for 120 s on the FTO substrate via UV-Vis spectroscopy. (b) SEM images of bare FTO and Pt sputtered for 120 s on the FTO substrate.

Revision made (colored in blue):

(Methods, page 39)

The Pt catalyst was placed on the decoupled region of the FTO substrate by employing a 108 Auto Sputter coater (Ted Pella, Redding, CA, USA), which was operated for 120 s under 0.1 mbar of Ar atmosphere and a current of 10 mA.

<Reviewer 3>

The manuscript describes the use of 2D chiral perovskite combined with 3D perovskite and chiral catalysts for overall water splitting. While it addresses an interesting application of 2D chiral perovskites, and is a comprehensive piece of work, there are several confusing points, as detailed below, that should be addressed before the consideration for publication.

Remark:

We thank the reviewer for reviewing our work and providing insightful comments and constructive suggestions. We have carefully revised our manuscript according to the suggestions, with more specific analysis and profound consideration. We sincerely hope that the details provided below will satisfy the reviewer's concerns.

Comment 1:

"OIHP" and "ABPE" in the abstract are not defined.

Author's Response:

We thank the reviewer for their valuable comment regarding the acronyms used in the abstract. We have revised minimize the use of acronyms to avoid confusion.

Revision made (colored in blue):

(Abstract)

The oxygen evolution reaction, which involves high overpotential and slow charge-transport kinetics, plays a critical role in determining the efficiency of solar-driven water splitting. The chiral-induced spin selectivity phenomenon has been utilized to reduce by-product production

and hinder charge recombination, thereby enhancing the oxygen evolution reaction process and encompassing a major breakthrough in photoelectrochemical water splitting. To fully exploit the spin polarization effect, we herein propose a dual spin-controlled perovskite photoelectrode. The 3D perovskite serves as an efficient light absorber layer while the 2D chiral perovskite functions as a spin polarizer to effectively align the spin states of charge carriers in the 2D chiral perovskite/3D-stacked arrangement. Compared to other investigated chiral organic cations, *R/S*-NEA enable strong spin-orbital coupling due to strengthened π - π stacking interactions. The resulting NEA-based chiral 2D/3D perovskite photoelectrodes achieved a high spin polarizability of approximately 75%. Moreover, spin relaxation was prevented by employing a chiral spin-selective *L*-NiFeOOH catalyst on top of the hole transport layer, which enables the secondary spin alignment to promote the generation of triplet oxygen. This dual spin-controlled 2D/3D perovskite photoanode provides efficient solar-driven water splitting, achieving a remarkable 13.17% of applied-bias photon-to-current efficiency with a photocurrent density of 23 mA cm⁻² at 1.23 V_{RHE} and an onset potential of 0.2 V_{RHE}. After connecting the perovskite photocathode with *L*-NiFeOOH/*S*-NEA 2D/3D photoanode in series, the resulting co-planar photoelectrochemical–photoelectrochemical water-splitting device exhibited a solar-to-hydrogen efficiency of 12.55%, the best reported for perovskite-based overall photoelectrochemical water splitting so far.

Comment 2:

In discussing “In addition, the spin polarization of chiral catalysts remains relatively low at around 50%”, it would be useful to use an equation to define the spin polarization explicitly.

Author's Response:

We appreciate the detailed comment. Following this advice, we have explicitly defined the degree of spin polarization along with the relevant equation. The anisotropy of the spin-dependent current represents the spin-polarization degree (P_V), as defined by the following equation:

$$P_V = \frac{I_{down} - I_{up}}{I_{down} + I_{up}} \times 100\%, \quad (\text{R3})$$

where I_{down} and I_{up} represent the measured spin-dependent currents at specific voltage (V) when the tip is pre-magnetized with either down- or up-field orientations, respectively. Following the reviewer's advice, it is more logical to compare the spin polarization of chiral catalysts after the definition. We are grateful for the insightful comment.

Revision made (colored in blue):

(page 4, line 15)

However, such a photoanode generally exhibits a low photocurrent and high overpotential because the oxide semiconductor absorber retains poor light harvesting ability and electrical conductivity. The anisotropy of the spin-dependent current represents the spin-polarization degree (P_V), as defined by the following equation:

$$P_V = \frac{I_{down} - I_{up}}{I_{down} + I_{up}} \times 100\%, \quad (1)$$

where I_{down} and I_{up} are the measured spin-dependent currents at specific voltage V when the tip is pre-magnetized with either down or up field orientations, respectively. The spin polarization of the chiral catalysts remains relatively low at around 50%. Therefore,

Comment 3:

The sentence “The introduction of NEA organic cations can improve the angular momentum of chiral 2D OIHPs, thereby leading to enhanced circular polarization” is ambiguous. What angular momentum is being improved (electrons in the conduction band, electrons in the valence band, etc)? The phrase “angular momentum” has been used in other paragraphs as well, so it is important to clarify what it means. Is “enhanced circular polarization” referring to enhanced circular dichroism, or something else?

Author’s Response:

We appreciate the critical comment regarding the explanation of the improvement resulting from the introduction of NEA organic cations. When we used the term of "angular momentum," we intended to convey the rotation and tilting of the lattice structure induced by the interaction between various organic chiral cations and the lead halide layer as they enter the interspace between the inorganic layers. However, as pointed out by the reviewer, using the term "angular momentum" may lead to misconception, as it could imply a direct influence on the degeneracy of energy states. Therefore, it seems more appropriate to revise the description of the change induced by chiral organic cations to "asymmetric distortion" of the inorganic layer. Additionally, to convey the concept more clearly, we have revised "enhanced circular polarization" to "enhanced circular dichroism." We express our gratitude to the reviewer for his/her invaluable comment.

Revision made (colored in blue):

(page 5, line 15)

To reinforce this phenomenon, we utilized naphthyl ethylamine (NEA) having larger delocalized molecular orbitals than MBA, and thus capable of stronger π - π stacking

interactions between the naphthalene rings. The introduction of NEA organic cations can improve the asymmetric distortion of inorganic layer, thereby leading to enhanced circular dichroism and improved chiroptical properties.³² Furthermore,

(page 7, line 8)

implying that the former has lower chiroptical activity than the latter. Chiroptical properties are strongly influenced by chiral-induced helical electric field, which is proportional to the radius of the helically arranged chiral cations. In this context, the smaller size of BA in comparison to MBA leads to the formation of a helix with lower asymmetric distortion, resulting in lower chiroptical activity.³² On the other hand,

Comment 4:

For a strictly racemic mixture compound, the spin polarization degree should be zero. Is there any rationale behind the observed small degree of spin polarization (a few percent)?

Author's Response:

A racemic mixture contains equal proportion of left- and right-handed enantiomers of a chiral molecule. When *rac*-NEA 2D OIHP was observed through mCP-AFM measurements after pre-magnetizing the tip (Fig. R22a), the current mapping image consisted of many sub-regions exhibiting predominantly either *R*-type or *S*-type characteristics. For example, four distinct sub-regions showing the current behavior based on up-/down-magnetized states are displayed in Fig. R22b. In sub-region where the *R*-configuration is slightly dominant, such as regions I and IV, an up-magnetization preferred degree of spin polarization ($P_{@-4V} > 0$) is observed. Conversely, in sub-region where the *S*-configuration is slightly dominant, such as regions II and III, a down-magnetization preferred degree of spin polarization ($P_{@-4V} < 0$) is observed.

The degree of spin polarization for *rac*-NEA 2D still remains significantly lower than those of *R/S*-NEA 2D, indicating the absence of the CISS effect. While achiral *meso*-type molecules would exhibit a 0% degree of spin polarization, chiral NEA molecules do not possess *meso*-type configuration. Hence, we attempted the mCP-AFM measurements using NF catalysts synthesized with various configurations of tartaric acid, including *L/D*-tartaric acid as well as *meso*-type.

Both *L*-NF and *D*-NF demonstrate favorable spin polarization behavior for down- and up-magnetization, respectively (Fig. R23a,b). In the case of *racemic*-NF (*rac*-NF), synthesized in the presence of a mixture of *D* and *L* enantiomers at equal ratio, tiny spin polarization effect is observed, albeit at a very low efficiency of 1.87% (Fig. R23c). By contrast, the *meso*-type NF catalyst exhibited zero degree of spin polarization. It implies that the *meso*-NF catalyst is entirely composed of materials lacking chirality, thereby indicating that there is no influence from up- and down-magnetization direction. We have incorporated discussion on *racemic* mixtures and *meso* compound into the manuscript, as suggested by the reviewer.

Fig. R22. (a) Current mapping of the *rac*-NEA 2D OIHP films obtained with up-magnetized Co-Cr-coated tip via mCP-AFM measurement. (b) Spin-dependent current behavior depending upon a pre-magnetization direction of tip at the different sub-regions of the *rac*-NEA 2D OIHP film.

Fig. R23. Spin-dependent current behavior depending upon a pre-magnetization direction of tip for (a) *L*-NF, (b) *D*-NF, (c) *rac*-NF and (d) *meso*-NF on the Ni foam measured using mCP-AFM in the range from -1.0 to 1.0 V.

Revision made (colored in blue):

(Supplementary Information; Supplementary Fig. S25c are added)

Supplementary Fig. S25. Spin-dependent current behavior depending upon a pre-magnetization direction of tip for (a) *D*-NF, (b) *rac*-NF and (c) *meso*-NF on the Ni foam measured using mCP-AFM in the range from -1.0 to 1.0 V.

(page 18, line 16)

In addition, the *L*-/*D*-NF catalysts showed high $P_{-0.75V}$ of 57.89% and 56.48%, respectively, as determined by mCP-AFM (Fig. 3c, Supplementary Fig. S25a), which were far superior to *rac*-NF and *meso*-NF ($P_{-0.75V}$ of 1.87% and 0%, respectively; Supplementary Fig. S25b). This signifies that the chirality of *L*-/*D*-NF is not an intrinsic property but rather induced by the presence of either *L*-TA or *D*-TA, respectively.

Comment 5:

While P3HT was used in partial replacement of the Spiro-MeOTAD to preserve spin polarization of the electrons, how does the Kapchen film impact spin transport? Was mCP-AFM experiments done to confirm the effect of Kapchen film, as well as the outer nickel layer?

Author's Response:

Kapchen is a Ketjen black of carbon nanoparticles. We correct the Kapchen to the more accurate term of Ketjen. For the Ketjen/S88:P12/NEA 2D/3D OIHP device, noise signal occurs during the measurement due to the tip scraping by the carbon powder, making mCP-AFM measurement challenging. Therefore, we fabricated the devices passivated with a bilayer of Ni foil/Ketjen (*i.e.*, Ni/Ketjen/S88:P12/NEA 2D/3D OIHP structure) to observe spin-polarized current based on the handedness of chiral NEA 2D (Fig. R24).

Even with the passivation of Ni/Ketjen, *S*-NEA 2D exhibited a spin polarization degree (P_{-10V}) of 45.88% (Fig. R24a), while *R*-NEA 2D showed a P_{-10V} of 42.33% (Fig. R24b), indicating that even with a passivation layer, the spin is maintained without significant spin relaxation during propagation. However, when using *rac*-NEA 2D, the P_{-10V} decreased to 1.42% (Fig. R24c), suggesting the absence of the CISS effect. Although the addition of the Ni/Ketjen layer results in a slight decrease in the degree of spin polarization, the spin relaxation time of carbon nanoparticles is sufficiently long (~ 100 ns) to enable spin transport (*Nat. Commun.* **2016**, *7*, 12232; *Nat. Commun.* **2023**, *14*, 2831), leading to the manifestation of the CISS-driven OER at the electrode/electrolyte interface.

Fig. R24. Spin-dependent current generated of Ni/Ketjen/S88:P12/NEA 2D/3D OIHP in which various configurations of NEA were utilized: (a) *S*-NEA 2D, (b) *R*-NEA 2D and (b) *rac*-NEA 2D. The mCP-AFM was measured in the range from -10 to 10 V.

Revision made (colored in blue):

(Supplementary Information; Supplementary Fig. S12 are added)

Supplementary Fig. S12. Spin-dependent current generated of Ni/Ketjen/S88:P12/NEA 2D/3D OIHP in which various configurations of NEA were utilized: (a) *S*-NEA 2D, (b) *R*-NEA 2D and (b) *rac*-NEA 2D. mCP-AFM was measured in the range from -10 to 10 V.

(page 11, line 23)

To establish an electrical connection between the Ni foil and the HTL, Ketjen (a highly conductive type of carbon) was incorporated to facilitate interlayer bonding. To investigate the extent of spin relaxation caused by Ni foil and Ketjen, we fabricated the devices passivated with a bilayer of Ni foil/Ketjen (*i.e.*, Ni/Ketjen/S88:P12/NEA 2D/3D OIHP structure) to observe spin-polarized current depending upon the handedness of chiral NEA 2D (Supplementary Fig. S12). Even with the passivation of Ni/Ketjen, *S*-NEA 2D exhibited a spin polarization degree (P_{-10V}) of 45.88% (Supplementary Fig. S12a), while *R*-NEA 2D showed a P_{-10V} of 42.33% (Supplementary Fig. S12b), indicating that even with a passivation layer, the spin is maintained without significant spin relaxation during propagation. However, when

using *rac*-NEA 2D, the P_{-10V} decreased to 1.42% (Supplementary Fig. S12c), suggesting the absence of the CISS effect. Although the addition of the Ni/Ketjen layer results in a slight decrease in the degree of spin polarization, the spin relaxation time of carbon nanoparticles is sufficiently long (~100 ns) to enable spin transport,^{39,40} leading to the manifestation of the CISS-driven OER at the electrode/electrolyte interface. To complete photoanode fabrication,

(References)

41. Náfrádi, B., Choucair, M., Dinse K. P. & Forró, L. Room temperature manipulation of long lifetime spins in metallic-like carbon nanospheres. *Nat. Commun.* **7**, 12232, (2016)

42. Márkus, B. G., Gmitra, M., Dóra, B., Csősz, G., Fehér, T., Szirmai, P., Náfrádi, B., Zólyomi, V., Forró, L., Fabian J. & Simon F. Ultralong 100 ns spin relaxation time in graphite at room temperature. *Nat. Commun.* **14**, 2831, (2023)

(Methods, page 38)

Following OIHP preparation, conductive carbon powder Ketjen was spread thereon, and then Ni foil was placed on top.

Comment 6:

Figure S5 shows the X-ray diffraction pattern in the reciprocal space. All the dots on the same ring should correspond to the same 2θ . The x label and y label should be q_{xyz} (momentum transfer), rather than 2θ .

Author's Response:

Supplementary Fig. S5 shows the analysis conducted using two-dimensional X-ray diffraction with a general area detector diffraction system (2D GADDS XRD), where 2D mapping is

possible as the detector plane angle (y label) moves. Depending on whether the pattern is dotted or ring-type, the analysis can determine the direction of crystal growth, either in-plane or out-of-plane. Therefore, Supplementary Fig. S5 differs from grazing-incidence wide-angle X-ray scattering (GIWAXS), and it is appropriate to use " 2θ " as the x label. To avoid confusion with XRD patterns in reciprocal space, the manuscript is revised to accurately reflect the method as "2D GADDS XRD." We appreciate the reviewer's meticulous comment

Revision made (colored in blue):

(page 8, line 18)

In the 2D XRD with general area detector diffraction system (2D GADDS XRD) analysis of the 3D OIHP in Supplementary Fig. S5a, ring-type peaks corresponding to the (001), (002), and (210) planes at 14.41° , 28.61° , and 35.01° , respectively, indicate that the crystal orientations are random. Meanwhile, after depositing the chiral molecules, the (002) and (004) planes of the 2D chiral OIHPs emerged as distinct dot-patterned peaks in the 2D GADDS XRD spectra in Supplementary Fig. S5b–d, which is consistent with the XRD spectrum in Fig. 1b.

(Supplementary Information; Caption of Supplementary Fig. S5 are modified)

Supplementary Fig. S5. 2D X-ray diffraction with general area detector diffraction system (2D GADDS XRD) spectra of (a) 3D OIHP, (b) BA 2D/3D OIHP, (c) MBA 2D/3D OIHP, and (d) NEA 2D/3D OIHP on FTO substrates.

Comment 7:

For the 2D/3D perovskite structure fabrication, the ink only includes chiral organic ammonium salts, so what is the formation mechanism of 2D perovskite? Ion exchange due to thermal diffusion? When 2D perovskites are formed, there should be a byproduct of achiral ammonium salts. Why not use the 2D perovskite solution as the precursor to directly form a 2D perovskite layer?

Author's Response:

To fabricate chiral 2D perovskite on top of the 3D perovskite ((FAPbI₃)_{0.95}(MAPbBr₃)_{0.05}), only chiral cation salt of either MBAI, NEABr, or BAI was dissolved in iso-propyl alcohol (IPA) to prepare chiral ligand solution. When dripping the IPA-based solution on the 3D perovskite, 2D PbX₂ (PbI₂, PbBr₂) forms an intermediate phase on the surface of the 3D perovskite, combined with IPA. Subsequently, amine ions (NH³⁺) involved in MBA, NEA, and BA, which is electrostatically interact to the PbX₂, replace the IPA between the IPA-PbX₂ intermediate, leading to the initiation of 2D layer. The intercalated bulk cation experiences van der Waals interactions between organic cations or π - π interactions between benzene rings, resulting in the formation of a thermodynamically stable layered structure.

When N,N-Dimethylformamide (DMF) is used as the solvent for the chiral ligand solution, the underlying 3D perovskite completely dissolves and disappears, as depicted in Fig. R25. Consequently, to deposit the chiral 2D perovskite on top of the 3D perovskite, a process based on the partial dissolution-involved interface formation needs to be employed. We have provided a more detailed explanation of this process in the manuscript.

Fig. R25. Photographs of 3D perovskite and 2D/3D perovskite in which S-NEA 2D layer was formed based on IPA solvent based chiral ligand solution. The 3D perovskite film after dripping of DMF-based S-NEA chiral ligand solution.

Revision made (colored in blue):

(Methods, page 38)

the sample was thermally annealed at 100 °C for 1 min, 120 °C for 3 min, or 100 °C for 1 min, respectively. When dripping the IPA-based solution on the 3D perovskite, 2D PbX₂ (PbI₂, PbBr₂) forms an intermediate phase on the surface of the 3D perovskite, combined with IPA. Subsequently, amine ions (NH³⁺) involved in MBA, NEA, and BA, which is electrostatically interact to the PbX₂, replace the IPA between the IPA-PbX₂ intermediate, leading to the initiation of 2D layer. The intercalated bulk cation experiences van der Waals interactions between organic cations or π - π interactions between benzene rings, resulting in the formation of a thermodynamically stable layered structure. Consequently,

Comment 8:

For the discussion of the X-ray diffraction (the end of page 7), The observation of (002) and (004) plane can't suggest that the layered inorganic framework is perpendicular to the substrate. Instead, the appearance of (002) and (004) suggests that the layered inorganic framework is parallel to the substrate.

Author's Response:

We appreciate the detailed comment regarding our chiral 2D perovskite. It has been confirmed that the layered inorganic framework on the (002) and (004) planes is oriented horizontally on the substrate. We have incorporated this observation into the revised manuscript.

Revision made (colored in blue):

(page 8, line 24)

Supplementary Fig. S5b–d, which is consistent with the XRD spectrum in Fig. 1b. This observation suggests that the 2D chiral OIHPs have grown horizontally on top of the 3D OIHP regardless of chiral molecule type. Under this circumstance,

Comment 9:

For the SAED (page 9, line 186), how was the crystallinity confirmed via FFT imaging of the white region? In terms of the FWHM?

Author's Response:

To calculate the crystal size of the 2D and 3D perovskites, we utilized the Scherrer equation:

$$\tau = \frac{K \lambda}{\beta \cos(\theta)}$$

τ represents the crystal size (in nanometers) of the ordered domains, K is a dimensionless shape factor (typically assumed to be 0.9 for perovskites), λ stands for the X-ray wavelength of Cu K α radiation (0.15405 nm), β denotes the line broadening at full-width half the maximum

intensity (FWHM), and θ is the Bragg angle. The FWHM of the (002) peak in the FFT image of the white region in Supplementary Fig. S8d was calculated using MATLAB (see Fig. R26). For S-NEA 2D, the β and θ values corresponding to the (002) plane were found to be 1.59 and 2.71 degrees, respectively, indicating a very small crystallite size of 0.0873 nm and thus a low crystallinity. This is a common issue that needs to be addressed in future research involving chiral 2D perovskites. Through additional analysis guided by the reviewer's advice, our understanding of crystallinity has been further strengthened.

Fig. R26. Calculation of the full-width half the maximum intensity (FWHM) of S-NEA 2D OIHP film from the FFT image of white region in Supplementary Fig. S8d.

Revision made (colored in blue):

(Supplementary Information; Supplementary Fig. S9 are added)

Supplementary Fig. S9. Calculation of the full-width half the maximum intensity (FWHM) of S-NEA 2D OIHP film from the FFT image of white region in Supplementary Fig. S8d.

(page 10, line 2)

The crystallinity was also confirmed via FFT imaging of the white region. Furthermore, to calculate the crystal size of the 2D and 3D perovskites, we utilized the Scherrer equation:

$$\tau = \frac{K \lambda}{\beta \cos(\theta)}$$

τ represents the crystal size (in nanometers) of the ordered domains, K is a dimensionless shape factor (typically assumed to be 0.9 for perovskites), λ stands for the X-ray wavelength of Cu $K\alpha$ radiation (0.15405 nm), β denotes the line broadening at full-width half the maximum intensity (FWHM), and θ is the Bragg angle. The FWHM of the (002) peak in the FFT image of the white region in Supplementary Fig. S8d was calculated using MATLAB (see Supplementary Fig. S9). For S-NEA 2D, the β and θ values corresponding to the (002) plane were found to be 1.59 and 2.71 degrees, respectively, indicating a very small crystallite size of 0.0873 nm and thus a low crystallinity. This is a common issue that needs to be addressed in future research involving chiral 2D perovskites. The SAED pattern in Supplementary Fig. S8e

Comment 10:

For the PL, the author claims that the PL revealed the suppressed recombination of photogenerated charges in the devices without the SFL. Is this accurate? The radiative recombination rate is an intrinsic property of the material and is hard to change, while the rate of non-radiative recombination will vary with the density of the defect states. The weakening of PL intensity means the enhancement of non-radiative recombination. The author can quantitatively see the recombination rate by testing the fluorescence lifetime.

Author's Response:

As the reviewer mentioned, we additionally conducted time-resolved photoluminescence (TRPL) measurements to quantitatively evaluate the recombination rate. The devices having a configuration of 3D, *S*-NEA 2D/3D, and *rac*-NEA 2D/3D films on SLG were utilized for the measurements, as demonstrated in Fig. R26. It is well-known that when the excitation fluence exceeds 10^{15} carriers per cm^3 , both mono- and bimolecular recombination coexist. In the case of the TRPL without a quencher, the fast lifetime (τ_1) possibly implies the non-radiative bimolecular recombination of charge carriers, while the slow lifetime (τ_2) represents the non-radiative monomolecular recombination of the carriers by trapping or detrapping. Therefore, the obtained curves were fitted with bi-exponential decay curve and the fitting results are summarized in Table R1. Both τ_1 and τ_2 for the *S*-NEA 2D/3D and *rac*-NEA 2D/3D films were found to be longer as compared to the 3D film without NEA 2D. These results suggest that the lifetime of photogenerated charge carriers was effectively elongated by introduction of NEA 2D, which likely results from the reduced defect density at the 2D/3D interface. Furthermore, the recombination process between bimolecular and monomolecular can be discernible by comparing the weight fraction of the two lifetime values. The weight fraction of τ_2 in 3D film is much higher (63.07%) than that of *S*-NEA 2D/3D (40.60%) and *rac*-NEA 2D/3D (40.19%).

This speculates that the 3D OIHP without the NEA 2D exhibits higher monomolecular recombination induced by higher trapping mechanism as compared to the 3D OIHP with NEA 2D. This observation highly supports our research in which the introduction of chiral NEA 2D OIHP results in the elongated lifetime of photogenerated charge carriers.

Fig. R26. Time-resolved PL spectra of 3D, *S*-NEA 2D/3D, and *rac*-NEA 2D/3D films on soda-lime glass (SLG) measured without a quencher.

Table. R1. TRPL fitting data of the fast (τ_1) and slow (τ_2) lifetime components and weight fraction (A) obtained using a biexponential model for the OIHP on the SLG samples.

	A ₁ (%)	τ_1 (ns)	A ₂ (%)	τ_2 (ns)
3D	36.93255	0.82	63.06745	3.33
S-NEA 2D/3D	59.39655	0.83	40.60345	4.53
rac-NEA 2D/3D	59.81247	0.83	40.18753	4.43

Revision made (colored in blue):

(Supplementary Information; Supplementary Fig. S20 are added)

Supplementary Fig. S20. Time-resolved PL spectra of 3D, *S*-NEA 2D/3D, and *rac*-NEA 2D/3D films on soda-lime glass (SLG) measured without a quencher.

(Supplementary Information; Supplementary Table S1 are added)

Supplementary Table. S1. TRPL fitting data of the fast (τ_1) and slow (τ_2) lifetime components and weight fraction (*A*) obtained using a biexponential model for the OIHP on the SLG samples.

	A_1 (%)	τ_1 (ns)	A_2 (%)	τ_2 (ns)
3D	36.93255	0.82	63.06745	3.33
S-NEA 2D/3D	59.39655	0.83	40.60345	4.53
rac-NEA 2D/3D	59.81247	0.83	40.18753	4.43

(page 16, line 1)

The results of the PL analysis indicate that the introduction of the *S*-NEA 2D SP enhances not only the lifetime of the spin-polarized charge carriers but also the charge extraction kinetics.

We additionally conducted time-resolved photoluminescence (TRPL) measurements to quantitatively evaluate the recombination rate. The devices having a configuration of 3D, *S*-NEA 2D/3D, and *rac*-NEA 2D/3D films on SLG were utilized for the measurements, as

demonstrated in Supplementary Fig. S20. It is well-known that when the excitation fluence exceeds 10^{15} carriers per cm^3 , both mono- and bimolecular recombination coexist. In the case of the TRPL without a quencher, the fast lifetime (τ_1) possibly implies the non-radiative bimolecular recombination of charge carriers, while the slow lifetime (τ_2) represents the non-radiative monomolecular recombination of the carriers by trapping or detrapping. Therefore, the obtained curves were fitted with bi-exponential decay curve and the fitting results are summarized in Supplementary Table S1. Both τ_1 and τ_2 for the *S*-NEA 2D/3D and *rac*-NEA 2D/3D films were found to be longer as compared to the 3D film without NEA 2D. These results suggest that the lifetime of photogenerated charge carriers was effectively elongated by introduction of NEA 2D, which likely results from the reduced defect density at the 2D/3D interface. Furthermore, the recombination process between bimolecular and monomolecular can be discernible by comparing the weight fraction of the two lifetime values. The weight fraction of τ_2 in 3D film is much higher (63.07%) than that of *S*-NEA 2D/3D (40.60%) and *rac*-NEA 2D/3D (40.19%). This speculates that the 3D OIHP without the NEA 2D exhibits higher monomolecular recombination induced by higher trapping mechanism as compared to the 3D OIHP with NEA 2D. This observation highly supports our research in which the introduction of chiral NEA 2D OIHP results in the elongated lifetime of photogenerated charge carriers.

(Methods, page 40)

the sample was prepared by subjecting it to a focused ion beam (Crossbeam 350, ZEISS, Germany) operating at an accelerating voltage of 2 kV. The energy level of each semiconductor was analyzed by ultraviolet photoelectron spectroscopy (UPS, Axis-NOVA, and Ultra DLD, UK) under He I radiation (21.21 eV) to investigate the energy band position of sample. The secondary-electron cutoff region (E_{cutoff}) and valence-band edge (E_{edge}) were calculated

according to Supplementary Note 2 (Supporting Information). TRPL spectra were recorded with an excitation beam wavelength of 340 nm (FluoroMaxPlus, Horiba, Kyoto, Japan).

Comment 11:

photons are preferentially directed toward the 3D/SLG interface in all three devices (i.e., the 3D OIHP, S-NEA 2D/3D OIHP, and rac-NEA 2D/3D OIHP devices) without traveling through the 2D OIHP layer.' is not clear to me. How can it be connected to the PL intensity?

Author's Response:

We understand the concern raised by the reviewer. In cases where photons are preferentially directed toward the 3D/SLG interface, they are absorbed by the 3D OIHP before transitioning to the 2D OIHP. However, the band gap of the 3D OIHP (1.6 eV) is smaller than that of the 2D OIHP (2.9 eV) (see Fig. R12a), indicating that photons with energies below the band gap of the 3D OIHP may not be absorbed. Nevertheless, to avoid potential confusion as noted by the reviewer, we have removed and revised this sentence in the manuscript.

Revision made (colored in blue):

(page 15, line 19)

thereby revealing the suppressed recombination of photogenerated charges in the devices without the SP. On the other hand, when back-excitation was employed (as depicted in Supplementary Fig. S12b) wherein photons are excited toward the interface between the substrate and the 3D OIHP, the absence of a quenching layer resulted in no significant difference in PL intensity (Supplementary Fig. S12c). However, after introducing both the HTL and SnO₂ quencher layer under front-excitation, the S-NEA 2D/3D OIHP film showed the

lowest PL intensity, indicating rapid charge extraction (the dashed line in Fig. 2c). The results of the PL analysis

<Reviewer 4>

Remark:

We thank the reviewer for reviewing our work and providing insightful comments and constructive suggestions. We have carefully revised our manuscript according to the suggestions, with more specific analysis and profound consideration. We sincerely hope that the details provided below will satisfy the reviewer's concerns.

REVIEWER COMMENTS

Reviewer #1 (Remarks to the Author):

The authors have addressed my comments. The manuscript can be accepted for publication now.

Reviewer #2 (Remarks to the Author):

The authors have successfully addressed my comments with key additions. However, I have a doubt. It is said that the active area is of 0.06 cm². This is quite small, compared to what it is seen in the pictures. Is this just the area of the photoactive absorber while the catalyst part offers larger area to the electrolyte? This should be specified

Reviewer #3 (Remarks to the Author):

The authors have made substantial amount of effort in addressing the comments and concerns raised by the Reviewers. I have several follow-up comments, that should be clarified before publication.

Regarding to the reply to Comment 6: Could the author provide any literature where 2θ is used as the x label. In general, in 2D X-ray diffraction mapping, unless it is explicitly stated that y equals 0, the x axis cannot represent the 2θ .

Regarding to the reply to Comment 7: the author answered why they chose IPA rather than DMF to coat 2D layer on the top of 3D layer, but my question is why the author only added the chiral cation salt in IPA without PbX₂. Also, the formation process provided by the author can only explain the formation of the first layer of the 2D perovskite. Once 2D layer is formed, the original PbX₂ in 3D perovskite is protected by the long organic chains and no more 2D perovskite could form due to the lack of Pb source. That's the reason I ask why not add Pb source in IPA.

Regarding to the reply to Comment 10: The analysis of the TRPL is correct but the description is not accurate enough. The author claims that the defect density is lower in their 2D/3D structure, which is confirmed by the slower decay. But they also state that thereby revealing the suppressed recombination of photo-generated charges in the devices without the SP. The recombination rate is, by definition, equal to the reciprocal of the lifetime. Suppressed recombination means a longer lifetime in 3D structure without SP, which is not true. The author should modify this sentence to make it more accurate.

Reviewer #4 (Remarks to the Author):

Response Letter

Journal: Nature communications

Previous Manuscript ID: NCOMMS-23-52830A

Title: A Dual Spin-Controlled Chiral 2D/3D Perovskite Artificial Leaf for Efficient Overall Photoelectrochemical Water Splitting

Author(s): *Hyungsoo Lee, Chan Uk Lee, Juwon Yun, Chang-Seop Jeong, Wooyong Jeong, Jaehyun Son, Young Sun Park, Subin Moon, Soobin Lee, Jun Hwan Kim, and Jooho Moon*

<Reviewer 2>

The authors have successfully addressed my comments with key additions. However, I have a doubt. It is said that the active area is of 0.06 cm^2 . This is quite small, compared to what it is seen in the pictures. Is this just the area of the photoactive absorber while the catalyst part offers larger area to the electrolyte? This should be specified

Remark:

We would like to thank the reviewer for evaluating our work. As reviewer suggested, we provide the experimental details. Our response to the reviewer's comments can be found below.

Author's Response:

As per the reviewer's comment, we are aware of the importance of ensuring that the light absorption area of the 3D perovskite matches the active area of catalyst exposed to the electrolyte where electrochemical reaction occurs. Photogenerated charge carriers form as the 3D perovskite absorbs the incoming light. If the active area of the catalyst is smaller than the light absorption area, bottleneck effects may occur, resulting in an increased resistance per unit area due to a larger quantity of carriers passing through. This could lead to rapid degradation of the water splitting device. On the other hand, attaching a catalyst with a larger active area than the light absorption area may facilitate charge transfer, but also may artificially broaden the electrochemical active site, making it difficult to make an accurate assessment.

Except for Supplementary Fig. S40 which was measured under outdoor conditions, all photoanodes used for PEC performance measurements were conducted with a metallic photomask attached to the sample to define the specific light absorption area of 0.06 cm^2 , as depicted in Figure R1a. Additionally, the chiral *L*-NF catalyst matches the area of 0.06 cm^2 ,

then attaching onto the top surface of the photoanodes. Furthermore, all electrodes except for the active area were passivated with epoxy to prevent the multi-layered photoanode from exposing to the electrolyte.

The photoanode presented in Supplementary Fig. S40 was specifically designed for outdoor condition measurement. It is difficult to observe hydrogen/oxygen bubbles generated from small active area of 0.06 cm^2 under natural sunlight. For clear demonstration of hydrogen generation under outdoor condition, we fabricated the absorbers with a light absorption area and catalyst size with each active area of 0.5 cm^2 , as depicted in Figure R1b, and then we filmed the unbiased hydrogen generation on the video. To avoid the confusion regarding the active area mismatch, we have incorporated the explanation into the revised manuscript.

Fig. R1. (a) Photographs of photomask with an area of 0.06 cm^2 to define the light absorption area and front side view of the OIHP photoanode with catalyst. (b) A photograph of the photoanode water-splitting device with an active area of 0.5 cm^2 that can be operated under outdoor sunlight.

Revision made (colored in blue):

(Supporting Information; Supplementary Fig. S13a is added)

Supplementary Fig. S13. (a) Photograph of photomask with an area of 0.06 cm^2 to define the light absorption area. Linear sweep voltammetry (LSV) curves of (b) *S*-NEA 2D/3D OIHP, (c) *R*-NEA 2D/3D OIHP, and (d) *rac*-NEA 2D/3D OIHP photoanodes with various spiro-MeOTAD:P3HT volume ratios in a K-Pi electrolyte (pH 6.5) under 1 sun illumination.

(Caption of Fig. 2 is modified)

Fig. 2. The PEC performance of the 2D chiral OIHP-based photoanodes and the effect of spin polarization therein. (a) A schematic illustration of a photoanode comprising NiFeOOH/Ni/Ketjen/HTL/NEA-2D/3D/SnO₂/FTO layers. (b) LSV curves of the OIHP photoanodes without and with the *S*-NEA 2D OIHP layer using various spiro-MeOTAD and P3HT formulations for the HTL conducted in K-Pi electrolyte (pH 6.5) under 1 sun illumination with an active area of 0.06 cm^2 . (c) Steady-state PL spectra of 3D OIHP, *S*-NEA 2D/3D OIHP, and *rac*-NEA 2D/3D OIHP films with (w/) and without (w/o) a quencher on SLG or an SnO₂ substrate with an S88:P12 HTL, respectively.

(Caption of Fig. 2 is modified)

Fig. 3. The chirality of the prepared NiFeOOH catalysts and their application in OIHP-based photoanodes. (a) LSV curves and (b) the corresponding Tafel plots of *L*-NF (dark red) and *meso*-NF (gray) on Ni foam. (c) The spin-polarization degrees of *L*-NF/Ni and *meso*-NF/Ni measured using mCP-AFM. (d) LSV curves of *L*-NF/*S*-NEA 2D/3D (red), *meso*-NF/*S*-NEA 2D/3D (green), *L*-NF/3D (gray), and *meso*-NF/3D (orange) photoanodes with an active area of 0.06 cm². (e) O₂ generation during the PEC reaction by the *L*-NF/*S*-NEA 2D/3D and *L*-NF/3D photoanodes detected using an oxygen sensor. The Faradaic efficiency of O₂ generation by both photoanodes was calculated by considering the theoretical O₂ generation. All of the electrochemical measurements were conducted in 0.5 M K-Pi (pH 6.5) under 1 sun illumination.

(Supporting Information; Caption of Supplementary Fig. S16 is modified)

Supplementary Fig. S16. As the concentration of organic *S*-NEA cation increases (from 2.5 to 7.5 mg), it is possible to control the thickness of *S*-NEA layer on top of the 3D OIHP. LSV curves of *S*-NEA/3D photoanodes as a function of the thickness of *S*-NEA layer under 1 sun illumination in a K-Pi electrolyte (pH 6.5) with an area of 0.06 cm².

(Supporting Information; Caption of Supplementary Fig. S29 is modified)

Supplementary Fig. S29. LSV curves of *L*-NF/*S*-NEA 2D/3D (red-solid), *L*-NF/*R*-NEA 2D/3D (red-dotted), *D*-NF/*R*-NEA 2D/3D (blue-solid), and *D*-NF/*S*-NEA 2D/3D (blue-dotted) photoanodes with an area of 0.06 cm².

(Supporting Information; Caption of Supplementary Fig. S32 is modified)

Supplementary Fig. S32. (a) Operational stability of *L*-NF/*S*-NEA 2D/3D OIHP photoanode under continuous operation in 0.5 M K-Pi (pH 6.5) electrolyte at 1.2 V_{RHE} in an ambient atmosphere with an area of 0.06 cm². (b) Photographs of front and back of the OIHP photoanode after the stability test. (c) LSV curves of *L*-NF before and after stability test.

(Supporting Information; Caption of Supplementary Fig. S40 is modified)

Supplementary Fig. S40. A photograph of the co-planar photocathode-photoanode water-splitting device that can operate under outdoor sunlight. The device had a total active area of 1 cm² (the summation of the photocathode area (0.5 cm²) and photoanode area (0.5 cm²)).

(page 13, line 5)

Fig. 2b shows linear sweep voltammetry (LSV) measurements of the photoanodes in 0.5 M potassium phosphate buffer (K-Pi) electrolyte (pH 6.5). All of the photoanodes were analyzed for the same light absorption area of 0.06 cm², as defined by a metallic photomask (Supplementary Fig. S13a). Additionally, the active area of the catalyst also matched 0.06 cm². A photocurrent density (J_{ph}) of 14.27 mA cm⁻² at 1.23 V_{RHE} along with a V_{onset} of 0.4 V_{RHE} was observed...

<Reviewer 3>

The authors have made substantial amount of effort in addressing the comments and concerns raised by the Reviewers. I have several follow-up comments, that should be clarified before publication.

Remark:

We would like to thank the reviewer for helpful comments and evaluating our study as a detailed study. The quality of our manuscript has improved according to the reviewer's comments. The followings are the answers to the reviewer's particular questions:

Comment 1:

Regarding to the reply to Comment 6: Could the author provide any literature where 2θ is used as the x label. In general, in 2D X-ray diffraction mapping, unless it is explicitly stated that γ equals 0, the x axis cannot represent the 2θ .

Author's Response:

The 2D X-ray diffraction (XRD) measurements we conducted utilize the general area detector diffraction system (GADDS), employing a scanning method where the detector scans the reflection angle of 2θ (*Coatings* **6**, 54 (2016); *Geochem. Geophys. Geosyst.* **16**, 3778–3788 (2015)). The 2D XRD with GADDS involves measuring intensity variations as a function of the γ -angle (the angle tilted with respect to the substrate) and 2θ , analyzing the crystalline characteristics of materials in both out-of-plane ($\gamma=90^\circ$) and in-plane directions ($70^\circ < \gamma < 90^\circ$, $90^\circ < \gamma < 110^\circ$) (Fig. R2).

Numerous studies have reported the analysis of 2D XRD, including investigations into perovskite materials (*J. Mater. Chem. A* **7**, 8218–8225 (2019); *Adv. Energy Mater.* **9**, 1901719

(2019)). In response to the reviewer's comments, we recognized the omission of defining the x-axis as 2θ without a definition for the y-axis, so that we have added explanation for the y-axis, specifically the γ -angle. Additionally, we have revised the labeling of the x-axis as 2θ based on references from previous studies and have provided detailed modifications to the experimental conditions (Fig. R3). Following the reviewer's advice, we have focused on analyzing the crystallography of the perovskite films.

Fig. R2. Experimental setup for conducting 2D X-ray diffraction using the general area detector diffraction system: ω represents the incident angle at which the X-ray beam is directed onto the sample surface. The 2θ refers to the detector angle at which the X-ray beam is diffracted and detected by the detector. The γ -angle indicates the tilted angle with respect to the substrate.

Fig. R3. 2D X-ray diffraction spectra with general area detector diffraction system (2D GADDS XRD) of (a) 3D OIHP, (b) BA 2D/3D OIHP, (c) MBA 2D/3D OIHP, and (d) NEA 2D/3D OIHP on FTO substrates conducted with $70^\circ < \gamma < 110^\circ$.

Revision made (colored in blue):

(Supporting Information; Supplementary Fig. S5 are modified)

Supplementary Fig. S5. 2D X-ray diffraction spectra with general area detector diffraction system (2D GADDS XRD) of (a) 3D OIHP, (b) BA 2D/3D OIHP, (c) MBA 2D/3D OIHP, and (d) NEA 2D/3D OIHP on FTO substrates conducted with $70^\circ < \gamma < 110^\circ$.

(Methods, page 40, line 8)

2D-XRD patterns were collected using a D8-Discover instrument (Bruker, USA) equipped with a GADDS four-circle detector. The 2D XRD spectra with GADDS were measured as a function of the γ -angle, which indicates the tilted angle with respect to the substrate, from 70° to 110° . HR-TEM and elemental mapping were conducted on a JEM-ARM200F (JEOL, Tokyo, Japan) with an integrated EDX system at an accelerating voltage of 200 kV.

Comment 2:

Regarding to the reply to Comment 7: the author answered why they chose IPA rather than DMF to coat 2D layer on the top of 3D layer, but my question is why the author only added the chiral cation salt in IPA without PbX₂. Also, the formation process provided by the author can only explain the formation of the first layer of the 2D perovskite. Once 2D layer is formed, the original PbX₂ in 3D perovskite is protected by the long organic chains and no more 2D perovskite could form due to the lack of Pb source. That's the reason I ask why not add Pb source in IPA.

Author's Response:

We appreciate the detailed comment. We had attempted to prepare a 2D perovskite layer of (NEA)₂PbBr₄ composition by adding PbBr₃ to isopropyl alcohol (IPA). However, as shown in Fig. R4, only NEABr was dissolved in IPA, while PbBr₃ remained without dissolution. Experimentally, we suffered from the insolubility of PbX₃ salt in IPA. This is consistent with the assertion made by Edward H. Sargent *et al.* (*Nat. Commun.* **12**, 3472 (2021)) that PbX₂ frameworks do not dissolve in IPA and rather remain. When dripping the IPA-based solution containing the NEA chiral cation onto the 3D perovskite ((FAPbI₃)_{0.95}(MAPbBr₃)_{0.05}), the surface of the 3D perovskite partially underwent to be composed into FAI, MABr, and PbX₂. Upon the encounter between IPA and PbX₂ during the dripping, the IPA molecule intercalates with PbX₂ (PbI₂ or PbBr₂) to form an intermediate phase of 2D IPA-PbX₂ on the surface of the 3D perovskite. Subsequently, amine ions (NH³⁺) involved in NEA chiral cations contained in IPA-based dripping solution replace the 2D IPA-PbX₂ intermediate by electrostatically interacting with PbX₂, leading to the formation of the NEA 2D perovskite. The intercalated chiral cations experience van der Waals interactions between organic cations or π - π interactions between benzene rings, resulting in a thermodynamically stable layered structure with chirality.

This film formation process is sufficient for the fabrication of a nano-scale conformal 2D chiral perovskite layer on the 3D perovskite.

Various studies have successfully fabricated the 2D perovskite layers on top of the 3D perovskites using organic cation-containing solution systems. The thickness of the 2D perovskite layer can be readily adjustable by varying the concentration of the organic cation-dissolved solution (*Adv. Funct. Mater.* **28**, 17, 1706923 (2018)). This approach has been reported to achieve the 2D perovskite layers with the thickness ranging from 10 to 60 nm (*Chem* **7**, 1903-1916 (2021)), which are actively employed in numerous studies involving 2D/3D perovskites (*Nat. Energy* (2024). <https://doi.org/10.1038/s41560-024-01470-5>; *Nat. Energy* **5**, 79–88 (2020)). Therefore, our chiral 2D perovskite layer with a thickness of 10 nm is sufficiently thin and can be reproducibly fabricated by dripping the chiral organic cation-dissolved IPA solution.

To demonstrate the successful fabrication of the chiral 2D/3D perovskite, X-ray diffraction (XRD) measurements were conducted. For the device with only 3D perovskite, as depicted in Fig. R5, distinct (100), (110), (111), (002), (210), and (211) planes of the 3D perovskite are observed. Also, for the device with chiral 2D/3D perovskite, the planes of the 3D perovskite are clearly observed, along with the distinct observation of the (002) and (004) planes of chiral NEA cation-based 2D perovskite. Scanning electron microscopy (SEM) was performed to confirm the morphology of chiral 2D/3D perovskite. As shown in Fig. R6a, the 3D OIHP layer had grown uniformly on the SnO₂/FTO substrate without pin-holes. After fabrication of 2D chiral OIHP on the 3D OIHP, it is observed that a thin film forms on top of the 3D perovskite, as shown in Fig. R6b. Cross-sectional SEM images showed that the 3D perovskite was fabricated with a thickness of ~340 nm (Fig. R6c). After the deposition of the NEA-based 2D chiral perovskite, a conformal thin film with a thickness of ~10 nm was observed on the upper surface of the 3D perovskite, as shown in Fig. R6d. To clarify the 2D

and 3D perovskite, we performed the backscattered SEM images (COMPO mode). In the COMPO mode of cross-sectional SEM analysis, dark color typically signifies materials composed of light-weight atoms with low density. In Fig. R6e, the prominent bright color observed in the 3D/SnO₂/FTO device indicates that the 3D perovskite is composed of heavy atoms with high density, whereas the 2D/3D/SnO₂/FTO device displays a thin and dark layer onto the 3D perovskite in Fig. R6f. The thin and dark contrast of the 2D perovskite compared to the 3D perovskite is attributed to the lower density of the 2D perovskite, resulting from the presence of a greater quantity of organic cations in the layered structure. This observation corresponds to our TEM images (Supplementary Fig. S8), confirming the successful fabrication of a chiral 2D perovskite layer with a thickness of approximately 10 nm. Supplementary data have been included to address the insufficient evidence concerning the outcomes of the 2D perovskite layer fabricated on the 3D perovskite layer. We appreciate the reviewer for the valuable comment.

Fig. R4. Photographs showing the solutions containing (a) NEABr, (b) NEABr and PbBr₃, and (c) PbBr₃ in iso-propyl alcohol (IPA) solvent.

Fig. R5. XRD spectra of 3D OIHP and the NEA 2D chiral OIHP layer on top of 3D OIHP.

Fig. R6. Surface scanning electron microscopy (SEM) images of (a) 3D OIHP and (b) NEA 2D/3D OIHP on a silicon dioxide (SnO₂)/FTO substrate. The cross-sectional SEM images of (c) 3D OIHP and (d) NEA 2D/3D OIHP on a SnO₂/FTO substrate. The cross-sectional backscattered SEM images (COMPO mode) of (e) 3D OIHP and (f) NEA 2D/3D OIHP on a SnO₂/FTO substrate.

Revision made (colored in blue):

(Supporting Information; Supplementary Fig. S6 are added)

Supplementary Fig. S6. Surface scanning electron microscopy (SEM) images of (a) 3D OIHP and (b) NEA 2D/3D OIHP on a silicon dioxide (SnO₂)/FTO substrate. The cross-sectional SEM images of (c) 3D OIHP on a SnO₂/FTO substrate. The cross-sectional backscattered SEM images (COMPO mode) of (d) 3D OIHP and (e) NEA 2D/3D OIHP on a SnO₂/FTO substrate.

(page 9, line 5)

Scanning electron microscopy (SEM) revealed differences in the microstructure of bare 3D OIHP and 2D chiral OIHP-coated 3D OIHP. As shown in Supplementary Fig. S6a, the 3D OIHP layer had grown uniformly on the SnO₂/FTO substrate without pin-holes. After fabrication of 2D chiral OIHP on the 3D OIHP, it is observed that a thin film forms on top of the 3D perovskite, as shown in Supplementary Fig. S6b. Cross-sectional SEM images showed that the 3D perovskite was fabricated with a thickness of ~340 nm (Supplementary Fig. S6c). After the deposition of the NEA-based 2D chiral perovskite, a conformal thin film with a thickness of ~10 nm was observed on the upper surface of the 3D perovskite, as shown in Fig.

1c. To clarify the 2D and 3D perovskite, we performed the backscattered SEM images (COMPO mode). In the COMPO mode of cross-sectional SEM analysis, dark color typically signifies materials composed of light-weight atoms with low density. In Supplementary Fig. S6d, the prominent bright color observed in the 3D/SnO₂/FTO device indicates that the 3D perovskite is composed of heavy atoms with high density, whereas the 2D/3D/SnO₂/FTO device displays a thin and dark layer onto the 3D perovskite in Supplementary Fig. S6e. The thin and dark contrast of the 2D perovskite compared to the 3D perovskite is attributed to the lower density of the 2D perovskite, resulting from the presence of a greater quantity of organic cations in the layered structure. AFM topographical analysis

(Methods, page 37, line 22)

To prepare the 2D chiral perovskite layers as following the previous reports,^{56,57} either MBAI (2.5 mg), NEABr (2.5 mg), or BAI (2.5 mg) chiral cation salt was dissolved in 1 mL of IPA under continuous stirring.

(References)

- 56 Proppe, A. H., Johnston, A., Teale, S., Mahata, A., Bermudez, R. Q., Jung, E. H., Grater, L., Cui, T., Filleter, T., Kim, C. Y., Kelley, S. O., Angelis, F. D. & Sargent, E. H. Multication perovskite 2D/3D interfaces form via progressive dimensional reduction. *Nat Commun* **12**, 3472 (2021).
- 57 Sutanto, A. A., Caprioglio, P., Drigo, N., Hofstetter, Y. J., Benito, I. G., Queloz, V. I. E., Neher, D., Nazeeruddin, M. K., Stolterfoht, M., Vaynzof, Y., & Grancini, G. 2D/3D perovskite engineering eliminates interfacial recombination losses in hybrid perovskite solar cells. *Chem* **7**, 1903-1916 (2021).

Comment 3:

Regarding to the reply to Comment 10: The analysis of the TRPL is correct but the description is not accurate enough. The author claims that the defect density is lower in their 2D/3D structure, which is confirmed by the slower decay. But they also state that “thereby revealing the suppressed recombination of photo-generated charges in the devices without the SP”. The recombination rate is, by definition, equal to the reciprocal of the lifetime. Suppressed recombination means a longer lifetime in 3D structure without SP, which is not true. The author should modify this sentence to make it more accurate.

Author’s Response:

We appreciate the critical comment regarding the explanation of the TRPL and PL analysis resulting from the introduction of SP on 3D perovskite. We have also recognized the insufficient description on the fact that recombination is suppressed in the 3D structure without SP. We have revised the manuscript to clarify that the longer lifetime and increased intensity observed upon the addition of SP indicate the suppressed recombination in the 2D/3D structure. We express our gratitude to the reviewer for his/her valuable comment.

Revision made (colored in blue):

(page 16, line 5)

Compared to the device without an SP, the introduction of the *rac*-NEA-based 2D SP caused a slight increase in the PL intensity whereas using the *S*-NEA-based 2D SP provided the highest PL intensity, **thereby revealing the suppressed recombination of photogenerated charges in the devices with the SP.** On the other hand,

REVIEWERS' COMMENTS

Reviewer #2 (Remarks to the Author):

No comments. The authors already addressed my comments in the previous version, and I don't see any problem in the latest submission

Reviewer #3 (Remarks to the Author):

The authors have addressed my comments adequately, and the paper is ready for publication.

Reviewer #4 (Remarks to the Author):

Response Letter

Journal: Nature communications

Previous Manuscript ID: NCOMMS-23-52830B

Title: A Dual Spin-Controlled Chiral 2D/3D Perovskite Artificial Leaf for Efficient Overall Photoelectrochemical Water Splitting

Author(s): *Hyungsoo Lee, Chan Uk Lee, Juwon Yun, Chang-Seop Jeong, Wooyong Jeong, Jaehyun Son, Young Sun Park, Subin Moon, Soobin Lee, Jun Hwan Kim, and Jooho Moon*

<Reviewer 2>

No comments. The authors already addressed my comments in the previous version, and I don't see any problem in the latest submission.

Remark:

We would like to express our gratitude to the reviewer for evaluating our work. Thanks to the comments provided previously by the reviewer, our research has been strengthened.

<Reviewer 3>

The authors have addressed my comments adequately, and the paper is ready for publication.

Remark:

We would like to thank the reviewer for helpful comments and evaluating our study as a detailed study. The quality of our manuscript has improved according to the reviewer's previous comments.

Revision made (colored in blue):

(Numbering of the affiliations for the authors)

Hyungsoo Lee^{1†}, Chan Uk Lee^{1†}, Juwon Yun¹, Chang-Seop Jeong¹, Wooyong Jeong¹, Jaehyun

Son¹, Young Sun Park¹, Subin Moon¹, Soobin Lee¹, Jun Hwan Kim¹, and Jooho Moon^{1}*

¹Department of Materials Science and Engineering, Yonsei University

50 Yonsei-ro Seodaemun-gu, Seoul, 03722, Republic of Korea

(Abstract, line 8)

The 3D perovskite serves as a light absorber while the 2D chiral perovskite functions as a spin polarizer to align the spin states of charge carriers. Compared to other investigated chiral organic cations, *R-/S-naphthyl ethylamine* enable strong spin-orbital coupling due to strengthened π - π stacking interactions. The resulting *naphthyl ethylamine*-based chiral 2D/3D perovskite photoelectrodes achieved a high spin polarizability of 75%.

(Modified the Supplementary Fig. S23)

Supplementary Fig. S23. (a) XRD and (b) Raman spectra of *L*-NiFeOOH on Ni foam. And XPS spectra of *L*-NiFeOOH and *meso*-NiFeOOH: deconvolution of the (c) Ni 2*p*, (d) Fe 2*p*, and (e) O 1*s* regions.

(page 18, line 11, XPS analysis)

Furthermore, we performed X-ray photoelectron spectroscopy (XPS) analysis. The results revealed identical XPS spectra for both *L*-NF and *meso*-NF. Peaks corresponding to Ni³⁺ 2*p*_{1/2} and 2*p*_{3/2} were observed at 867.5 and 854.1 eV, respectively, while a peak at 851.6 eV originated from Ni⁰ in the Ni foil substrate (Supplementary Fig. S23c). Similarly, peaks for Fe³⁺ 2*p*_{1/2} and 2*p*_{3/2} were observed at 722.4 and 709.9 eV, respectively (Supplementary Fig. S23d). Finally, the O 1*s* peak was deconvoluted into peaks at 530.8 and 529.5 eV (Supplementary Fig. S23e),

representing the metal-O bond (O^{2-}) and metal-hydroxyl bond ($-OH$), respectively.

(Fig. 5e was changed to Supplementary Fig. S42)

Supplementary Fig. S42. An overall water-splitting performance comparison of previously reported unassisted PEC-PEC systems with our device. The size of the circle corresponds to the STH retention ratio of the devices. j , retention photocurrent density; j_0 , initial photocurrent density.

(page 28, line 25)

Hence, our unassisted water-splitting device exhibited excellent water-splitting performance that outperforms previous OIHP-based photoelectrodes and is even comparable to the STH efficiency achieved with expensive III-V group semiconductors (Supplementary Fig. S42). Details of the overall PEC-PEC water-splitting devices shown in Supplementary Fig. S42 are summarized in Supplementary Table S3.